# Revisiting Actor-Critic Methods in Discrete Action Off-Policy Reinforcement Learning

## Abstract

Value-based approaches such as `DQN` are the default methods for off-policy reinforcement learning with discrete-action environments such as Atari. While soft actor-critic (`SAC`) is highly effective in continuous control, its discrete counterpart (`DSAC`) performs surprisingly poorly on challenging discrete-action domains such as Atari. Consequently, starting from `DSAC`, we revisit the design of actor-critic methods in this setting. First, we determine that the coupling between the actor and critic entropy is the primary reason behind the poor performance of `DSAC`. We demonstrate that by merely decoupling these components, `DSAC` can have comparable performance as `DQN`. Motivated by this insight, we introduce a flexible off-policy actor-critic framework that subsumes `DSAC` as a special case. Our framework allows using an $m$-step Bellman operator for the critic update, and enables combining standard policy optimization methods with entropy regularization to instantiate the resulting actor objective. Theoretically, we prove that the proposed methods can guarantee convergence to the optimal regularized value function in the tabular setting, matching the convergence rates in prior work. Empirically, we demonstrate that these methods can approach the performance of `DQN` on standard Atari games, and do so even without entropy regularization or explicit exploration.

## 1 Introduction

Value-based algorithms such as `DQN` (Mnih et al., 2013) and its derivatives (`Rainbow` (Hessel et al., 2018), `IQN` (Dabney et al., 2018), `M-DQN` (Vieillard et al., 2020b)) are commonly used in deep reinforcement learning. These methods can effectively learn from off-policy data, making them sample-efficient in complex environments. In particular, they are well-suited for environments with discrete actions and have achieved strong performance on large-scale benchmarks such as Atari 2600 (Bellemare et al., 2013).

In contrast, common policy-based methods such as `PPO` (Schulman et al., 2017), `TRPO` (Schulman, 2015) are on-policy and do not effectively reuse the data collected by past policies, making them sample-inefficient. Although prior work has developed off-policy variants of these popular algorithms (Queeney et al., 2021; Meng et al., 2023; Gan et al., 2024), these methods have not been thoroughly evaluated in discrete-action settings. On the other hand, several works have primarily focused on off-policy actor-critic techniques in continuous action spaces. In this setting, soft-actor critic (`SAC`) achieves strong empirical performance on standard control benchmarks. More recent works in this setting include (Zhu et al., 2024; Vieillard et al., 2021). While Zhu et al. (2024) introduce broader $q$-exponential policy families that generalize beyond Gaussian parametrization, Vieillard et al. (2021) extend the value-based `M-DQN` framework to a continuous action actor-critic method.

The strong empirical performance of `SAC` has motivated prior work to adapt it to offline RL (Xiao et al., 2023) as well as the online off-policy discrete-action setting (Christodoulou, 2019) (the focus of our work). However, the resulting discrete variant of `SAC` (abbreviated as `DSAC`) has poor empirical performance on standard benchmarks such as Atari. There have been numerous other attempts to design an effective off-policy actor-critic algorithm in

this setting. For example, Neumann et al. (2018) proposes an alternative algorithm called `GreedyAC`. This algorithm samples actions from an entropy-regularized proposal policy and maximizes the actor policy's log-likelihood on those samples (without employing an additional entropy regularization). Although `GreedyAC` offers a new perspective, it has not been evaluated on large-scale benchmarks such as Atari, nor directly compared to established value-based methods like `DQN`.

Other lines of work, including alternative discrete-action extensions of `SAC` (Wang et al., 2016; Xu et al., 2021; Zhou et al., 2022), do report results on large-scale domains against `DQN`, but introduce multiple interacting components which complicate implementation and can limit practical performance. For example, the `ACER` algorithm in Wang et al. (2016) requires off-policy Retrace (Munos et al., 2016), bias correction, and a dueling network architecture, while `SD-SAC` in Zhou et al. (2022) requires ad-hoc actor regularization and double-averaged $Q$-clipping. Motivated by the gap between off-policy value-based and policy-based methods, we aim to answer the question:

*Can we design simple actor-critic methods that can approach the performance of `DQN` in the off-policy discrete-action setting?*

Given the recent interest in applications such as reinforcement learning with human feedback (Ouyang et al., 2022), developing better actor-critic methods for discrete action environments is increasingly important. Moreover, doing so can also help address the limitations of value-based methods. For example, `DQN` and its successors lack a principled mechanism for exploration, often relying on $\epsilon$-greedy strategies that require manual tuning and are known to be brittle (Hessel et al., 2018). On the other hand, actor-critic methods such as `SAC` rely on entropy regularization in both the actor and critic updates. Importantly, these methods do not require explicit exploration making them potentially easier to tune on a new environment. Furthermore, when used with complex function approximation, value-based methods are prone to the *delusional bias* (Lu et al., 2018), which arises from independently performing a greedy update per next state. This ignores the joint distribution over states and actions and may yield a target $Q$-function that is not realizable by the function class. In contrast, `SAC` avoids the delusional bias by backing up the value function under the current actor policy rather than using greedy max-based backups.

To make progress towards our goal, we first investigate the poor empirical performance of discrete `SAC` (`DSAC`) in Christodoulou (2019) on Atari. Prior work attributes the poor performance of `DSAC` to the use of a fixed target entropy (Xu et al., 2021) and $Q$-value underestimation bias (Zhou et al., 2022). Proposed remedies include adaptive entropy targets (Xu et al., 2021) and entropy coefficient regularization with a clipped double-averaged $Q$ objective (Zhou et al., 2022). Recent work (Neumann et al., 2025) modifies the `DSAC` objective, and propose surrogates that include an explicit KL regularization term. While all such modifications can improve the method's stability, they introduce additional hyper-parameters, complicating the method without matching the performance of `DQN`. Furthermore, these modifications lack any theoretical justification even in the simple tabular setting.

**Contribution 1**: We perform an extensive ablation study on `DSAC`. Our results show that simply disabling the entropy regularization in the critic update (i.e., using the standard Bellman operator for policy evaluation) while keeping all other components (entropy-regularized soft actor update, automatic entropy tuning) fixed yields a stable variant of `DSAC`. This variant does not require either double $Q$-learning or additional hyper-parameters, and is competitive with `DQN` across Atari games (see Fig. 1). We also find that `DSAC` with a carefully tuned per-game critic entropy coefficient can achieve similar performance as `DQN`. Hence, we conclude that critic entropy substantially impacts the empirical performance of `DSAC`.

In order to explain the good empirical performance of the proposed `DSAC` variant, and to systematically design related algorithms, it is necessary to develop a more general approach. Prior works (Vieillard et al., 2020a; Xiao, 2022) have proposed actor-critic frameworks in the discrete action setting. While Vieillard et al. (2020a) show that `SAC` falls within their general framework, the proposed `DSAC` variant can not be captured by this framework. In

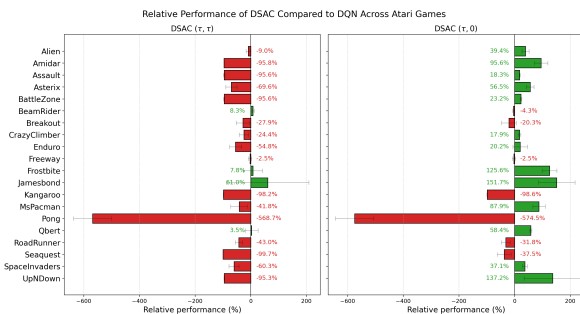

**Figure 1:** Performance of `DSAC` relative to `DQN` across 20 Atari games, with and without critic entropy. The left plot shows that the default `DSAC` underperforms `DQN` on most games, whereas incorporating a hard Bellman operator during policy evaluation (right plot) substantially improves `DSAC`'s performance.

particular, in Vieillard et al. (2020a), the actor and critic entropy is closely coupled, and the framework cannot support using entropy regularization asymmetrically (e.g. entropy regularization for actor, but not for the critic). On the other hand, Xiao (2022) propose a policy gradient framework and analyze the actor objective in `SAC`. However, they assume that $Q$ is estimated via a black-box procedure and do not instantiate the critic update.

**Contribution 2**: In Section 3, we develop a general off-policy actor-critic framework for the discrete-action setting, introducing new objectives, with variants of `DSAC` arising as special cases. In particular, in the policy evaluation step for the critic, we use a look-ahead target formed by either the soft *critic entropy*-regularized or the standard hard Bellman operator. The policy optimization step for the actor consists of two stages: (i) computing an intermediate policy (for example, by using either the `NPG` (Kakade, 2001) or `SPMA` (Asad et al., 2024) updates), and (ii) projecting (using either the forward or reverse KL divergence) the intermediate policy onto the class of realizable policies while simultaneously maximizing a proximal entropy regularization term (referred to as the *actor entropy*). Importantly, our framework decouples the use of critic and actor entropy is consequently more flexible.

**Contribution 3**: In Section 4, we analyze the proposed actor-critic framework in the simplified tabular setting. In particular, in Theorem 1, we reduce the problem of analyzing the sub-optimality in the entropy-regularized value function to (i) bounding the policy evaluation error for the critic and (ii) bounding the regret for an online convex optimization problem related to the policy update step for the actor. Our modular framework can be used to analyze different combinations of the actor and critic. For example, in Corollary 13, we use our framework to provide a theoretical guarantee for the `NPG` update with actor entropy and $m$ steps of the hard Bellman operator for policy evaluation, a variant that matches the performance of `DQN` in the function approximation setting. Furthermore, for certain special cases, our theoretical convergence rates match those in Vieillard et al. (2020a).

**Contribution 4**: In Section 5, we empirically evaluate the objectives instantiated using our actor-critic framework and compare them against `DQN`. Our results reveal three key findings. First, similar to `DSAC`, our objectives benefit from the hard Bellman operator for policy evaluation, resulting in improved performance. Second, unlike `DSAC`, the proposed objectives achieve performance competitive with `DQN` even without entropy regularization or explicit exploration. Third, the choice of forward vs. reverse KL divergence when projecting the intermediate policy onto the class of realizable policies does not matter in most cases.

## 2 Preliminaries

We consider an infinite-horizon discounted Markov decision process (MDP), defined as $\mathcal{M} = \langle \mathcal{S}, \mathcal{A}, \mathcal{P}, r, \rho, \gamma \rangle$, where $\mathcal{S}$ and $\mathcal{A}$ denote the state and action spaces, $\mathcal{P} : \mathcal{S} \times \mathcal{A} \to \Delta_{\mathcal{S}}$ is the transition probability function, $r : \mathcal{S} \times \mathcal{A} \to [0, 1]$ is the reward function, $\rho \in \Delta_{\mathcal{S}}$ is the initial state distribution, and $\gamma \in [0, 1)$ is the discount factor. Throughout this paper, we assume that the state and action spaces are finite but potentially large.

For a fixed $s \in \mathcal{S}$, the policy $\pi$ induces a distribution $\pi(\cdot|s)$ over the actions. The action-value function $q^\pi : \mathcal{S} \times \mathcal{A} \to \mathbb{R}$ of policy $\pi$ is defined as $q^\pi(s, a) := \mathbb{E}[\sum_{t=0}^\infty \gamma^t r(s_t, a_t)|s_0 =$

$s, a_0 = a]$, with $s_t \sim p(\cdot|s_{t-1}, a_{t-1})$ and $a_t \sim \pi(\cdot|s_t)$. Given an initial state $s \sim \rho$, the corresponding value function is defined as $v^\pi(s) := \mathbb{E}_{a \sim \pi(.|s)}[q^\pi(s, a)]$. The advantage function $\mathfrak{a}^\pi : \mathcal{S} \times \mathcal{A} \to \mathbb{R}$ induced by $\pi$ is represented by $\mathfrak{a}^\pi(s, a) := q^\pi(s, a) - v^\pi(s)$. We define $J(\pi) := v^\pi(\rho) = \mathbb{E}_{s \sim \rho}[v^\pi(s)]$ as the expected discounted cumulative reward.

Given the Shannon entropy function $\mathcal{H}(\pi) = -\sum_a \pi(a) \ln(\pi(a))$, if $\tau \geq 0$ is the entropy coefficient, then, the soft (entropy-regularized) counterparts of the above functions (Liu et al., 2024; Vieillard et al., 2020a) are defined as: $v_\tau^\pi(s) := v^\pi(s) + \tau \sum_{t=0}^\infty \gamma^t [\mathcal{H}(\pi(\cdot|s_t)) \mid s_0 = s]$, $J_\tau(\pi) := \mathbb{E}_{s \sim \rho}[v_\tau^\pi(s)]$, $q_\tau^\pi(s, a) := \mathbb{E}_{s' \sim \mathcal{P}(\cdot|s,a)}[r(s, a) + \gamma v_\tau^\pi(s')]$ and $\mathfrak{a}_\tau^\pi(s, a) := q_\tau^\pi(s, a) - v_\tau^\pi(s) - \tau \ln(\pi(a|s))$. Note that the soft value and action-value functions, $v_\tau^\pi(s)$ and $q_\tau^\pi(s, a)$, are bounded and lie within the interval $[0, H_\tau]$, where $H_\tau := \frac{1 + \tau \ln(A)}{1 - \gamma}$ (Liu et al., 2024).

Furthermore, for a fixed $s \in \mathcal{S}$ and an arbitrary pair of policies $\pi_1$ and $\pi_2$, the *soft* Bellman operator $T_\tau^\pi$ is defined such that: $(T_\tau^{\pi_1} v_\tau^{\pi_2})(s) = \mathbb{E}_{a \sim \pi_1(\cdot|s)}[q_\tau^{\pi_2}(s, a) - \tau \ln(\pi_1(a|s))] = (T^{\pi_1} v_\tau^{\pi_2})(s) + \tau \mathcal{H}(\pi_1(\cdot|s))$ and $(T_\tau^{\pi_1} q_\tau^{\pi_2})(s, a) = (T^{\pi_1} q_\tau^{\pi_2})(s, a) + \tau \mathbb{E}_{s' \sim \mathcal{P}(\cdot|s,a)} \mathcal{H}(\pi_1(\cdot|s'))$. If $\tau = 0$, we refer to the corresponding operator as the *hard* Bellman operator. The objective is to $\max_{\pi \in \Pi} J_\tau(\pi)$, where $\Pi$ is the set of feasible policies. We denote the optimal entropy-regularized policy by $\pi_\tau^* := \arg\max_\pi J_\tau(\pi)$ and its corresponding value function as $v_\tau^*$.

# 3 A GENERAL OFF-POLICY ACTOR-CRITIC FRAMEWORK

Building on the empirical findings from Section 1, we present a general off-policy actor-critic framework that subsumes DSAC and results in new actor objectives. In Section 3.1, we focus on the policy evaluation step and instantiate the resulting critic objective. In Section 3.2, we focus on two alternative policy updates and instantiate the corresponding actor objectives.

## 3.1 POLICY EVALUATION

At iteration $t \in [K]$ of the actor-critic algorithm, we evaluate the current policy $\pi_t$ using the $m$-step entropy-regularized Bellman operator. The coefficient, $\zeta \geq 0$, controls the strength of entropy regularization (referred to as the *critic entropy*). The corresponding estimate of the entropy-regularized $q$ function at iteration $t$ is denoted by $q_\zeta^t$ and computed recursively as:

$$q_\zeta^0 = q^{\pi_0} \quad ; \quad \forall t \geq 1, \quad q_\zeta^t = \mathbb{P}_{[0, H_\tau]}[(T_\zeta^{\pi_t})^m q_\zeta^{t-1}], \tag{1}$$

where $\mathbb{P}_{[0, H_\tau]}$ projects each entry onto the $[0, H_\tau]$ interval. As $m \to \infty$, the algorithm exactly evaluates the policy $\pi_t$ and $q_\zeta^t$ converges to the fixed point $q^{\pi_t}$. In this special case, setting $\zeta = 0$ recovers the standard $q$ function.

**Handling function approximation:** Eq. (1) requires updating the state-action value function for each state and action. In settings where the state or action space is large, this is not computationally feasible and we aim to approximate this update. To this end, we focus on the off-policy setting and define the critic objective using the standard squared loss (Haarnoja et al., 2018). Specifically, we use the *replay buffer* $\mathcal{D}_t$ consisting of (state, action, next state, reward) pairs obtained by the policies in the previous iterations. We define $\phi$ as the parameters of a model (typically a neural network) parameterizing the critic and $q_\phi$ as the corresponding function. For defining the critic objective, the details of the model are irrelevant and are implicit in the $q_\phi$ notation. We use Eq. (1) to construct a *one-step look-ahead target* (corresponding to $m = 1$[1]) and define the critic objective $\mathcal{L}_t(\phi)$ as:

$$\mathbb{E}_{\substack{(s,a,s',r(s,a)) \sim \mathcal{D}_t \\ a' \sim \pi_t(\cdot|s')}} \left\| q_\phi(s, a) - \mathbb{P}_{[0, H_\tau]}[r(s, a) + \gamma [q_\zeta^{t-1}(s', a') - \zeta \ln(\pi_t(a'|s'))]] \right\|_2^2. \tag{2}$$

With a slight abuse of notation, by $(s, a, s', r(s, a)) \sim \mathcal{D}_t$, we mean that the tuple is sampled from a discrete distribution over $\mathcal{D}_t$. We set $q_\zeta^t = q_{\phi_t}$ where $\phi_t \approx \arg\min \mathcal{L}_t(\phi)$. Similar to Haarnoja et al. (2018), in practice, we do not clip the look-ahead target and the optimization is done over the $(s, a, s', r(s, a))$ tuples in a randomly-sampled batch from $\mathcal{D}_t$.

---

[1]We focus on $m = 1$ for simplicity. The objective for $m > 1$ can be defined analogously.

### 3.2 Policy Optimization

At iteration $t \in [K]$ in the actor critic algorithm, the policy optimization step uses the $q$ function estimates from Section 3.1 to update the policy. For a state $s \in \mathcal{S}$, we first compute an intermediate policy $\pi_{t+1/2}$ using two representative methods — (i) the natural policy gradient (NPG) or policy mirror descent (PMD) (Kakade, 2001; Xiao, 2022) update and (ii) the recently proposed SPMA update (Asad et al., 2024). We note that the framework is not limited to these choices. With a suitable step-size $\eta_t$ and appropriate normalization,

$$\pi_{t+1/2}(a|s) \propto \pi_t(a|s) \exp(\eta_t \, q_\zeta^t(s,a)) \qquad (\textbf{NPG}) \tag{3}$$

$$\pi_{t+1/2}(a|s) \propto \pi_t(a|s) \left[1 + \eta_t \left(q_\zeta^t(s,a) - v_\zeta^t(s)\right)\right] \qquad (\textbf{SPMA}) \tag{4}$$

In the special case, when $\zeta = 0$, we note that the SPMA update can use a sufficiently small step-size to avoid an explicit normalization across the actions (Asad et al., 2024). On the other hand, the NPG update always requires an explicit normalization across actions to ensure that $\pi_{t+1/2}$ is a valid probability distribution. We now use a proximal update to incorporate entropy-regularization in the policy optimization step. Given $\pi_{t+1/2}$ and the entropy regularization parameter $\tau \geq 0$, the updated policy $\pi_{t+1}$ can be computed in two alternative ways that aim to find the "closest" policy $\pi$ to $\pi_{t+1/2}$ while encouraging the resulting policy to have sufficiently high entropy (referred to as the *actor entropy*[2]). Specifically, if $\tau_t := \tau \eta_t \geq 0$ is the entropy regularization parameter at iteration $t$, we use either the forward KL (FKL) or reverse KL (RKL) divergence to measure the proximity between policies. The resulting updates are given by:

$$\pi_{t+1}(\cdot|s) = \underset{\pi(\cdot|s) \in \Delta}{\arg\min} \, \mathrm{KL}(\pi_{t+1/2}(\cdot|s) \| \pi(\cdot|s)) - \tau_t \mathcal{H}(\pi(\cdot|s)) \qquad (\textbf{FKL}) \tag{5a}$$

$$\pi_{t+1}(\cdot|s) = \underset{\pi(\cdot|s) \in \Delta}{\arg\min} \, \mathrm{KL}(\pi(\cdot|s) \| \pi_{t+1/2}(\cdot|s)) - \tau_t \mathcal{H}(\pi(\cdot|s)) \qquad (\textbf{RKL}) \tag{5b}$$

Note that in the special case when $\tau = 0$, $\pi_{t+1} = \pi_{t+1/2}$. Furthermore, the objective in Eq. (5b) is convex in $\pi$ and the resulting update can be obtained in closed form where $\pi_{t+1}(a|s) \propto [\pi_{t+1/2}(a|s)]^{\frac{1}{1+\tau_t}}$ for all $(s,a)$. Combining Eqs. (5a) and (5b) with Eqs. (3) and (4) gives rise to four possible ways of updating the policy. We instantiate the corresponding actor objectives in the function approximation setting below.

**Handling function approximation:** Eqs. (5a) and (5b) require updating the state-action value function for each state and action. In order to scale to large state-action spaces, we use function approximation in the policy space. Specifically, given the parameters $\theta$ of a model parameterizing the actor and $\pi(\theta)$ as the corresponding policy, we define $\Pi_\theta = \{\pi \,|\, \exists \, \theta \text{ s.t } \pi = \pi(\theta)\}$ as the set of realizable policies. Similar to Section 3.1, the choice of the model is implicit in the $\pi(\theta)$ notation. Following Haarnoja et al. (2018), we modify Eqs. (5a) and (5b) to optimize (i) only over the states in the replay buffer $\mathcal{D}_t$ and (ii) over the restricted policy class $\Pi_\theta$ to get the following updates: $\pi_{t+1} = \arg\min_{\pi \in \Pi_\theta} \sum_{s \sim \mathcal{D}_t} \left[\mathrm{KL}(\pi_{t+1/2}(\cdot|s) \| \pi(\cdot|s)) - \right.$ $\left. \tau_t \mathcal{H}(\pi(\cdot|s))\right]$ and $\pi_{t+1} = \arg\min_{\pi \in \Pi_\theta} \sum_{s \sim \mathcal{D}_t} \left[\mathrm{KL}(\pi(\cdot|s) \| \pi_{t+1/2}(\cdot|s)) - \tau_t \mathcal{H}(\pi(\cdot|s))\right]$ for the FKL and RKL variants respectively. Following Vaswani et al. (2021); Lavington et al. (2023); Tomar et al. (2020); Xiong et al. (2024), we convert the above projection problem into an unconstrained optimization over $\theta$, and form $\ell_t(\theta)$, the corresponding actor objective. We define $\pi_{t+1} = \pi(\theta_{t+1})$ where $\theta_{t+1} \approx \arg\max_\theta \ell_t(\theta)$. For each variant, we append the postfix $(\tau, \zeta)$ to denote its dependence on the actor and critic entropy. The actor objective $\ell_t(\theta)$ can be defined as one of four possible choices:

$$\mathbb{E}_{s \sim \mathcal{D}_t} \left[\mathbb{E}_{a \sim \pi_\theta(.|s)} \left[q_\zeta^t(s,a) - \tau \ln(\pi_\theta(a|s))\right] - \frac{1}{\eta_t} \mathrm{KL}(\pi_\theta(\cdot|s) \,\|\, \pi_t(\cdot|s))\right] \qquad (\texttt{NPG-RKL}(\tau, \zeta))$$

---

[2]The actor entropy coefficient $\tau$ is independent from the critic entropy coefficient $\zeta$. Intuitively, actor entropy regularizes the distribution over actions at the current state, whereas the critic entropy regularizes the distribution in the next-state.

$$\mathbb{E}_{s \sim \mathcal{D}_t} \left[ \mathbb{E}_{a \sim \pi_\theta(\cdot|s)} [\ln \left(1 + \eta_t (q_\zeta^t(s,a) - v_\zeta^t(s))\right) - \tau_t \ln \pi_\theta(a|s)] - \mathrm{KL}(\pi_\theta(\cdot|s)||\pi_t(\cdot|s)) \right]$$
$$(\texttt{SPMA-RKL}(\tau, \zeta))$$

$$\mathbb{E}_{s \sim \mathcal{D}_t} \left[ \mathbb{E}_{a \sim \pi_t(\cdot|s)} \left[ \frac{\exp(\eta_t \, q_\zeta^t(s,a))}{\sum_{a'} \pi_t(a') \exp(\eta_t \, q_\zeta^t(s,a'))} \ln \left( \frac{\pi_\theta(a|s)}{\pi_t(a|s)} \right) \right] + \tau_t \mathcal{H}(\pi_\theta(.|s)) \right]$$
$$(\texttt{NPG-FKL}(\tau, \zeta))$$

$$\mathbb{E}_{s \sim \mathcal{D}_t} \left[ \mathbb{E}_{a \sim \pi_t(\cdot|s)} \left[ \frac{(1 + \eta_t (q_\zeta^t(s,a) - v_\zeta^t(s)))}{\sum_{a'} \pi_t(a') \, (1 + \eta_t (q_\zeta^t(s,a') - v_\zeta^t(s)))} \ln \left( \frac{\pi_\theta(a|s)}{\pi_t(a|s)} \right) \right] + \tau_t \mathcal{H}(\pi_\theta(.|s)) \right]$$
$$(\texttt{SPMA-FKL}(\tau, \zeta))$$

**Comparing the objectives**: Note that the `NPG-RKL` and `SPMA-RKL` objectives differ in the first term, which is linear in the $q$ function for `NPG-RKL`, while it is logarithmic in the advantage for `SPMA-RKL`. Similarly, the FKL variants also differ in the first term. Crucially, in the special case of zero critic entropy and $\eta \leq 1 - \gamma$, `SPMA-FKL`$(\tau, 0)$ does not require an explicit normalization over the actions (Asad et al., 2024), making it easier to implement in practice. Finally, we note that for the RKL variants, the expectation is over the actions sampled from $\pi_\theta$. The objective can be optimized by calculating the full expectation, using importance sampling, or applying the reparameterization trick. On the other hand, the FKL variants involve an expectation over the actions sampled from $\pi_t$, simplifying the resulting implementation. In Algorithm 1 in Appendix B, we present the complete actor-critic pseudo-code in the function approximation setting.

**Comparison to existing methods:** We note that `NPG-RKL`$(\tau, \tau)$ recovers the off-policy variant of `MDPO` studied in Tomar et al. (2020). In the limit that $\eta_t \to \infty$, `NPG-RKL`$(\tau, \tau)$ recovers the original `SAC` objective in Haarnoja et al. (2018). This is intuitive since `SAC` can be viewed as soft policy iteration. Importantly, `NPG-RKL`$(\tau, 0)$ and $\eta_t \to \infty$ recovers the `DSAC` variant that demonstrated good empirical performance in Fig. 1. In the special case of the tabular setting and $\zeta = \tau$ (i.e. the actor and critic entropy is coupled and fixed), the `NPG-RKL` variant is the same as that proposed in Vieillard et al. (2020a). On the other hand, the `SPMA-RKL`$(\tau, \zeta)$ objective is novel, and has not been studied in the previous literature. In the tabular setting, `NPG-FKL`$(\tau, \tau)$ is the same as the objective proposed by Mei et al. (2019), but the two objectives differ under function approximation.

Finally we note that in the on-policy setting where states are sampled from $d^{\pi_t}$, the distribution induced by policy $\pi_t$ (instead of $\mathcal{D}_t$), setting $\tau = \zeta = 0$ and $m = \infty$ for the critic (corresponding to exact policy evaluation) we can recover the framework in Vaswani et al. (2021), and its instantiations (Tomar et al., 2020; Asad et al., 2024). Next, we consider the simplified tabular setting, and prove theoretical guarantees for the RKL variants.

## 4 THEORETICAL GUARANTEE IN THE TABULAR SETTING

We consider the tabular setting and analyze the actor-critic algorithm when using decoupled non-zero critic and actor entropy. For the critic, we consider estimating the $q$ functions using Eq. (1). For the actor, we consider using the RKL variant in Eq. (5b) in conjunction with the `NPG` and `SPMA` updates in Eqs. (3) and (4), and denote the corresponding variants as *soft NPG* and *soft SPMA* respectively.

In Theorem 1 below, we first reduce bounding the sub-optimality in the entropy-regularized value function to a per-state online convex optimization problem. It is important to note that this reduction is independent of the specific actor and critic updates.

**Theorem 1** (Generic Reduction with Actor Entropy)**.** *If $\pi_\tau^*$ is the optimal entropy-regularized policy whose value function is $v_\tau^*$, for a $q_\zeta^t$ obtained via the policy evaluation scheme at iteration $t$, define $\epsilon_t := q_\zeta^t - q_\tau^{\pi_t}$. For any sequence of policies $\{\pi_0, \pi_1, \ldots, \pi_{K-1}\}$, if $\bar{\pi}_K$ is the corresponding uniform mixture policy, then,*

$$\left\| v_\tau^* - v_\tau^{\bar{\pi}_K} \right\|_\infty \leq \frac{\|Regret(K)\|_\infty}{K(1-\gamma)} + \frac{2 \sum_{t \in [K]} \|\epsilon_t\|_\infty}{K(1-\gamma)},$$

where $(Regret(K))(s) := \sum_{t=0}^{K-1} \left[ \langle \pi_\tau^*(\cdot|s) - \pi_t(\cdot|s), q_\zeta^t(s, \cdot) \rangle + \tau \left[ \mathcal{H}(\pi_\tau^*(\cdot|s)) - \mathcal{H}(\pi_t(\cdot|s)) \right] \right]$ is the regret incurred on an online optimization problem for each state $s \in \mathcal{S}$.

The above result shows that the sub-optimality of a mixture policy (one that randomly chooses a policy from $\{\pi_0, \pi_1, \ldots, \pi_{K-1}\}$) obtained by using a generic policy optimization method can be bounded in terms of the regret incurred by the method, and the sum of the *policy evaluation errors* incurred in estimating $q_\tau^{\pi_t}$. Since $\epsilon_t$ depends on $\pi_t$, it depends on the specific policy optimization method, and we now bound it for both soft NPG and soft SPMA.

**Corollary 1** (Policy Evaluation Error). *Using the policy evaluation update in Eq.* (1) *and the soft NPG or soft SPMA policy update with $\eta_t = \frac{1}{c+\tau(t+1)}$ for a constant $c \geq 0$, if $\delta(\tau, \zeta) := \frac{|\tau - \zeta| \ln(A)}{1-\gamma}$, the error $\epsilon_t$ can be bounded for all $t \in [K]$ as:*

$$\epsilon_t := \|\boldsymbol{\epsilon}_t\|_\infty = O\left( \frac{\gamma^m}{(1-\gamma)^4} \left[ \left( \ln(K)^2 \left( \frac{1}{t} + (\gamma^m)^{\frac{t}{2}} \right) + \frac{1}{K} \right) + \frac{1}{K^2} \right] + \frac{\delta(\tau, \zeta)}{1-\gamma} \right).$$

Note that as $m$ (the number of steps of the Bellman operator) increases, the policy is evaluated more accurately, and $\epsilon_t$ decreases. As $m \to \infty$, $\epsilon_t \to O(\delta(\tau, \zeta))$. Moreover, as $t$ increases, $\epsilon_t$ decreases and as $t \to K \to \infty$, $\epsilon_t \to O(\delta(\tau, \zeta))$, which quantifies the mismatch between the actor and critic entropy and is equal to zero when $\zeta = \tau$.

Next, we show that both soft NPG and soft SPMA can control the regret for the online optimization problem defined in Theorem 1.

**Corollary 2** (Regret Bounds with Actor Entropy). *Suppose $\pi_0(\cdot|s)$ is the uniform distribution over actions for state $s$. For a sequence $\{q_\zeta^t\}_{t=0}^{K-1}$ with $\left\| q_\zeta^t \right\|_\infty \leq H_\tau$, the regret for soft NPG and soft SPMA with $\eta_t = \frac{1}{c+\tau(t+1)}$ for constant $c \geq 0$ can be bounded as:*

$$\max_s \left| \sum_{t=0}^{K-1} \left[ \langle \pi_\tau^*(\cdot|s) - \pi_t(\cdot|s), q_\zeta^t(s, \cdot) \rangle + \tau \left[ \mathcal{H}(\pi_\tau^*(\cdot|s)) - \mathcal{H}(\pi_t(\cdot|s)) \right] \right] \right| = O\left( \frac{\ln(K)}{(1-\gamma)^2} \right)$$

The actor entropy in Eq. (5b) makes the online functions strongly-convex and hence both methods incur only a logarithmic regret (Orabona, 2019). Combining the results in Corollaries 1 and 2 with Theorem 1, we obtain our main theorem for both soft NPG and soft SPMA.

**Theorem 2** (Sub-optimality of Soft NPG/Soft SPMA). *Let $\pi_\tau^*$ be the optimal entropy-regularized policy with value function $v_\tau^*$. Consider the soft NPG or soft SPMA updates with step size $\eta_t = \frac{1}{c+\tau(t+1)}$ for constant $c$ defined in Theorems 5 and 7 and $\pi_0(\cdot|s)$ as the uniform policy over actions for all $s \in \mathcal{S}$. Using policy evaluation step in Eq.* (1), *with $\delta(\tau, \zeta) := \frac{|\tau - \zeta| \ln(A)}{1-\gamma}$, the resulting uniform mixture policy $\bar{\pi}_K$ satisfies the following sub-optimality bound:*

$$\left\| v_\tau^{\bar{\pi}_K} - v_\tau^* \right\|_\infty = O\left( \frac{\ln(K)}{K(1-\gamma)^3} + \frac{\gamma^m}{K(1-\gamma)^5} \left[ \ln(K)^2 \left( \ln(K) + \frac{1}{1-\sqrt{\gamma}} \right) + \frac{1}{K} \right] + \frac{\delta(\tau, \zeta)}{(1-\gamma)^2} \right)$$

Hence, the sub-optimality can be bounded as $\tilde{O}\left( \frac{1}{K} + \frac{\gamma^m}{K} + \delta(\tau, \zeta) \right)$ up to logarithmic factors. Note that when the actor and critic entropy is coupled ($\zeta = \tau$), both soft NPG and SPMA have an $O(1/K)$ convergence for the resulting mixture policy. For the practical variant that uses $m = 1$ and $\zeta = 0$, the above result can achieve an $O(1/K)$ convergence to an $O(\tau)$ neighbourhood of $v_\tau^*$. All proofs are deferred to Appendix C, where we also explore the case without actor entropy i.e. $\tau = 0$ and prove an $O(1/\sqrt{K})$ rate.

**Comparison to existing results:** Vieillard et al. (2020a) consider the special case of soft NPG with coupled actor and critic entropy ($\zeta = \tau$) and $m = 1$, and also establish a convergence rate of $O(1/K)$. On the other hand, Xiao (2022, Theorem 4.4) analyze soft NPG with coupled actor and critic entropy ($\zeta = \tau$) and $m = \infty$, and prove an $O(1/K)$ convergence rate. In comparison, our framework can match these convergence rates while being more flexible — we can support $m \in [1, \infty)$ (left as an open question in Vieillard et al. (2020a)), allow for decoupling of actor and critic entropy and support policy optimization methods

beyond `NPG` (e.g., `SPMA`) that have sublinear regret and Lipschitz policy updates (see the proof of Theorem 3 in Appendix C). In the next section, we demonstrate that two methods (soft `NPG` and soft `SPMA` with decoupled actor and critic entropy, $m = 1$) instantiated by our framework have good empirical performance.

## 5 Empirical Evaluation

We evaluate the proposed off-policy policy optimization methods on various Atari 2600 games (Bellemare et al., 2013). We first present an ablation study of `DSAC`, and subsequently investigate the impact of actor and critic entropy and the direction of the KL divergence on the empirical performance of `DSAC` and the proposed objectives. Following Tomar et al. (2020); Asad et al. (2024), we use the `stable-baselines3` framework (Raffin et al., 2021) with five random seeds, reporting average expected returns with 95% confidence intervals. Experimental details and hyper-parameters for all algorithms are provided in Appendix E.1.

**Conclusion 1: Using $\zeta = 0$ Improves DSAC Performance:** In Fig. 2, we ablate `DSAC` (with a single $Q$-network), comparing $\zeta = 0$ versus $\zeta = \tau$ in the presence of actor entropy (i.e., $\tau \neq 0$). To set $\tau$, we use either a fixed grid-searched entropy coefficient or use the adaptive scheme in Christodoulou (2019). Our results indicate that both $\zeta = 0$ and $\zeta = \tau$ can result in good performance with a *well-tuned fixed per environment* $\tau$ (red and purple vs. blue). Under this configuration, we also observe that on the discrete versions of MuJoCo's Acrobot, MountainCar, and Pendulum reported in Neumann et al. (2018), `DSAC` performs on par with `GreedyAC`, which employs an alternative policy update (see Fig. 3). These observation are consistent with Vieillard et al. (2020b), which tunes a fixed $\tau$ for the entropy-regularized value-based `Soft-DQN` and `M-DQN` methods[3].
On the other hand, the adaptive scheme in Christodoulou (2019) sets $\tau^t \approx \arg\min \mathcal{E}_t(\alpha) = \mathbb{E}_{s \sim \mathcal{D}_t} \mathbb{E}_{a \sim \pi_t(.|s)}[-\alpha \ln(\pi_t(a|s)) - \alpha\bar{\mathcal{H}}]$ where $\bar{\mathcal{H}}$ is the fixed *target entropy*. When using this adaptive scheme, $\zeta = 0$ yields performance comparable to `DQN` (green vs. blue), same as the 20-game results in Fig. 1, while setting $\zeta = \tau$ results in much worse performance (orange line). Hence, in order to retain good empirical performance while avoiding the need for manual tuning, we recommend using the adaptive scheme for $\tau$ and setting $\zeta = 0$.
For this configuration, we test the hypothesis in Zhou et al. (2022) that double $Q$-learning (Fujimoto et al., 2018) adversely affects `DSAC` performance. In Fig. 6 in Appendix E.2, we compare single and double $Q$-learning across 8 Atari games and observe that, under our setup, double $Q$-learning yields similarly strong results and does not degrade performance. We also evaluate the hypothesis in Xu et al. (2021) that when using the adaptive scheme to set $\tau$, choosing a fixed target entropy $\bar{\mathcal{H}}$ harms the `DSAC` performance. The results in Fig. 6 show that our proposed `DSAC` configuration (using the same adaptive scheme as in Christodoulou (2019)) significantly outperforms those reported in Xu et al. (2021, Figure 2).

**Conclusion 2: All Proposed Objectives with $\zeta = 0$ Have Similar Good Performance:** From Fig. 4, we observe that when using adaptive $\tau$, all proposed actor objectives defined in Section 3 have relatively (compared to `DQN`) bad performance when $\zeta = \tau$, and good performance with $\zeta = 0$. Moreover, similar to `DSAC`$(\tau, 0)$, these methods have good performance even with 1 gradient descent step (on a randomly sampled batch from the replay buffer) for both the actor and critic objectives. Consequently, for the subsequent results, we set $\zeta = 0$ with adaptive $\tau$. Notably, the results in the second row, columns one and three of Fig. 4 correspond to our novel objectives `SPMA-FKL`$(\tau, 0)$ and `SPMA-RKL`$(\tau, 0)$, which are motivated by our empirical findings on `DSAC`. In addition, the results shown in the second row and second column correspond to the `NPG-FKL`$(\tau, 0)$ objective, which is different from the formulation of Mei et al. (2019) in the function approximation setting. We also compare with the common on-policy baseline `PPO` (Schulman et al., 2017) in Appendix E.6 (Fig. 19, Fig. 20), and observe that the proposed methods significantly outperform it.

---

[3]In contrast to `DSAC`, both `Soft-DQN` and `M-DQN` also require tuning an $\epsilon$-greedy parameter and an additional parameter related to entropy regularization (Vieillard et al., 2020b).

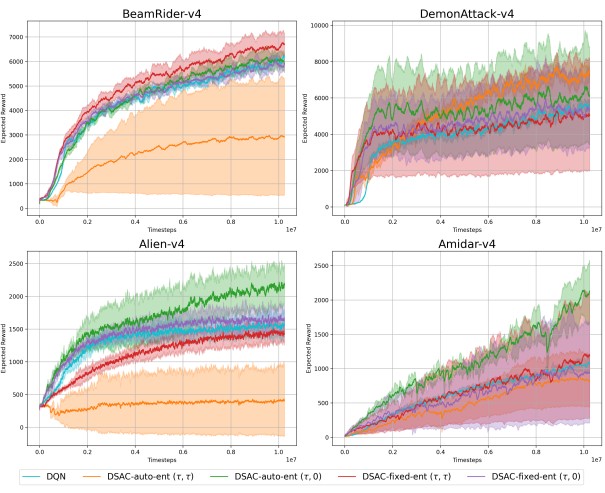

**Figure 2:** `DSAC` ablation study. Training `DSAC` with the adaptive entropy coefficient loss from the original paper with $\zeta = 0$ consistently achieves strong performance compared to `DQN` (green vs. blue curves). In contrast, using $\zeta = \tau$ without a well-tuned fixed $\tau$ can lead to poor performance (orange curve).

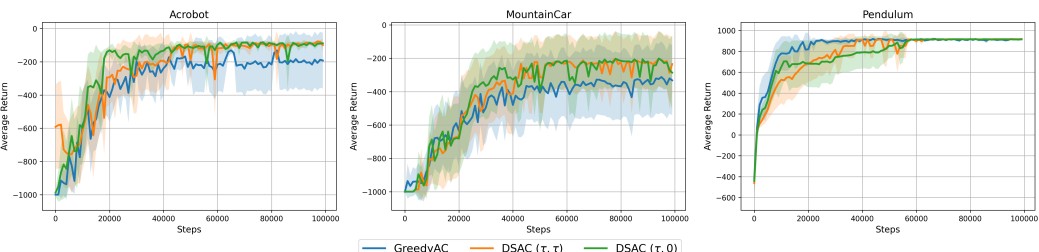

**Figure 3:** Comparing `GreedyAC` (Neumann et al., 2018) with the default `DSAC` (with and without critic entropy and a fixed entropy coefficient) using batch size of 32.

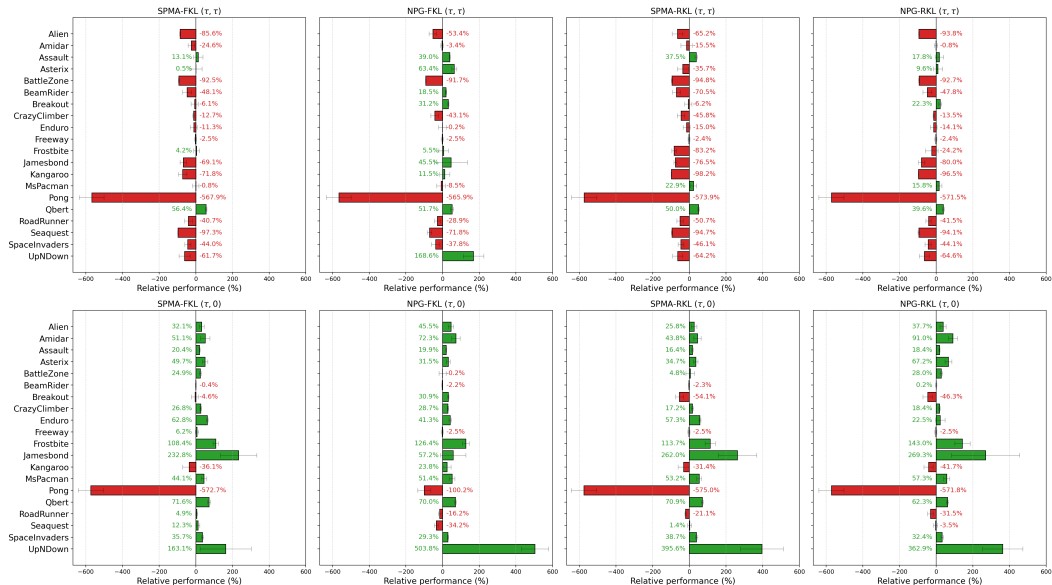

**Figure 4:** Disabling critic entropy while retaining the adaptive entropy coefficient loss, as in `DSAC`, yields substantial performance gains for both our forward and reverse KL actors (top vs. bottom rows).

**Conclusion 3: Entropy Regularization is Only Sometimes Important:** We study the effect of disabling entropy regularization for both actor and critic (i.e., $\tau = 0$ and

$\zeta = 0$) when the actor objective $\ell_t(\theta)$ is optimized by $n$ gradient steps. For $n = 1$, all objectives, including $\mathtt{DSAC}(0,0)$, suffer from a sharp drop in policy entropy and degraded performance (see Figs. 8 and 10), indicating insufficient exploration in the absence of entropy terms. However, increasing $n$ alleviates this effect for our objectives: on most games, setting $n = 10$ preserves higher entropy and yields more stable learning, resulting in good overall performance. Fig. 5 demonstrates that our methods can match $\mathtt{DQN}$ for large $n$, while $\mathtt{DSAC}(0,0)$ consistently underperforms.

This gap can be explained by the lack of regularization in $\mathtt{DSAC}$. In particular, recall that $\mathtt{DSAC}$ is a limiting case of $\mathtt{NPG\text{-}RKL}$ with $\eta_t \to \infty$, and hence does not have the reverse KL regularization term (see the $\mathtt{NPG\text{-}RKL}$ objective in Section 3). Consequently, for $n > 1$, $\mathtt{NPG\text{-}RKL}(0,0)$ benefits from additional optimization steps under KL regularization, whereas $\mathtt{DSAC}(0,0)$ lacks both the KL and entropy terms. Overall, Fig. 10 in Appendix E.4 shows that several of our objectives remain competitive with $\mathtt{DQN}$ without explicit entropy regularization, provided that the actor objective is sufficiently optimized.

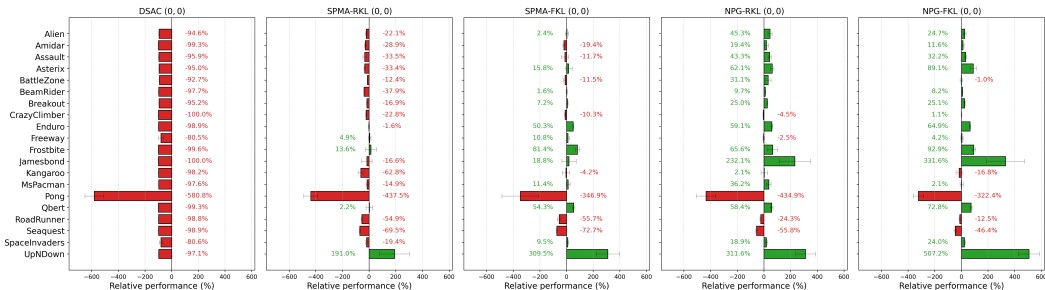

**Figure 5:** Unlike $\mathtt{DSAC}(0,0)$, our objectives can match $\mathtt{DQN}$ without entropy regularization, when increasing the number of actor gradient updates $n$ to 10.

**Conclusion 4: Direction of KL Mostly Does Not Matter:** Chan et al. (2022) have studied the impact of the KL direction in the discrete-action setting when the intermediate policy is Boltzmann i.e. $\pi_{t+1/2} \propto \exp(q_\tau^t(s,a)/\tau)$, and have reported no significant differences in the performance. We revisit this question in the context of our actor objectives, evaluating $\mathtt{SPMA}$ and $\mathtt{NPG}$ under two regimes: (i) $\zeta = 0, \tau = 0$ and (ii) $\tau \neq 0$ with $\zeta = 0$. In the $\zeta = 0, \tau = 0$ case and with $n = 10$ (since $n = 1$ performs poorly), using forward KL results in a clear advantage for $\mathtt{SPMA}$, while $\mathtt{NPG}$ shows no consistent trend ( Figs. 13 and 14). In the case $\tau \neq 0$ and $\zeta = 0$, we find no consistent evidence that either KL direction systematically outperforms the other, regardless of $n$ or the choice of intermediate policy (see Figs. 15 to 18 in Appendix E.5).

## 6 CONCLUSION AND FUTURE WORK

We revisited the design of off-policy actor-critic methods in the discrete-action setting, where value-based methods such as $\mathtt{DQN}$ dominate. By decoupling the actor and critic entropy, we identified a variant of $\mathtt{DSAC}$ that matches $\mathtt{DQN}$. Building on this insight, we introduced a flexible off-policy actor-critic framework that subsumes $\mathtt{DSAC}$ variants as a special case. This framework resulted in novel actor objectives that can match or exceed $\mathtt{DQN}$ without explicit exploration. For example, we introduced the $\mathtt{SPMA\text{-}FKL}(\tau,0)$ objective which samples from $\pi_t$ and avoids an explicit action normalization over the actions, making it easier to implement in practice. Moreover, $\mathtt{SPMA\text{-}FKL}(\tau,0)$ admits a convex objective in $\theta$ when using log-linear policies, making it a prospective candidate for theoretical analysis when incorporating function approximation. For future work, we aim to: (i) theoretically analyze the actor objectives with function approximation and guide algorithmic design; (ii) develop strategies for automatically adapting the actor step size $\eta_t$; (iii) further investigate the behavior of the proposed objectives without entropy regularization; and (iv) extend the framework to continuous-action settings.

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

# Supplementary Material

## A  ACTOR OBJECTIVE INSTANTIATIONS

In this section, we instantiate our forward and reverse KL-based objectives in Equations 5a and 5b using soft NPG, and soft SPMA in the function approximation setting.

### A.1  NPG-FKL$(\tau, \zeta)$

$$\pi_{t+1} = \arg\min_{\pi_\theta \in \Pi} \mathbb{E}_{s \sim \mathcal{D}_t} \left[ \mathrm{KL}(\pi_{t+1/2}(\cdot|s)||\pi_\theta(\cdot|s)) - \tau_t \mathcal{H}(\pi_\theta(\cdot|s)) \right]$$

$$= \arg\min_{\pi_\theta \in \Pi} \mathbb{E}_{s \sim \mathcal{D}_t} \left[ \mathbb{E}_{a \sim \pi_t(\cdot|s)} \left[ -\frac{\exp(\eta_t\, q_\zeta^t(s,a))}{\mathcal{Z}_t(s)} \ln \left( \frac{\pi_\theta(a|s)}{\frac{\pi_t(a|s)\,\exp(\eta_t q_\zeta^t(s,a))}{\mathcal{Z}_t(s)}} \right) \right] - \tau_t \mathcal{H}(\pi_\theta(\cdot|s)) \right]$$

$$\text{(Using the NPG update in Eq. (3))}$$

$$= \arg\min_{\pi_\theta \in \Pi} \mathbb{E}_{s \sim \mathcal{D}_t} \left[ \mathbb{E}_{a \sim \pi_t(\cdot|s)} \left[ -\frac{\exp(\eta_t\, q_\zeta^t(s,a))}{\mathcal{Z}_t(s)} \ln \left( \frac{\pi(a|s)}{\pi_t(a|s)} \right) \right] - \tau_t \mathcal{H}(\pi_\theta(\cdot|s)) \right]$$

$$\text{(dropping the constant w.r.t } \pi)$$

$$= \arg\max_{\pi_\theta \in \Pi} \mathbb{E}_{s \sim \mathcal{D}_t} \left[ \mathbb{E}_{a \sim \pi_t(\cdot|s)} \left[ \frac{\exp(\eta_t\, q_\zeta^t(s,a))}{\sum_{a'} \pi_t(a')\,\exp(\eta_t\, q_\zeta^t(s,a))} \ln \left( \frac{\pi_\theta(a|s)}{\pi_t(a|s)} \right) \right] + \tau_t \mathcal{H}(\pi_\theta(\cdot|s)) \right]$$

### A.2  SPMA-FKL$(\tau, \zeta)$

$$\pi_{t+1} = \arg\min_{\pi_\theta \in \Pi} \mathbb{E}_{s \sim \mathcal{D}_t} \left[ \mathrm{KL}(\pi_{t+1/2}(\cdot|s)||\pi_\theta(\cdot|s)) - \tau_t \mathcal{H}(\pi_\theta(\cdot|s)) \right]$$

$$= \arg\min_{\pi_\theta \in \Pi} \mathbb{E}_{s \sim \mathcal{D}_t} \left[ \mathbb{E}_{a \sim \pi_t(\cdot|s)} \left[ -\frac{\left(1 + \eta_t(q_\zeta^t(s,a) - v_\zeta(s)^t)\right)}{\mathcal{Z}_t(s)} \ln \left( \frac{\pi_\theta(a|s)}{\frac{\pi_t(a|s)\,(1 + \eta_t\,(q_\zeta^t(s,a) - v_\zeta^t(s)))}{\mathcal{Z}_t(s)}} \right) \right] \right.$$

$$- \tau_t \mathcal{H}(\pi_\theta(\cdot|s))\Big]$$

$$\text{(Using the SPMA update in Eq. (4))}$$

$$= \underset{\pi_\theta \in \Pi}{\arg\min} \, \mathbb{E}_{s \sim \mathcal{D}_t} \left[ \mathbb{E}_{a \sim \pi_t(\cdot|s)} \left[ - \frac{\left( 1 + \eta_t(q_\zeta^t(s,a) - v_\zeta^t(s)) \right)}{\mathcal{Z}_t(s)} \ln\left( \frac{\pi_\theta(a|s)}{\pi_t(a|s)} \right) \right] \right.$$

$$\left. - \tau_t \mathcal{H}(\pi_\theta(\cdot|s)) \right] \qquad \text{(dropping the constant w.r.t } \pi)$$

$$= \underset{\pi_\theta \in \Pi}{\arg\max} \, \mathbb{E}_{s \sim \mathcal{D}_t} \left[ \mathbb{E}_{a \sim \pi_t(\cdot|s)} \left[ \frac{\left( 1 + \eta_t(q_\zeta^t(s,a) - v_\zeta^t(s)) \right)}{\sum_{a'} \pi_t(a'|s) \left( 1 + \eta_t(q_\zeta^t(s,a') - v_\zeta^t(s)) \right)} \ln\left( \frac{\pi(a|s)}{\pi_t(a|s)} \right) \right] \right.$$

$$\left. + \tau_t \mathcal{H}(\pi_\theta(\cdot|s)) \right]$$

### A.3 NPG-RKL$(\tau, \zeta)$

$$\pi_{t+1} = \underset{\pi_\theta \in \Pi}{\arg\min} \, \mathbb{E}_{s \sim \mathcal{D}_t} \left[ \mathrm{KL}(\pi_\theta(\cdot|s)||\pi_{t+1/2}(\cdot|s)) - \tau_t \mathcal{H}(\pi_\theta(\cdot|s)) \right]$$

$$= \underset{\pi_\theta \in \Pi}{\arg\min} \, \mathbb{E}_{s \sim \mathcal{D}_t} \mathbb{E}_{a \sim \pi_\theta(\cdot|s)} \left[ (1 + \tau_t) \ln(\pi_\theta(a|s)) - \ln(\pi_{t+1/2}(a|s)) \right]$$

$$= \underset{\pi_\theta \in \Pi}{\arg\min} \, \mathbb{E}_{s \sim \mathcal{D}_t} \mathbb{E}_{a \sim \pi_\theta(\cdot|s)} \left[ (1 + \tau_t) \ln(\pi_\theta(a|s)) - \ln\left( \pi_t(a|s) \exp(\eta_t \, q_\zeta^t(s,a)) \right) \right]$$

$$\text{(Using the NPG update in Eq. (3) and since } \mathcal{Z}_t \text{ can be marginalized out)}$$

$$= \underset{\pi_\theta \in \Pi}{\arg\min} \, \mathbb{E}_{s \sim \mathcal{D}_t} \mathbb{E}_{a \sim \pi_\theta(\cdot|s)} \left[ (1 + \tau_t) \ln(\pi_\theta(a|s)) - \ln(\pi_t(a|s)) - \eta_t \, q_\zeta^t(s,a)) \right]$$

$$= \underset{\pi_\theta \in \Pi}{\arg\min} \, \mathbb{E}_{s \sim \mathcal{D}_t} \mathbb{E}_{a \sim \pi_\theta(\cdot|s)} \left[ (1 + \tau \eta_t) \ln(\pi_\theta(a|s)) - \eta_t \left( q_\zeta^t(s,a) + \frac{1}{\eta_t} \ln(\pi_t(a|s)) \right) \right]$$

$$\text{(Since } \tau_t = \eta_t \, \tau )$$

$$= \underset{\pi_\theta \in \Pi}{\arg\max} \, \mathbb{E}_{s \sim \mathcal{D}_t} \mathbb{E}_{a \sim \pi_\theta(\cdot|s)} \left[ q_\zeta^t(s,a) - \tau \ln(\pi_\theta(a|s)) - \frac{1}{\eta_t} \ln\left( \frac{\pi_\theta(a|s)}{\pi_t(a|s)} \right) \right]$$

$$= \underset{\pi_\theta \in \Pi}{\arg\max} \, \mathbb{E}_{s \sim \mathcal{D}_t} \left[ \mathbb{E}_{a \sim \pi_\theta(\cdot|s)} \left[ q_\zeta^t(s,a) - \tau \ln(\pi_\theta(a|s)) \right] - \frac{1}{\eta_t} \mathrm{KL}(\pi_\theta(\cdot|s) \, || \, \pi_t(\cdot|s)) \right]$$

Given an estimate of the entropy-regularized $q$ function, DSAC is a special of NPG-RKL by setting $\eta_t = \infty$ resulting in the following surrogate loss:

$$\pi_{t+1} = \underset{\pi_\theta \in \Pi}{\arg\max} \, \mathbb{E}_{s \sim \mathcal{D}_t} \mathbb{E}_{a \sim \pi_\theta(\cdot|s)} \left[ \left( q_\zeta^t(s,a) \right) - \tau \ln(\pi_\theta(a|s)) \right]$$

### A.4 SPMA-RKL$(\tau, \zeta)$

$$\pi_{t+1} = \underset{\pi_\theta \in \Pi}{\arg\min} \, \mathbb{E}_{s \sim \mathcal{D}_t} \left[ \mathrm{KL}(\pi_\theta(\cdot|s)||\pi_{t+1/2}(\cdot|s)) - \tau_t \mathcal{H}(\pi_\theta(\cdot|s)) \right]$$

$$= \underset{\pi_\theta \in \Pi}{\arg\min} \, \mathbb{E}_{s \sim \mathcal{D}_t} \mathbb{E}_{a \sim \pi_\theta(\cdot|s)} \left[ (1 + \tau_t) \ln(\pi(a|s)) - \ln(\pi_{t+1/2}(a|s)) \right]$$

$$= \arg\min_{\pi_\theta \in \Pi} \mathbb{E}_{s \sim \mathcal{D}_t} \mathbb{E}_{a \sim \pi_\theta(\cdot|s)} \left[ (1 + \tau_t) \ln(\pi_\theta(a|s)) - \ln\left( \pi_t(a|s) [1 + \eta_t \left( q_\zeta^t(s,a') - v_\zeta^t(s) \right)] \right) \right]$$

$$\text{(Using the SPMA update in Eq. (4))}$$

$$= \underset{\pi_\theta \in \Pi}{\arg\max} \, \mathbb{E}_{s \sim \mathcal{D}_t} \left[ \mathbb{E}_{a \sim \pi_\theta(\cdot|s)} \ln \left( 1 + \eta_t(q_\zeta^t(s,a) - v_\zeta^t(s)) \right) - \mathrm{KL}(\pi_\theta||\pi_t) - \tau_t \ln \pi_\theta(a|s) \right]$$

## B    General Off-Policy Actor-Critic Pseudocode

---

**Algorithm 1:** General Off-Policy Actor-Critic Framework

---

1: **Input:** $\theta_0$ ($\pi_0$'s parameters), $\phi_0$ ($q_\zeta^0$'s parameters), $\pi_\theta$ (function approximation for actor), $q_\phi$ (function approximation for critic), $\mathcal{L}_t$ (critic loss), $\ell_t$ (actor loss), $K$ (total iterations), $N$ (number of environment steps), $n$ (number of policy optimization steps), $\alpha$ (inner-loop step-size)

2: **for** $t = 0$ **to** $K - 1$ **do**

3:    Interact with the environment for $N$ steps to collect data using $\pi_t$:
       $D_t \leftarrow D_t \cup \{s_i, a_i, r(s_i, a_i), s_{i+1}\}_{i=1}^N$

4:    $\phi_t = \arg\min \mathcal{L}_t(\phi)$ ; $q_\zeta^t = q_{\phi_t}$

5:    Initialize inner-loop: $\omega_0 = \theta_t$

6:    **for** $j = 0$ **to** $n - 1$ **do**

7:       $\omega_{j+1} = \omega_j - \alpha \nabla_\omega \ell_t(\omega_j)$

8:    **end for**

9:    $\theta_{t+1} = \omega_n$

10:   $\pi_{t+1}(\cdot|s) = \pi_{\theta_{t+1}}(\cdot|s)$

11: **end for**

12: **Return:** $\theta_K$

---

## C    Theoretical Results

### C.1    Proof of Theorem 1

**Theorem 1** (Generic Reduction with Actor Entropy). *If $\pi_\tau^*$ is the optimal entropy-regularized policy whose value function is $v_\tau^*$, for a $q_\zeta^t$ obtained via the policy evaluation scheme at iteration $t$, define $\epsilon_t := q_\zeta^t - q_\tau^{\pi_t}$. For any sequence of policies $\{\pi_0, \pi_1, \ldots, \pi_{K-1}\}$, if $\bar{\pi}_K$ is the corresponding uniform mixture policy, then,*

$$\left\| v_\tau^* - v_\tau^{\bar{\pi}_K} \right\|_\infty \leq \frac{\|Regret(K)\|_\infty}{K(1-\gamma)} + \frac{2 \sum_{t \in [K]} \|\epsilon_t\|_\infty}{K(1-\gamma)},$$

*where $(Regret(K))(s) := \sum_{t=0}^{K-1} \left[ \langle \pi_\tau^*(\cdot|s) - \pi_t(\cdot|s), q_\zeta^t(s, \cdot) \rangle + \tau \left[ \mathcal{H}(\pi_\tau^*(\cdot|s)) - \mathcal{H}(\pi_t(\cdot|s)) \right] \right]$ is the regret incurred on an online optimization problem for each state $s \in \mathcal{S}$.*

*Proof.*

$$v_\tau^{\pi_\tau^*} - v_\tau^\pi = \mathcal{T}_\tau^{\pi_\tau^*} v_\tau^{\pi_\tau^*} - v_\tau^\pi \qquad \text{(Since } v_\tau^{\pi_\tau^*} \text{ is a fixed point of } \mathcal{T}_\tau^{\pi_\tau^*})$$

$$= [\mathcal{T}_\tau^{\pi_\tau^*} v_\tau^\pi - v_\tau^\pi] + [\mathcal{T}_\tau^{\pi_\tau^*} v_\tau^{\pi_\tau^*} - \mathcal{T}_\tau^{\pi_\tau^*} v_\tau^\pi] \qquad \text{(Add/subtract } \mathcal{T}_\tau^{\pi_\tau^*} v_\tau^\pi)$$

$$= [\mathcal{T}_\tau^{\pi_\tau^*} v_\tau^\pi - v_\tau^\pi] + \gamma \mathcal{P}_{\pi_\tau^*}(v_\tau^{\pi_\tau^*} - v_\tau^\pi) \qquad \text{(Using the definition of } \mathcal{T}_\tau^{\pi_\tau^*})$$

$$\implies v_\tau^{\pi_\tau^*} - v_\tau^\pi = (I - \gamma \mathcal{P}_{\pi_\tau^*})^{-1} \left[ \mathcal{T}_\tau^{\pi_\tau^*} v_\tau^\pi - \mathcal{T}_\tau^\pi v_\tau^\pi \right]$$

Summing up from $t = 0$ to $t = K - 1$ and dividing by $K$,

$$v_\tau^* - \frac{\sum_{t=0}^{K-1} v_\tau^{\pi_t}}{K} = \frac{1}{K} (I - \gamma \mathcal{P}_{\pi_\tau^*})^{-1} \sum_{t=0}^{K-1} \left[ \mathcal{T}_\tau^{\pi_\tau^*} v_\tau^{\pi_t} - v_\tau^{\pi_t} \right]$$

$$\text{(By definition of } v_\tau^*)$$

$$\implies v_\tau^* - v_\tau^{\bar{\pi}_K} = \frac{1}{K} (I - \gamma \mathcal{P}_{\pi_\tau^*})^{-1} \sum_{t=0}^{K-1} \left[ \mathcal{T}_\tau^{\pi_\tau^*} v_\tau^{\pi_t} - v_\tau^{\pi_t} \right] \qquad \text{(Since } v_\tau^{\bar{\pi}_K} = \frac{\sum_{t=0}^{K-1} v_\tau^{\pi_t}}{K})$$

$$= \frac{1}{K} (I - \gamma \mathcal{P}_{\pi_\tau^*})^{-1} \sum_{t=0}^{K-1} \left[ \mathcal{T}_\tau^{\pi_\tau^*} v_\tau^{\pi_t} - \mathcal{T}_\tau^{\pi_t} v_\tau^{\pi_t} \right] \qquad \text{(Since } v_\tau^\pi = \mathcal{T}_\tau^\pi v_\tau^\pi)$$

$$\implies \left\| v_\tau^* - v_\tau^{\bar{\pi}_K} \right\|_\infty = \frac{1}{K} \left\| (I - \gamma \mathcal{P}_{\pi_\tau^*})^{-1} \sum_{t=0}^{K-1} \left[ \mathcal{T}_\tau^{\pi_\tau^*} v_\tau^{\pi_t} - \mathcal{T}_\tau^{\pi_t} v_\tau^{\pi_t} \right] \right\|_\infty$$

$$\leq \frac{1}{K} \left\| (I - \gamma \mathcal{P}_{\pi_\tau^*})^{-1} \right\|_\infty \left\| \sum_{t=0}^{K-1} \left[ \mathcal{T}_\tau^{\pi_\tau^*} v_\tau^{\pi_t} - \mathcal{T}_\tau^{\pi_t} v_\tau^{\pi_t} \right] \right\|_\infty$$

(By definition of matrix norm)

$$\leq \frac{1}{K(1-\gamma)} \left\| \sum_{t=0}^{K-1} \left[ \mathcal{T}_\tau^{\pi_\tau^*} v_\tau^{\pi_t} - \mathcal{T}_\tau^{\pi_t} v_\tau^{\pi_t} \right] \right\|_\infty$$

(Since $\left\| (I - \gamma \mathcal{P}_\pi)^{-1} \right\|_\infty = \|\sum_{t=0}^\infty [\gamma \mathcal{P}_\pi]^t\|_\infty \leq \sum_{t=0}^\infty \gamma^t = \frac{1}{1-\gamma}$)

Let us calculate $\left[ \mathcal{T}_\tau^{\pi_\tau^*} v_\tau^{\pi_t} - \mathcal{T}_\tau^{\pi_t} v_\tau^{\pi_t} \right](s)$.

$$\left[ \mathcal{T}_\tau^{\pi_\tau^*} v_\tau^{\pi_t} - \mathcal{T}_\tau^{\pi_t} v_\tau^{\pi_t} \right](s) = (\mathcal{T}_\tau^{\pi_\tau^*} v_\tau^{\pi_t})(s) - (\mathcal{T}_\tau^{\pi_t} v_\tau^{\pi_t})(s)$$

$$= \mathbb{E}_{a \sim \pi_\tau^*} \left[ q_\tau^{\pi_t}(s,a) - \tau \ln(\pi_\tau^*(a|s)) \right]$$

$$- \mathbb{E}_{a \sim \pi_t} \left[ q_\tau^{\pi_t}(s,a) - \tau \ln(\pi_t(a|s)) \right] \quad \text{(By definition of } \mathcal{T}_\tau^{\pi_1} v_\tau^{\pi_2})$$

$$= \langle \pi_\tau^*(\cdot|s) - \pi_t(\cdot|s), q_\tau^{\pi_t}(s,\cdot) \rangle$$

$$- \tau \left[ \langle \pi_\tau^*(\cdot|s), \ln(\pi_\tau^*(\cdot|s)) \rangle - \langle \pi_t(\cdot|s), \ln(\pi_t(\cdot|s)) \rangle \right]$$

$$= \langle \pi_\tau^*(\cdot|s) - \pi_t(\cdot|s), q_\tau^{\pi_t}(s,\cdot) \rangle + \tau \left[ \mathcal{H}(\pi_\tau^*(\cdot|s)) - \mathcal{H}(\pi_t(\cdot|s)) \right]$$

(By definition of $\mathcal{H}(\pi(\cdot|s))$)

$$= \langle \pi_\tau^*(\cdot|s) - \pi_t(\cdot|s), q_\zeta^t(s,\cdot) \rangle + \tau \left[ \mathcal{H}(\pi_\tau^*(\cdot|s)) - \mathcal{H}(\pi_t(\cdot|s)) \right]$$

$$+ \langle \pi_\tau^*(\cdot|s) - \pi_t(\cdot|s), q_\tau^{\pi_t}(s,\cdot) - q_\zeta^t(s,\cdot) \rangle$$

(Add/Subtract $\langle \pi_\tau^*(\cdot|s) - \pi_t(\cdot|s), q_\zeta^t(s,\cdot) \rangle$)

$$\leq \langle \pi_\tau^*(\cdot|s) - \pi_t(\cdot|s), q_\zeta^t(s,\cdot) \rangle + \tau \left[ \mathcal{H}(\pi_\tau^*(\cdot|s)) - \mathcal{H}(\pi_t(\cdot|s)) \right]$$

$$+ \left\| \pi_\tau^*(\cdot|s) - \pi_t(\cdot|s) \right\|_1 \left\| q_\tau^{\pi_t}(s,\cdot) - q_\zeta^t(s,\cdot) \right\|_\infty$$

(By Holder's inequality)

$$\leq \langle \pi_\tau^*(\cdot|s) - \pi_t(\cdot|s), q_\zeta^t(s,\cdot) \rangle + \tau \left[ \mathcal{H}(\pi_\tau^*(\cdot|s)) - \mathcal{H}(\pi_t(\cdot|s)) \right]$$

$$+ 2 \left\| q_\tau^{\pi_t}(s,\cdot) - q_\zeta^t(s,\cdot) \right\|_\infty \quad \text{(Since } \|\pi_\tau^*(\cdot|s) - \pi_t(\cdot|s)\|_1 \leq 2)$$

Define $\text{Regret}(K,u,s) = \sum_{t=0}^{K-1} \left[ \langle u(\cdot|s) - \pi_t(\cdot|s), q_\zeta^t(s,\cdot) \rangle + \tau \left[ \mathcal{H}(u(\cdot|s)) - \mathcal{H}(\pi_t(\cdot|s)) \right] \right]$ as the regret incurred for state $s$ when the comparator is policy $u$. Hence,

$$\sum_{t=0}^{K-1} \left[ \mathcal{T}_\tau^{\pi_\tau^*} v_\tau^{\pi_t} - \mathcal{T}_\tau^{\pi_t} v_\tau^{\pi_t} \right](s) \leq \text{Regret}(K,\pi_\tau^*,s) + 2 \sum_{t=0}^K \left\| q_\tau^{\pi_t}(s,\cdot) - q_\zeta^t(s,\cdot) \right\|_\infty$$

$$\left\| \sum_{t=0}^{K-1} \left[ \mathcal{T}_\tau^{\pi_\tau^*} v_\tau^{\pi_t} - \mathcal{T}_\tau^{\pi_t} v_\tau^{\pi_t} \right] \right\|_\infty \leq \max_s \text{Regret}(K,\pi_\tau^*,s) + 2 \sum_{t=0}^K \left\| \boldsymbol{\epsilon}_t \right\|_\infty \quad \text{(By definition of } \boldsymbol{\epsilon}_t)$$

Using the definition of $\text{Regret}(K) = \left[ \text{Regret}(K,\pi_\tau^*,s_i) \right]_{i=1}^S \in \mathbb{R}^S$,

$$\left\| v_\tau^* - v_\tau^{\bar{\pi}_K} \right\|_\infty \leq \frac{\|\text{Regret}(K)\|_\infty}{K(1-\gamma)} + \frac{2 \sum_{t \in [K]} \|\boldsymbol{\epsilon}_t\|_\infty}{K(1-\gamma)}$$

$\square$

In the special case $(0, \zeta)$, i.e., no entropy regularization from the actor side, the problem reduces to online linear optimization, as in Politex (Abbasi-Yadkori et al., 2019), yielding the following corollary.

**Corollary 3** (Generic Reduction without Actor Entropy). *If $\pi^*$ is the optimal policy whose value function is equal to $v^*$, for an estimate of $q_\tau^{\pi_t}$ at iteration $t$ s.t. $\boldsymbol{\epsilon}_t = q_\zeta^t - q_\tau^{\pi_t}$ and for*

any sequence of policies $\{\pi_0, \pi_1, \ldots, \pi_{K-1}\}$, if $\bar{\pi}_K$ is the corresponding mixture policy, then,

$$\left\|v^* - v^{\bar{\pi}_K}\right\|_\infty \le \frac{\|Regret(K)\|_\infty}{K\,(1-\gamma)} + \frac{2\sum_{t\in[K]}\|\epsilon_t\|_\infty}{K\,(1-\gamma)},$$

where $(Regret(K))(s) := \sum_{t=0}^{K-1}\left[\langle\pi^*(\cdot|s) - \pi_t(\cdot|s), q_\zeta^t(s,\cdot)\rangle\right]$ is the regret incurred on an online linear optimization problem for each state $s \in \mathcal{S}$.

## C.2 Proof of Corollary 1

### C.2.1 Generic Policy Evaluation

**Theorem 3** (Generic policy evaluation)**. *Using the policy evaluation scheme in Eq. (1), if $\delta(\tau, \zeta) := \frac{|\tau - \zeta|\,\ln(A)}{1-\gamma}$, and $TV_i := \|\pi_i(\cdot|s) - \pi_{i-1}(\cdot|s)\|_1$, then, for all $t \in [K]$,*

$$\epsilon_t := \|\epsilon_t\|_\infty = \left\|q_\tau^{\pi_t} - q_\zeta^t\right\|_\infty$$

$$\le \frac{H_\tau\,\gamma^m}{1-\gamma}\sum_{i=1}^t(\gamma^m)^{t-i}\max_s\left[TV_i + \tau(1-\gamma)\left|\mathcal{H}(\pi_i(\cdot|s)) - \mathcal{H}(\pi_{i-1}(\cdot|s))\right|\right]$$

$$+ \sum_{i=0}^t(\gamma^m)^{t-i}\,\delta(\tau, \zeta)$$

*Furthermore, if for all $i \in [t]$, $TV_i \le \frac{1}{2}$, then, for any constant $C \in (0, 1/2)$,*

$$\epsilon_t \le \frac{H_\tau\,\gamma^m}{(1-\gamma)}\sum_{i=1}^t(\gamma^m)^{t-i}\max_s\left[TV_i + \tau(1-\gamma)\left[TV_i\ln\left(\frac{A}{C}\right) + \left(\frac{\ln(A)}{2} + \sqrt{2}\right)\sqrt{C}\right]\right]$$

$$+ \sum_{i=0}^t(\gamma^m)^{t-i}\,\delta(\tau, \zeta)$$

*Proof.*

$$\epsilon_t = \left\|q_\tau^{\pi_t} - q_\zeta^t\right\|_\infty = \left\|q_\tau^{\pi_t} - \mathcal{P}_{[0,H_\tau]}[(T_\zeta^{\pi_t})^m q_\zeta^{t-1}]\right\|_\infty \qquad \text{(Using the update)}$$

$$= \left\|\mathcal{P}_{[0,H_\tau]}[q_\tau^{\pi_t}] - \mathcal{P}_{[0,H_\tau]}[(T_\zeta^{\pi_t})^m q_\zeta^{t-1}]\right\|_\infty \qquad \text{(Since } q_\tau^{\pi_t} \in [0, H_\tau])$$

$$= \left\|\mathcal{P}_{[0,H_\tau]}[(T_\tau^{\pi_t})^m q_\tau^{\pi_t}] - \mathcal{P}_{[0,H_\tau]}[(T_\zeta^{\pi_t})^m q_\zeta^{t-1}]\right\|_\infty \qquad \text{(Since } q_\tau^{\pi_t} = T_\tau^{\pi_t} q_\tau^{\pi_t})$$

$$\le \left\|(T_\tau^{\pi_t})^m q_\tau^{\pi_t} - (T_\zeta^{\pi_t})^m q_\zeta^{t-1}\right\|_\infty \qquad \text{(Since projections are non-expansive)}$$

$$\le \left\|(T_\tau^{\pi_t})^m q_\tau^{\pi_t} - (T_\tau^{\pi_t})^m q_\zeta^{t-1}\right\|_\infty + \left\|(T_\tau^{\pi_t})^m q_\zeta^{t-1} - (T_\zeta^{\pi_t})^m q_\zeta^{t-1}\right\|_\infty$$
$$\text{(Add/Subtract } (T_\tau^{\pi_t})^m q_\zeta^{t-1} \text{ and using triangle inequality)}$$

$$\le \left\|(T_\tau^{\pi_t})^m q_\tau^{\pi_t} - (T_\tau^{\pi_t})^m q_\zeta^{t-1}\right\|_\infty + \underbrace{\frac{|\zeta - \tau|}{1-\gamma}\ln(A)}_{:=\delta(\tau,\zeta)} \qquad \text{(Using Lemma 4)}$$

$$= \left\|(T_\tau^{\pi_t})^m q_\tau^{\pi_t} - (T_\tau^{\pi_t})^m q_\zeta^{t-1}\right\|_\infty + \delta(\tau, \zeta)$$

$$\le \gamma^m\left\|q_\tau^{\pi_t} - q_\zeta^{t-1}\right\|_\infty + \delta(\tau, \zeta) \qquad \text{(Since } T_\tau^\pi \text{ is a } \gamma \text{ contraction)}$$

$$\le \gamma^m\left\|q_\tau^{\pi_t} - q_\tau^{\pi_{t-1}}\right\|_\infty + \gamma^m\left\|q_\tau^{\pi_{t-1}} - q_\zeta^{t-1}\right\|_\infty + \delta(\tau, \zeta)$$
$$\text{(Add/subtract } q_\tau^{\pi_{t-1}} \text{ and using triangle inequality)}$$

$$= \gamma^m\left\|q_\tau^{\pi_t} - q_\tau^{\pi_{t-1}}\right\|_\infty + \gamma^m\,\epsilon_{t-1} + \delta(\tau, \zeta)$$

The first term is the difference in the $q_\tau$ functions between two consecutive policies, and we bound it next. For all $(s, a)$,

$$q_\tau^{\pi_t}(s, a) - q_\tau^{\pi_{t-1}}(s, a) = \gamma \sum_{s'} \mathcal{P}(s'|s, a) v_\tau^{\pi_t}(s') - \gamma \sum_{s'} \mathcal{P}(s'|s, a) v_\tau^{\pi_{t-1}}(s')$$

$$= \gamma \, \mathbb{E}_{s' \sim \mathcal{P}(\cdot|s,a)} [v_\tau^{\pi_t}(s') - v_\tau^{\pi_{t-1}}(s')]$$

$$\implies \|q_\tau^{\pi_t} - q_\tau^{\pi_{t-1}}\|_\infty \leq \gamma \|v_\tau^{\pi_t} - v_\tau^{\pi_{t-1}}\|_\infty$$

Let us now bound the difference in the $v_\tau$ functions between two consecutive policies.

$$\|v_\tau^{\pi_t} - v_\tau^{\pi_{t-1}}\|_\infty = \|T_\tau^{\pi_t} v_\tau^{\pi_t} - T_\tau^{\pi_t} v_\tau^{\pi_{t-1}} + T_\tau^{\pi_t} v_\tau^{\pi_{t-1}} - T_\tau^{\pi_{t-1}} v_\tau^{\pi_{t-1}}\|_\infty$$

$$\text{(Since } T_\tau^\pi v_\tau^\pi = v_\tau^\pi \text{ and add/subtract } T_\tau^{\pi_t} v_\tau^{\pi_{t-1}})$$

$$\leq \|T_\tau^{\pi_t} v_\tau^{\pi_t} - T_\tau^{\pi_t} v_\tau^{\pi_{t-1}}\|_\infty + \|T_\tau^{\pi_t} v_\tau^{\pi_{t-1}} - T_\tau^{\pi_{t-1}} v_\tau^{\pi_{t-1}}\|_\infty$$

$$\text{(Triangle inequality)}$$

$$\leq \gamma \|v_\tau^{\pi_t} - v_\tau^{\pi_{t-1}}\|_\infty + \|T_\tau^{\pi_t} v_\tau^{\pi_{t-1}} - T_\tau^{\pi_{t-1}} v_\tau^{\pi_{t-1}}\|_\infty$$

$$\text{(Since } T_\tau^\pi \text{ is a } \gamma\text{-contraction)}$$

$$\implies \|v_\tau^{\pi_t} - v_\tau^{\pi_{t-1}}\|_\infty \leq \frac{1}{1 - \gamma} \|T_\tau^{\pi_t} v_\tau^{\pi_{t-1}} - T_\tau^{\pi_{t-1}} v_\tau^{\pi_{t-1}}\|_\infty$$

In order to bound $\left\|T_\tau^{\pi_t} v_\tau^{\pi_{t-1}} - T_\tau^{\pi_{t-1}} v_\tau^{\pi_{t-1}}\right\|_\infty$, consider a fixed state $s$. By definition of $T_\tau^\pi$,

$$T_\tau^{\pi_t} v_\tau^{\pi_{t-1}}(s) - T_\tau^{\pi_{t-1}} v_\tau^{\pi_{t-1}}(s) = \langle \pi_t(\cdot|s) - \pi_{t-1}(\cdot|s), r(s, \cdot) \rangle$$

$$+ \gamma \sum_a [\pi_t(a|s) - \pi_{t-1}(a|s)] \, \mathbb{E}_{s' \sim \mathcal{P}(\cdot|s,a)} v_\tau^{\pi_{t-1}}(s')$$

$$+ \tau [\mathcal{H}(\pi_t(\cdot|s)) - \mathcal{H}(\pi_{t-1}(\cdot|s))]$$

$$\leq \|\pi_t(\cdot|s) - \pi_{t-1}\|_1 \, \|r(s, \cdot)\|_\infty$$

$$+ \gamma \|\pi_t(\cdot|s) - \pi_{t-1}\|_1 \, \|v_\tau^{\pi_{t-1}}\|_\infty$$

$$+ \tau [\mathcal{H}(\pi_t(\cdot|s)) - \mathcal{H}(\pi_{t-1}(\cdot|s))] \text{ (By Holder's inequality)}$$

$$\leq (1 + \gamma H_\tau) \|\pi_t(\cdot|s) - \pi_{t-1}(\cdot|s)\|_1$$

$$+ \tau |\mathcal{H}(\pi_t(\cdot|s)) - \mathcal{H}(\pi_{t-1}(\cdot|s))|$$

$$\text{(Since rewards are in } [0, 1] \text{ and } v_\tau^{\pi_{t-1}}(s) \leq H_\tau)$$

$$\leq H_\tau \|\pi_t(\cdot|s) - \pi_{t-1}(\cdot|s)\|_1 + \tau |\mathcal{H}(\pi_t(\cdot|s)) - \mathcal{H}(\pi_{t-1}(\cdot|s))|$$

$$\text{(Since } 1 \leq 1 + \tau \ln(A) = (1 - \gamma) H_\tau)$$

$$\implies \|T_\tau^{\pi_t} v_\tau^{\pi_{t-1}} - T_\tau^{\pi_{t-1}} v_\tau^{\pi_{t-1}}\|_\infty \leq H_\tau \left( \max_s [TV_i + \tau(1 - \gamma) |\mathcal{H}(\pi_t(\cdot|s)) - \mathcal{H}(\pi_{t-1}(\cdot|s))|] \right)$$

$$\text{(Since } 1 \leq (1 - \gamma) H_\tau)$$

Combining the above inequalities,

$$\|v_\tau^{\pi_t} - v_\tau^{\pi_{t-1}}\|_\infty \leq \frac{H_\tau}{(1 - \gamma)} \max_s [TV_i + \tau(1 - \gamma) |\mathcal{H}(\pi_t(\cdot|s)) - \mathcal{H}(\pi_{t-1}(\cdot|s))|]$$

$$\implies \|q_\tau^{\pi_t} - q_\tau^{\pi_{t-1}}\|_\infty \leq \frac{H_\tau}{(1 - \gamma)} \max_s [TV_i + \tau(1 - \gamma) |\mathcal{H}(\pi_t(\cdot|s)) - \mathcal{H}(\pi_{t-1}(\cdot|s))|]$$

$$\implies \epsilon_t \leq \underbrace{\frac{H_\tau \gamma^m}{(1 - \gamma)} \max_s [TV_i + \tau(1 - \gamma) |\mathcal{H}(\pi_t(\cdot|s)) - \mathcal{H}(\pi_{t-1}(\cdot|s))|]}_{:= B_t} + \gamma^m \epsilon_{t-1} + \delta(\tau, \zeta)$$

$$\implies \epsilon_t \leq B_t + \gamma^m \epsilon_{t-1} + \delta(\tau, \zeta)$$

Bounding $\epsilon_0$,

$$\epsilon_0 = q_\tau^{\pi_0} - q_\zeta^{\pi_0} = \mathbb{E}_{s' \sim \mathcal{P}(\cdot|s,a)}\left[r(s,a) + \gamma v_\tau^\pi(s')\right] - \mathbb{E}_{s' \sim \mathcal{P}(\cdot|s,a)}\left[r(s,a) + \gamma v_\zeta^\pi(s')\right]$$

(By definition)

$$= \gamma \mathbb{E}_{s' \sim \mathcal{P}(\cdot|s,a)}[v_\tau^{\pi_0}(s') - v_\zeta^{\pi_0}(s')]$$

$$= \gamma \mathbb{E}_{s' \sim \mathcal{P}(\cdot|s,a)}\left[\left[v^{\pi_0}(s') + \tau \sum_{t=0}^\infty \gamma^t \left[\mathcal{H}(\pi_0(\cdot|s_t))|s_0 = s'\right]\right.\right.$$

$$\left.\left. - [v^{\pi_0}(s') + \zeta \sum_{t=0}^\infty \gamma^t \left[\mathcal{H}(\pi_0(\cdot|s_t))|s_0 = s'\right]\right]\right]$$

(By definition)

$$\leq \frac{|\tau - \zeta| \ln(A)}{1 - \gamma} = \delta(\tau, \zeta)$$

(Since $\mathcal{H}(\pi(\cdot|s)) \leq \ln(A)$)

For a fixed $t \in [K]$, recursing from $i = t - 1$ to $i = 1$ and using that

$$\epsilon_t \leq (\gamma^m)^t \epsilon_0 + \sum_{i=1}^t (\gamma^m)^{t-i} \left(B_i + \delta(\tau, \zeta)\right)$$

$$\leq \frac{H_\tau \gamma^m}{(1-\gamma)} \sum_{i=1}^t (\gamma^m)^{t-i} \max_s \left[TV_i + \tau(1-\gamma)\left|\mathcal{H}(\pi_i(\cdot|s)) - \mathcal{H}(\pi_{i-1}(\cdot|s))\right|\right]$$

$$+ \sum_{i=0}^t (\gamma^m)^{t-i} \delta(\tau, \zeta)$$

Furthermore, if $TV_i \leq \frac{1}{2}$ for all $i \in [t]$, using Lemma 1, we can further upper-bound $|\mathcal{H}(\pi_i(\cdot|s)) - \mathcal{H}(\pi_{i-1}(\cdot|s))|$ to get that for any constant $C \in (0, 1/2)$,

$$|\mathcal{H}(\pi_i(\cdot|s)) - \mathcal{H}(\pi_{i-1}(\cdot|s))| \leq TV_i \ln\left(\frac{A}{C}\right) + \left(\frac{\ln(A-1)}{2} + \sqrt{2}\right)\sqrt{C}$$

Combining the above relations, in this case, we get that,

$$\epsilon_t \leq \frac{H_\tau \gamma^m}{(1-\gamma)} \sum_{i=1}^t (\gamma^m)^{t-i} \max_s \left[TV_i + \tau(1-\gamma)\left[TV_i \ln\left(\frac{A}{C}\right) + \left(\frac{\ln(A)}{2} + \sqrt{2}\right)\sqrt{C}\right]\right]$$

$$+ \sum_{i=0}^t (\gamma^m)^{t-i} \delta(\tau, \zeta)$$

$\square$

**Corollary 4.** *Using the policy evaluation scheme in Eq. (1) with $\zeta = \tau$, for all $t \in [K]$, if for all $i \in [t]$, $TV_i := \|\pi_i(\cdot|s) - \pi_{i-1}(\cdot|s)\|_1 \leq \frac{1}{2}$, then, for any constant $C \in (0, 1/2)$,*

$$\epsilon_t \leq \frac{H_\tau \gamma^m}{(1-\gamma)} \sum_{i=1}^t (\gamma^m)^{t-i} \max_s \left[TV_i + \tau(1-\gamma)\left[TV_i \ln\left(\frac{A}{C}\right) + \left(\frac{\ln(A)}{2} + \sqrt{2}\right)\sqrt{C}\right]\right]$$

*Proof.* Setting $\zeta = \tau$ in Theorem 3. $\square$

**Corollary 5.** *Using the policy evaluation scheme in Eq. (1) with $\zeta = 0$, for all $t \in [K]$, if for all $i \in [t]$, $TV_i := \|\pi_i(\cdot|s) - \pi_{i-1}(\cdot|s)\|_1 \leq \frac{1}{2}$, then, for any constant $C \in (0, 1/2)$,*

$$\epsilon_t \leq \frac{H_\tau \gamma^m}{(1-\gamma)} \sum_{i=1}^t (\gamma^m)^{t-i} \max_s \left[TV_i + \tau(1-\gamma)\left[TV_i \ln\left(\frac{A}{C}\right) + \left(\frac{\ln(A)}{2} + \sqrt{2}\right)\sqrt{C}\right]\right]$$

$$+ \frac{\tau \ln(A)}{(1-\gamma)^2}$$

*Proof.* Setting $\zeta = 0$ in Theorem 3. $\qquad\qquad\square$

The following proposition shows that the objective in Eq. (5b) admits a closed-form solution. Substituting the intermediate policies NPG and SPMA into the closed-form expression in Eq. (6) yields their entropy-regularized (actor entropy) counterparts, referred to as soft NPG and soft SPMA.

**Proposition 1.** *If* $\alpha_t := \frac{1}{1+\tau_t}$, *the closed-form solution for the proximal update in Eq.* (5b) *for any* $s, a$ *is given as,*

$$\pi_{t+1}(a|s) = \frac{[\pi_{t+1/2}(a|s)]^{\alpha_t}}{\sum_{a'}[\pi_{t+1/2}(a'|s)]^{\alpha_t}} \qquad (6)$$

*Proof.*

$$\pi_{t+1}(\cdot|s) = \underset{\pi(\cdot|s)\in\Delta}{\arg\min} \left[ \mathrm{KL}(\pi(\cdot|s)||\pi_{t+1/2}(\cdot|s)) - \tau_t \mathcal{H}(\pi(\cdot|s)) \right]$$

$$= \underset{\pi(\cdot|s)\in\Delta}{\arg\min} \mathbb{E}_{a\sim\pi(.|s)} \left[ (1 + \tau_t) \ln(\pi(.|s)) - \ln(\pi_{t+1/2}(\cdot|s)) \right]$$

$$\text{(Using the definition of } H(\pi(\cdot|s)))$$

$$= \underset{\pi(\cdot|s)\in\Delta}{\arg\min} \mathbb{E}_{a\sim\pi(.|s)} \left[ \ln(\pi(.|s)) - \alpha_t \ln(\pi_{t+1/2}(\cdot|s)) \right] \qquad (\text{Since } \alpha_t = \tfrac{1}{1+\tau_t})$$

$$= \underset{\pi(\cdot|s)\in\Delta}{\arg\min} \mathbb{E}_{a\sim\pi(.|s)} \left[ \ln(\pi(.|s)) - \ln([\pi_{t+1/2}(\cdot|s)]^{\alpha_t}) \right]$$

$$= \underset{\pi(\cdot|s)\in\Delta}{\arg\min} \mathrm{KL}(\pi(.|s)||\pi_{t+1/2}(\cdot|s)^{\alpha_t}) \qquad (\text{By definition of the KL divergence})$$

Using the fact that KL projection onto the simplex results in normalization, we get that,

$$\pi_{t+1}(a|s) = \frac{[\pi_{t+1/2}(a|s)]^{\alpha_t}}{\sum_{a'}[\pi_{t+1/2}(a'|s)]^{\alpha_t}}$$

$\qquad\qquad\square$

Note that when $\tau = 0$, we have $\tau_t = 0$ and $\alpha_t = 1$ for all $t$, recovering the standard unregularized updates for both NPG and SPMA. For $\tau > 0$, the soft updates can be expressed as: for any $s, a$, with $\alpha_t = \frac{1}{1+\tau_t}$ and $\tau_t = \eta_t \tau$,

$$\pi_{t+1}(a|s) = \frac{[\pi_t(a|s)]^{\alpha_t} \exp(\eta_t \alpha_t q_\zeta^t(s,a))}{\mathcal{Z}_t}, \qquad (7)$$

$$\text{with} \quad \mathcal{Z}_t = \sum_{a'}[\pi_t(a'|s)]^{\alpha_t} \exp(\eta_t \alpha_t q_\zeta^t(s,a')), \qquad \textbf{(Soft-NPG)}$$

$$\pi_{t+1}(a|s) = \frac{[\pi_t(a|s)]^{\alpha_t} \left[ 1 + \eta_t \left( q_\zeta^t(s,a) - v_\zeta^t(s) \right) \right]^{\alpha_t}}{\mathcal{Z}_t}, \qquad (8)$$

$$\text{with} \quad \mathcal{Z}_t = \sum_{a'}[\pi_t(a'|s)]^{\alpha_t} \left[ 1 + \eta_t \left( q_\zeta^t(s,a') - v_\zeta^t(s) \right) \right]^{\alpha_t}, \qquad \textbf{(Soft-SPMA)}$$

### C.2.2 POLICY ERROR BOUND FOR (SOFT) NPG

In the next corollary, we use Theorem 3 with $\zeta = \tau$ to instantiate the policy error bound for soft NPG with entropy-regularized policy evaluation.

**Corollary 6** (Policy evaluation with $\zeta = \tau$). *Using the policy evaluation update in Eq.* (1) *with* $\zeta = \tau$, *for soft NPG with* $\eta_t = \frac{1}{c+\tau\,(t+1)}$ *and a constant* $c \geq \max\left\{ \frac{8\,(1+\tau\,\ln(A))}{(1-\gamma)}, 32\,\tau\,\ln(A) \right\}$, *for all* $t \in [K]$, $\epsilon_t := \|\boldsymbol{\epsilon}_t\|_\infty$ *can be bounded as:*

$$\epsilon_t \leq \frac{8(1 + \tau\ln(A))\gamma^m}{(1-\gamma)^3} \left[ \left( 1 + \tau\ln\left(A\,K^4\right) \right) \right.$$

$$\left(\left(\ln(A\,K^4) + \frac{1 + \tau\ln(A)}{\tau(1-\gamma)}\right)\left(\frac{1}{t} + (\gamma^m)^{t/2}\right) + \frac{\sqrt{\tau}}{K}\right)$$

$$+ \tau\,(\ln(A) + 1)\,\frac{1}{K^2}\Bigg]$$

*Proof.* For a fixed iteration $t \in [K]$ and state $s \in \mathcal{S}$, let us first bound $\|\pi_{t+1}(\cdot|s) - \pi_t(\cdot|s)\|_1$.

$$\|\pi_{t+1}(\cdot|s) - \pi_t(\cdot|s)\|_1 \le \|\pi_{t+1}(\cdot|s) - \pi_{t+1/2}(\cdot|s)\|_1 + \|\pi_{t+1/2}(\cdot|s) - \pi_t(\cdot|s)\|_1$$
$$\text{(Triangle inequality)}$$

We first bound $\|\pi_{t+1/2}(\cdot|s) - \pi_t(\cdot|s)\|_1$. Using the mirror descent view of `NPG` (Xiao, 2022), the update can be written as:

$$\pi_{t+1/2}(\cdot|s) = \underset{\pi\in\Delta}{\arg\min}\left[-\eta_t\langle q_\tau^t(s,\cdot), \pi(\cdot|s)\rangle + \mathrm{KL}(\pi(\cdot|s)\|\pi_t(\cdot|s))\right]$$

$$\implies -\eta_t\langle q_\tau^t(s,\cdot), \pi_{t+1/2}(\cdot|s)\rangle + \mathrm{KL}(\pi_{t+1/2}(\cdot|s)\|\pi_t(\cdot|s)) \le -\eta_t\langle q_\tau^t(s,\cdot), \pi_t(\cdot|s)\rangle$$
$$+ \mathrm{KL}(\pi_t(\cdot|s)\|\pi_t(\cdot|s))$$

$$\implies \frac{1}{2}\|\pi_{t+1/2}(\cdot|s) - \pi_t(\cdot|s)\|_1^2 \le \mathrm{KL}(\pi_{t+1/2}(\cdot|s)\|\pi_t(\cdot|s)) \le \eta_t\langle q_\tau^t(s,\cdot), \pi_{t+1/2}(\cdot|s) - \pi_t(\cdot|s)\rangle$$
$$\text{(By Pinsker's inequality)}$$

$$\le \eta_t\,\|q_\tau^t(s,\cdot)\|_\infty\,\|\pi_{t+1/2}(\cdot|s) - \pi_t(\cdot|s)\|_1$$
$$\text{(By Holder's inequality)}$$

$$\implies \|\pi_{t+1/2}(\cdot|s) - \pi_t(\cdot|s)\|_1 \le 2\eta_t\,H_\tau \qquad \text{(Since } \|q_\tau^t(s,\cdot)\|_\infty \le H_\tau)$$

In order to bound $\|\pi_{t+1}(\cdot|s) - \pi_{t+1/2}(\cdot|s)\|_1$, we use the Eq. (5b) update. Specifically,

$$\pi_{t+1}(\cdot|s) = \underset{\pi\in\Delta}{\arg\min}\left[\mathrm{KL}(\pi(\cdot|s)\|\pi_{t+1/2}(\cdot|s)) - \tau_t\mathcal{H}(\pi(\cdot|s))\right]$$

$$\implies \mathrm{KL}(\pi_{t+1}(\cdot|s)\|\pi_{t+1/2}(\cdot|s)) - \tau_t\mathcal{H}(\pi_{t+1}(\cdot|s)) \le \mathrm{KL}(\pi_{t+1/2}(\cdot|s)\|\pi_{t+1/2}(\cdot|s))$$
$$- \tau_t\mathcal{H}(\pi_{t+1/2}(\cdot|s))$$

$$\frac{1}{2}\|\pi_{t+1}(\cdot|s) - \pi_{t+1/2}(\cdot|s)\|_1^2 \le \mathrm{KL}(\pi_{t+1}\|\pi_{t+1/2}(\cdot|s))$$
$$\text{(By Pinsker's inequality)}$$

$$\le \tau_t\mathcal{H}(\pi_{t+1}(\cdot|s)) - \tau_t\mathcal{H}(\pi_{t+1/2}(\cdot|s))$$

$$\le \tau_t\,\ln(A)$$

$$\implies \|\pi_{t+1}(\cdot|s) - \pi_{t+1/2}(\cdot|s)\|_1 \le \sqrt{2\,\tau\,\eta_t\,\ln(A)} \qquad \text{(since } \tau_t = \tau\,\eta_t)$$

Combining the above relations,

$$\|\pi_{t+1}(\cdot|s) - \pi_t(\cdot|s)\|_1 \le \sqrt{2\,\tau\,\eta_t\,\ln(A)} + 2\,\eta_t\,H_\tau \tag{9}$$

Using Eq. (9) in Theorem 3, for the special case $\|\pi_{t+1}(\cdot|s) - \pi_t(\cdot|s)\|_1 \le \frac{1}{2}$ for all $t \in [K]$, $\eta_t$ must satisfy the following conditions:

- $2\,\eta_t\,H_\tau \le \frac{1}{4} \implies \eta_t \le \frac{(1-\gamma)}{8\,(1+\tau\,\ln(A))}$
- $\sqrt{2\,\tau\,\eta_t\,\ln(A)} \le \frac{1}{4} \implies \eta_t \le \frac{1}{32\,\tau\,\ln(A)}$

For $\eta_t = \frac{1}{c+\tau(t+1)} \le \frac{1}{c}$, it is thus sufficient to ensure that,

$$c \ge \max\left\{\frac{8\,(1+\tau\,\ln(A))}{(1-\gamma)}, 32\,\tau\,\ln(A)\right\}$$

Given this constraint on $c$, we use Theorem 3 with $\zeta = \tau$ and $TV_i := \|\pi_i(\cdot|s) - \pi_{i-1}(\cdot|s)\|_1$, to get that, for any constant $C \in (0, 1/2)$, for all $t \in [K]$,

$$\epsilon_t \le \frac{H_\tau \gamma^m}{(1 - \gamma)} \sum_{i=1}^{t} (\gamma^m)^{t-i} \max_s \left[ TV_i + \tau(1-\gamma) \left[ TV_i \ln\left(\frac{A}{C}\right) + \left(\frac{\ln(A)}{2} + \sqrt{2}\right) \sqrt{C} \right] \right]$$

Observe that the choice of $c$ ensures $\|\pi_{t+1}(\cdot|s) - \pi_t(\cdot|s)\|_1 \le \frac{1}{2}$. Using Lemma 2 we obtain,

$$\|\pi_{t+1}(\cdot|s) - \pi_t(\cdot|s)\|_1 \le 4\,\tau\,\eta_t\,\ln(\frac{A}{C}) + 2\,\sqrt{\tau}\,C^{\frac{1}{4}} + 2\,\eta_t\,H_\tau$$

Combining the above with the upper bound on $\epsilon_t$, we get,

$$\epsilon_t \le \frac{H_\tau \gamma^m}{(1 - \gamma)} \sum_{i=1}^{t} (\gamma^m)^{t-i} \left[ \left(1 + \tau\,(1-\gamma)\,\ln(\frac{A}{C})\right) \left(2\,\eta_i\,(2\,\tau\,\ln(\frac{A}{C}) + H_\tau) + 2\,\sqrt{\tau}\,C^{\frac{1}{4}}\right) \right.$$

$$\left. + \tau(1-\gamma)\left(\frac{\ln(A)}{2} + \sqrt{2}\right)\sqrt{C} \right]$$

Setting $C = \frac{1}{K^4}$,

$$\le \frac{H_\tau \gamma^m}{(1-\gamma)} \left[ 2\left(1 + \tau\,\ln\left(A\,K^4\right)\right) \left(\left(2\ln(A\,K^4) + \frac{H_\tau}{\tau}\right) \sum_{i=1}^{t} \frac{(\gamma^m)^{t-i}}{(i+1)} + \frac{\sqrt{\tau}}{K(1-\gamma^m)}\right) \right.$$

$$\left. + \frac{\tau}{1-\gamma^m}\left(\frac{\ln(A)}{2} + \sqrt{2}\right)\frac{1}{K^2} \right] \qquad \text{(since } \gamma < 1)$$

Using Lemma 3, we can bound $\sum_{i=1}^{t} \frac{(\gamma^m)^{t-i}}{(i+1)} \le \frac{2}{1-\gamma^m}(\frac{1}{t} + (\gamma^m)^{t/2}) \le \frac{2}{1-\gamma}(\frac{1}{t} + (\gamma^m)^{t/2})$.

$$\implies \epsilon_t \le \frac{H_\tau \gamma^m}{(1-\gamma)} \left[ 4\left(1 + \tau\,\ln\left(A\,K^4\right)\right) \right.$$

$$\left(\left(2\ln(A\,K^4) + \frac{H_\tau}{\tau}\right)\left(\frac{1}{t} + (\gamma^m)^{t/2}\right) + \frac{\sqrt{\tau}}{K(1-\gamma^m)}\right)$$

$$\left. + \frac{\tau}{1-\gamma^m}\left(\frac{\ln(A)}{2} + \sqrt{2}\right)\frac{1}{K^2} \right]$$

$$\le \frac{8H_\tau \gamma^m}{(1-\gamma)^2} \left[ \left(1 + \tau\,\ln\left(A\,K^4\right)\right)\left(\left(\ln(A\,K^4) + \frac{H_\tau}{\tau}\right)(\frac{1}{t} + (\gamma^m)^{t/2}) + \frac{\sqrt{\tau}}{K}\right) \right.$$

$$\left. + \tau\,(\ln(A)+1)\,\frac{1}{K^2} \right] \qquad \text{(Since } \gamma < 1)$$

$$\square$$

In the next corollary, we use Theorem 3 with $\zeta = 0$ to instantiate the policy error bound for (soft) NPG with entropy-regularized policy evaluation.

**Corollary 7** (Policy evaluation with $\zeta = 0$). *Using the policy evaluation update in Eq. (1) with $\zeta = 0$, $\epsilon_t := \|\epsilon_t\|_\infty$ can be bounded as:*

- ***Soft NPG:*** *If $\eta_t = \frac{1}{c + \tau\,(t+1)}$ for a constant $c \ge \max\left\{\frac{8\,(1+\tau\,\ln(A))}{(1-\gamma)}, 32\,\tau\,\ln(A)\right\}$, then, for all $t \in [K]$,*

$$\epsilon_t \le \frac{8(1 + \tau \ln(A))\gamma^m}{(1-\gamma)^3}\left[\left(1 + \tau\ln\left(A\,K^4\right)\right)\right.$$

$$\left(\left(\ln(A\,K^4) + \frac{1 + \tau\ln(A)}{\tau(1-\gamma)}\right)\left(\frac{1}{t} + (\gamma^m)^{t/2}\right) + \frac{\sqrt{\tau}}{K}\right)$$

$$\left. + \tau\,(\ln(A)+1)\,\frac{1}{K^2}\right] + \frac{\tau}{(1-\gamma)^2}\ln(A)$$

- **NPG**: *If $\eta_t = \eta = \frac{(1-\gamma)\sqrt{\ln(A)}}{\sqrt{K}}$, then, for all $t \in [K]$,*

$$\epsilon_t \leq \frac{2\sqrt{\ln(A)}\,\gamma^m}{(1-\gamma)^3}\,\frac{1}{\sqrt{K}}$$

*Proof.* Using Theorem 3 for soft NPG and Corollary 5 for NPG, and following the same proof as Corollary 6. $\qquad\square$

### C.2.3 POLICY ERROR BOUND FOR (SOFT) SPMA

In the next corollary, we use Theorem 3 with $\zeta = \tau$ to instantiate the policy error bound for soft SPMA with entropy-regularized policy evaluation.

**Corollary 8** (Policy evaluation with $\zeta = \tau$)**.** *Using the policy evaluation update in Eq. (1) with $\zeta = \tau$, for soft SPMA with $\eta_t = \frac{1}{c + \tau\,(t+1)}$ and a constant $c \geq \max\left\{\frac{4(1+\tau\,\ln(A))}{(1-\gamma)}, 32\,\tau\,\ln(A)\right\}$, for all $t \in [K]$, $\epsilon_t$ can be bounded as:*

$$\epsilon_t := \|\boldsymbol{\epsilon}_t\|_\infty \leq \frac{8(1+\tau\ln(A))\gamma^m}{(1-\gamma)^3}\left[\left(1+\tau\ln\left(A\,K^4\right)\right)\right.$$

$$\left(\left(\ln(A\,K^4) + \frac{1+\tau\ln(A)}{\tau(1-\gamma)}\right)\left(\frac{1}{t} + (\gamma^m)^{t/2}\right) + \frac{\sqrt{\tau}}{K}\right)$$

$$\left. + \tau\,(\ln(A)+1)\,\frac{1}{K^2}\right]$$

*Proof.* For a fixed iteration $t \in [K]$ and state $s \in \mathcal{S}$, let us first bound $\|\pi_{t+1}(\cdot|s) - \pi_t(\cdot|s)\|_1$.

$$\|\pi_{t+1}(\cdot|s) - \pi_t(\cdot|s)\|_1 \leq \left\|\pi_{t+1}(\cdot|s) - \pi_{t+1/2}(\cdot|s)\right\|_1 + \left\|\pi_{t+1/2}(\cdot|s) - \pi_t(\cdot|s)\right\|_1$$

$$\text{(Triangle inequality)}$$

$$\leq \left\|\pi_{t+1}(\cdot|s) - \pi_{t+1/2}(\cdot|s)\right\|_1 + \eta_t \sum_a \pi_t(a)\left[q_\tau^t(s,a) - v_\tau^t(s)\right]$$

$$\text{(By the SPMA update in Eq. (4))}$$

$$\implies \|\pi_{t+1}(\cdot|s) - \pi_t(\cdot|s)\|_1 \leq \left\|\pi_{t+1}(\cdot|s) - \pi_{t+1/2}(\cdot|s)\right\|_1 + \eta_t H_\tau$$

$$\text{(Since } |q_\tau^t(s,a) - v_\tau^t(s)| \leq H_\tau\text{)}$$

In order to bound $\left\|\pi_{t+1}(\cdot|s) - \pi_{t+1/2}(\cdot|s)\right\|_1$, we use the Eq. (5b) update. Specifically,

$$\pi_{t+1}(\cdot|s) = \arg\min_{\pi \in \Delta}\left[\mathrm{KL}(\pi\|\pi_{t+1/2}(\cdot|s)) - \tau_t\mathcal{H}(\pi(\cdot|s))\right]$$

$$\implies \mathrm{KL}(\pi_{t+1}(\cdot|s)\|\pi_{t+1/2}(\cdot|s)) - \tau_t\mathcal{H}(\pi_{t+1}(\cdot|s)) \leq \mathrm{KL}(\pi_{t+1/2}(\cdot|s)\|\pi_{t+1/2}(\cdot|s))$$

$$- \tau_t\mathcal{H}(\pi_{t+1/2}(\cdot|s))$$

$$\frac{1}{2}\left\|\pi_{t+1}(\cdot|s) - \pi_{t+1/2}(\cdot|s)\right\|_1^2 \leq \mathrm{KL}(\pi_{t+1}(\cdot|s)\|\pi_{t+1/2}(\cdot|s))$$

$$\text{(By Pinsker's inequality)}$$

$$\leq \tau_t\mathcal{H}(\pi_{t+1}(\cdot|s)) - \tau_t\mathcal{H}(\pi_{t+1/2}(\cdot|s))$$

$$\leq \tau_t\ln(A) \qquad \text{(Since } \mathcal{H}(\pi) \in [0, \ln(A)]\text{)}$$

$$\implies \left\|\pi_{t+1}(\cdot|s) - \pi_{t+1/2}(\cdot|s)\right\|_1 \leq \sqrt{2\,\tau\,\eta_t\,\ln(A)} \qquad \text{(Since } \tau_t = \tau\,\eta_t\text{)}$$

Combining the above relations,

$$\|\pi_{t+1}(\cdot|s) - \pi_t(\cdot|s)\|_1 \leq \sqrt{2\,\tau\,\eta_t\,\ln(A)} + \eta_t\,H_\tau \qquad (10)$$

In order to use Theorem 3, we need to ensure that $\|\pi_{t+1}(\cdot|s) - \pi_t(\cdot|s)\|_1 \leq \frac{1}{2}$ for all $t \in [K]$. Using Eq. (10), it is sufficient to ensure that $\eta_t$ satisfies the following relations:

- $\eta_t \, H_\tau \le \frac{1}{4} \implies \eta_t \le \frac{(1-\gamma)}{4\,(1+\tau\,\ln(A))}$
- $\sqrt{2\,\tau\,\eta_t\,\ln(A)} \le \frac{1}{4} \implies \eta_t \le \frac{1}{32\,\tau\,\ln(A)}$

For $\eta_t = \frac{1}{c+\tau(t+1)} \le \frac{1}{c}$, it is thus sufficient to ensure that,

$$c \ge \max\left\{\frac{4\,(1+\tau\,\ln(A))}{(1-\gamma)}, 32\,\tau\,\ln(A)\right\}$$

Given this constraint on $c$, we use Theorem 3 with $\zeta = \tau$ and $TV_i := \|\pi_i(\cdot|s) - \pi_{i-1}(\cdot|s)\|_1$, to get that, for any constant $C \in (0, 1/2)$, for all $t \in [K]$,

$$\epsilon_t \le \frac{H_\tau\,\gamma^m}{(1-\gamma)} \sum_{i=1}^{t} (\gamma^m)^{t-i} \max_s \left[TV_i + \tau(1-\gamma)\left(TV_i\,\ln\left(\frac{A}{C}\right) + \left(\frac{\ln(A)}{2} + \sqrt{2}\right)\sqrt{C}\right)\right]$$

Observe that the choice of $c$ ensures $\|\pi_{t+1}(\cdot|s) - \pi_t(\cdot|s)\|_1 \le \frac{1}{2}$. Using Lemma 2 we obtain,

$$\|\pi_{t+1}(\cdot|s) - \pi_t(\cdot|s)\|_1 \le 4\,\tau\,\eta_t\,\ln(\frac{A}{C}) + 2\,\sqrt{\tau}\,C^{\frac{1}{4}} + 2\,\eta_t\,H_\tau$$

Combining the above with the upper bound on $\epsilon_t$, we get,

$$\epsilon_t \le \frac{H_\tau\,\gamma^m}{(1-\gamma)} \sum_{i=1}^{t} (\gamma^m)^{t-i}\left[\left(1 + \tau\,(1-\gamma)\,\ln(\frac{A}{C})\right)\left(2\,\eta_i\,(2\,\tau\,\ln(\frac{A}{C}) + H_\tau) + 2\,\sqrt{\tau}\,C^{\frac{1}{4}}\right)\right.$$
$$\left. + \tau(1-\gamma)\left(\frac{\ln(A)}{2} + \sqrt{2}\right)\sqrt{C}\right]$$

Setting $C = \frac{1}{K^4}$,

$$\le \frac{H_\tau\,\gamma^m}{(1-\gamma)}\left[2\,(1 + \tau\,\ln(A\,K^4))\left(\left(2\ln(A\,K^4) + \frac{H_\tau}{\tau}\right)\sum_{i=1}^{t}\frac{(\gamma^m)^{t-i}}{(i+1)} + \frac{\sqrt{\tau}}{K(1-\gamma^m)}\right)\right.$$
$$\left. + \frac{\tau}{1-\gamma^m}\left(\frac{\ln(A)}{2} + \sqrt{2}\right)\frac{1}{K^2}\right] \qquad \text{(since } \gamma < 1\text{)}$$

Using Lemma 3, we can bound $\sum_{i=1}^{t}\frac{(\gamma^m)^{t-i}}{(i+1)} \le \frac{2}{1-\gamma^m}(\frac{1}{t} + (\gamma^m)^{t/2}) \le \frac{2}{1-\gamma}(\frac{1}{t} + (\gamma^m)^{t/2})$.

$$\implies \epsilon_t \le \frac{H_\tau\,\gamma^m}{(1-\gamma)}\left[4\,(1 + \tau\,\ln(A\,K^4))\right.$$
$$\left(\left(2\ln(A\,K^4) + \frac{H_\tau}{\tau}\right)\left(\frac{1}{t} + (\gamma^m)^{t/2}\right) + \frac{\sqrt{\tau}}{K(1-\gamma^m)}\right)$$
$$\left. + \frac{\tau}{1-\gamma^m}\left(\frac{\ln(A)}{2} + \sqrt{2}\right)\frac{1}{K^2}\right]$$
$$\le \frac{8H_\tau\,\gamma^m}{(1-\gamma)^2}\left[\left(1 + \tau\,\ln(A\,K^4)\right)\left(\left(\ln(A\,K^4) + \frac{H_\tau}{\tau}\right)(\frac{1}{t} + (\gamma^m)^{t/2}) + \frac{\sqrt{\tau}}{K}\right)\right.$$
$$\left. + \tau\,(\ln(A) + 1)\,\frac{1}{K^2}\right] \qquad \text{(Since } \gamma < 1\text{)}$$

$\square$

In the next corollary, we use Theorem 3 with $\zeta = 0$ to instantiate the policy error bound for (soft) SPMA without entropy-regularized policy evaluation.

**Corollary 9** (Policy evaluation with $\zeta = 0$). *Using the policy evaluation update in Eq.* (1) *with (soft)* SPMA, $\epsilon_t := \|\boldsymbol{\epsilon}_t\|_\infty$ *can be bounded as:*

- **Soft SPMA**: *if* $\eta_t = \frac{1}{c + \tau\,(t+1)}$ *for a constant* $c \geq \max\left\{ \frac{4(1+\tau\,\ln(A))}{(1-\gamma)}, 32\,\tau\,\ln(A) \right\}$, *then, for all* $t \in [K]$,

$$
\epsilon_t \leq \frac{8(1+\tau\ln(A))\gamma^m}{(1-\gamma)^3}\left[ \left(1 + \tau\ln\left(A\,K^4\right)\right) \right.
$$
$$
\left(\left(\ln(A\,K^4) + \frac{1+\tau\ln(A)}{\tau(1-\gamma)}\right)\left(\frac{1}{t} + (\gamma^m)^{t/2}\right) + \frac{\sqrt{\tau}}{K}\right)
$$
$$
\left. + \tau\,(\ln(A)+1)\,\frac{1}{K^2} \right] + \frac{\tau}{(1-\gamma)^2}\ln(A)
$$

- SPMA: *if* $\eta_t = \eta = \min\left\{ \frac{1-\gamma}{2}, \frac{(1-\gamma)\sqrt{\ln(A)}}{\sqrt{K}} \right\}$, *then, for all* $t \in [K]$,

$$
\epsilon_t := \|\boldsymbol{\epsilon}_t\|_\infty \leq \frac{\sqrt{\ln(A)}\,\gamma^m}{(1-\gamma)^3}\,\frac{1}{\sqrt{K}}
$$

*Proof.* Using Theorem 3 for soft SPMA and Corollary 5 for SPMA, and following the same proof as Corollary 8. Note that the requirement for $\eta \leq \frac{1-\gamma}{2}$ is to ensure $\ln(1 + \eta_t\,\Delta^t(s,\cdot))$ in the SPMA update is well-defined (see the proof in Corollary 11 for details) □

### C.3 PROOF OF COROLLARY 2

#### C.3.1 GENERIC REGRET BOUND

**Theorem 4** (Generic Regret Bound). *Consider a sequence of linear functions* $f_t(\pi) := \langle \pi, d_t \rangle$ *for a sequence of vectors* $\{d_0, d_1, \ldots, d_{K-1}\}$ *s.t.* $\|d_t\|_\infty \leq D_t$. *Consider the following update at iteration* $t \in [K]$, *if* $\eta_t$ *is a step-size sequence,* $\tau_t = \eta_t\,\tau$, $\pi_0$ *is the uniform distribution and*

$$
\pi_{t+1} = \underset{\pi \in \Delta_A}{\arg\min}\left\{ \langle \pi, d_t \rangle + KL(\pi\|\pi_t) + \mathcal{R}_t(\pi) \right\}
$$
$$
\mathcal{R}_t(\pi) := \tau_t\,\mathcal{R}(\pi)
$$
$$
\mathcal{R}(\pi) := \ln(A) - \mathcal{H}(\pi) \geq 0
$$

*then, for any comparator* $u \in \Delta_A$,

$$
\sum_{t=0}^{K-1}\left[ \frac{\langle \pi_t - u, d_t \rangle}{\eta_t} + \tau\,[\mathcal{H}(u) - \mathcal{H}(\pi_t)] \right] \leq \sum_{t=0}^{K-1}\left[ \frac{KL(u\|\pi_t)}{\eta_t} - \frac{KL(u\|\pi_{t+1})}{\eta_t} - \tau KL(u\|\pi_{t+1}) \right]
$$
$$
+ \sum_{t=0}^{K-1}\frac{D_t^2}{2\,\eta_t}
$$

*Proof.* The following properties will be helpful in proving the theorem. For policies $\pi, \pi'$ and comparator $u$,

$$
\mathcal{R}_t(\pi) - \mathcal{R}_t(\pi') = \tau_t\,\langle\ln(\pi), \pi - \pi'\rangle - \tau_t\,KL(\pi'\|\pi) \qquad \text{(Entropy property)}
$$
$$
\langle u - \pi', \ln(\pi') - \ln(\pi)\rangle = KL(u\|\pi) - KL(u\|\pi') - KL(\pi'\|\pi) \qquad \text{(3 point property)}
$$
$$
\langle \pi - \pi_{t+1}, d_t + \ln(\pi_{t+1}) - \ln(\pi_t) + \tau_t\ln(\pi_{t+1})\rangle \geq 0 \qquad \text{(Optimality condition)}
$$

$$
[f_t(\pi_t) - f_t(u)] + \mathcal{R}_t(\pi_{t+1}) - \mathcal{R}_t(u) = \langle \pi_t - u, d_t \rangle + \langle \tau_t\ln(\pi_{t+1}), \pi_{t+1} - u \rangle
$$
$$
- \tau_t KL(u\|\pi_{t+1})
$$
$$
\text{(Entropy property with } \pi = \pi_{t+1}, \pi' = u)
$$
$$
= \langle \pi_{t+1} - u, d_t \rangle + \langle \pi_t - \pi_{t+1}, d_t \rangle
$$

$$+ \langle \tau_t \ln(\pi_{t+1}),\, \pi_{t+1} - u \rangle - \tau_t \mathrm{KL}(u||\pi_{t+1})$$

$$= \underbrace{\langle \pi_{t+1} - u,\, d_t + \tau_t \ln(\pi_{t+1}) - \ln(\pi_t) + \ln(\pi_{t+1}) \rangle}_{\leq 0 \text{ by the optimality condition for } \pi = u}$$

$$+ \langle \pi_{t+1} - u,\, \ln(\pi_t) - \ln(\pi_{t+1}) \rangle + \langle \pi_t - \pi_{t+1},\, d_t \rangle$$

$$- \tau_t \mathrm{KL}(u||\pi_{t+1}) \qquad \text{(Dropping the negative term)}$$

$$\leq \langle \pi_{t+1} - u,\, \ln(\pi_t) - \ln(\pi_{t+1}) \rangle + \langle \pi_t - \pi_{t+1},\, d_t \rangle$$

$$- \tau_t \mathrm{KL}(u||\pi_{t+1})$$

$$= \mathrm{KL}(u||\pi_t) - \mathrm{KL}(u||\pi_{t+1}) - \mathrm{KL}(\pi_{t+1}||\pi_t)$$

$$+ \langle \pi_t - \pi_{t+1},\, d_t \rangle - \tau_t \mathrm{KL}(u||\pi_{t+1})$$
$$\text{(3 point property with } u = u, \pi = \pi_t, \pi' = \pi_{t+1})$$

$$\leq \mathrm{KL}(u||\pi_t) - \mathrm{KL}(u||\pi_{t+1}) - \mathrm{KL}(\pi_{t+1}||\pi_t)$$

$$- \tau_t \mathrm{KL}(u||\pi_{t+1}) + \frac{1}{2}||\pi_t - \pi_{t+1}||_1^2 + \frac{1}{2}||d_t||_\infty^2$$
$$\text{(Fenchel-Young inequality)}$$

$$\leq \mathrm{KL}(u||\pi_t) - \mathrm{KL}(u||\pi_{t+1}) - \mathrm{KL}(\pi_{t+1}||\pi_t)$$

$$- \tau_t \mathrm{KL}(u||\pi_{t+1}) + \mathrm{KL}(\pi_{t+1}||\pi_t) + \frac{1}{2}||d_t||_\infty^2$$
$$\text{(Pinsker's inequality)}$$

$$= \mathrm{KL}(u||\pi_t) - \mathrm{KL}(u||\pi_{t+1}) - \tau_t \mathrm{KL}(u||\pi_{t+1}) + \frac{1}{2}||d_t||_\infty^2$$

$$\leq \mathrm{KL}(u||\pi_t) - \mathrm{KL}(u||\pi_{t+1}) - \tau_t \mathrm{KL}(u||\pi_{t+1}) + \frac{D_t^2}{2}$$
$$\text{(Since } ||d_t||_\infty \leq D_t)$$

$$= \mathrm{KL}(u||\pi_t) - \mathrm{KL}(u||\pi_{t+1}) - \eta_t \tau \mathrm{KL}(u||\pi_{t+1}) + \frac{D_t^2}{2}$$
$$\text{(Since } \tau_t = \eta_t \tau)$$

Rearranging and dividing both-sides by $\eta_t$ we get

$$\frac{\langle \pi_t - u,\, d_t \rangle}{\eta_t} + \mathcal{R}(\pi_{t+1}) - \mathcal{R}(u) \leq \frac{\mathrm{KL}(u||\pi_t)}{\eta_t} - \frac{\mathrm{KL}(u||\pi_{t+1})}{\eta_t} - \tau \mathrm{KL}(u||\pi_{t+1}) + \frac{D_t^2}{2\,\eta_t}$$

$$\frac{\langle \pi_t - u,\, d_t \rangle}{\eta_t} + \mathcal{R}(\pi_t) - \mathcal{R}(u) \leq [\mathcal{R}(\pi_t) - \mathcal{R}(\pi_{t+1})] + \frac{\mathrm{KL}(u||\pi_t)}{\eta_t} - \frac{\mathrm{KL}(u||\pi_{t+1})}{\eta_t}$$

$$- \tau \mathrm{KL}(u||\pi_{t+1}) + \frac{D_t^2}{2\,\eta_t}$$

Summing from $t = 0$ to $K - 1$,

$$\sum_{t=0}^{K-1} \left[ \frac{\langle \pi_t - u,\, d_t \rangle}{\eta_t} + \mathcal{R}(\pi_t) - \mathcal{R}(u) \right] \leq [\mathcal{R}(\pi_0) - \mathcal{R}(\pi_K)] + \sum_{t=0}^{K-1} \left[ \frac{\mathrm{KL}(u||\pi_t)}{\eta_t} - \frac{\mathrm{KL}(u||\pi_{t+1})}{\eta_t} \right.$$

$$\left. - \tau \mathrm{KL}(u||\pi_{t+1}) + \frac{D_t^2}{2\,\eta_t} \right]$$

$$\leq \sum_{t=0}^{K-1} \left[ \frac{\mathrm{KL}(u||\pi_t)}{\eta_t} - \frac{\mathrm{KL}(u||\pi_{t+1})}{\eta_t} - \tau \mathrm{KL}(u||\pi_{t+1}) \right]$$

$$+ \sum_{t=0}^{K-1} \frac{D_t^2}{2\,\eta_t}$$
$$\text{(Since } \mathcal{R}(\pi_0) = \ln(A) - \mathcal{H}(\pi_0) = 0 \text{ and } \mathcal{R}(\pi_K) \geq 0)$$

$$\square$$

### C.3.2 Regret Bound for (soft) NPG

In this section, we instantiate the regret and policy evaluation bounds for (soft) NPG

**Corollary 10** (Regret Bounds)**.** *Suppose $\pi_0(\cdot|s)$ is the uniform distribution over actions for each state $s$. For any sequence $\{q_\zeta^t\}_{t=0}^{K-1}$ satisfying $\left\|q_\zeta^t\right\|_\infty \le H_\tau$, the regret for (soft) NPG can be bounded as:*

- ***Soft NPG:** Setting $\eta_t = \frac{1}{c + \tau\,(t+1)}$ for a constant $c \ge 0$ to be determined later, guarantees that,*

$$\max_s \left| \sum_{t=0}^{K-1} \left[ \langle \pi_\tau^*(\cdot|s) - \pi_t(\cdot|s),\, q_\zeta^t(s, \cdot) \rangle + \tau \left[ \mathcal{H}(\pi_\tau^*(\cdot|s)) - \mathcal{H}(\pi_t(\cdot|s)) \right] \right] \right|$$

$$\le \frac{H_\tau^2}{2\,\tau} \left[ 1 + \ln(K) \right] + (c + \tau) \ln(A)$$

- ***NPG:** Setting $\eta_t = \eta = \frac{\sqrt{2}\,(1-\gamma)\,\sqrt{\ln(A)}}{\sqrt{K}}$ guarantees that,*

$$\max_s \left| \sum_{t=0}^{K-1} \left[ \langle \pi^*(\cdot|s) - \pi_t(\cdot|s),\, q_\zeta^t(s, \cdot) \rangle \right] \right| \le \frac{\sqrt{2\ln(A)}\,\sqrt{K}}{1 - \gamma}$$

*Proof.* First note that by using the mirror descent view of the (soft) NPG update (Xiao, 2022), it can be equivalently written as: for all $s \in \mathcal{S}$,

$$\pi_{t+1}(\cdot|s) = \underset{\pi \in \Delta}{\arg\min} \left[ -\eta_t \langle q_\zeta^t(s, \cdot), \pi(\cdot|s) \rangle + \mathrm{KL}(\pi(\cdot|s)||\pi_t(\cdot|s)) - \tau_t \mathcal{H}(\pi(\cdot|s)) \right]$$

By comparing to the update in Theorem 4, we note that $d_t = -\eta_t\,q_\zeta^t(s, \cdot)$ and $\|d_t\|_\infty = \eta_t \left\|q_\zeta^t\right\|_\infty \le \eta_t\,H_\tau$. If $\tau_t = \eta_t\tau$ and $\pi_0(\cdot|s)$ is a uniform distribution for each state $s$, we can instantiate Theorem 4 for each state $s$, and obtain the following regret bound for the comparator $u$.

$$\sum_{t=0}^{K-1} \left[ \langle u(\cdot|s) - \pi_t(\cdot|s),\, q_\zeta^t(s, \cdot) \rangle + \tau \left[ \mathcal{H}(u(\cdot|s)) - \mathcal{H}(\pi_t(\cdot|s)) \right] \right]$$

$$\le \sum_{t=0}^{K-1} \left[ \frac{\mathrm{KL}(u(\cdot|s)||\pi_t(\cdot|s))}{\eta_t} - \frac{\mathrm{KL}(u(\cdot|s)||\pi_{t+1}(\cdot|s))}{\eta_t} - \tau\mathrm{KL}(u(\cdot|s)||\pi_{t+1}(\cdot|s)) \right]$$

$$+ \frac{H_\tau^2}{2} \sum_{t=0}^{K-1} \eta_t$$

Now we consider two cases corresponding to NPG and its soft variant.

**Soft NPG:** Using that $\tau \ne 0$, setting $u = \pi_\tau^*$ and bounding the RHS in the above inequality,

$$\sum_{t=0}^{K-1} \left[ \langle \pi_\tau^*(\cdot|s) - \pi_t(\cdot|s),\, q_\zeta^t(s, \cdot) \rangle + \tau \left[ \mathcal{H}(\pi_\tau^*(\cdot|s)) - \mathcal{H}(\pi_t(\cdot|s)) \right] \right]$$

$$\le \sum_{t=1}^{K-1} \mathrm{KL}(\pi_\tau^*(\cdot|s)||\pi_t(\cdot|s)) \left[ \frac{1}{\eta_t} - \frac{1}{\eta_{t-1}} - \tau \right] + \frac{1}{\eta_0} \mathrm{KL}(\pi_\tau^*(\cdot|s)||\pi_0(\cdot|s)) + \frac{H_\tau^2}{2} \sum_{t=0}^{K-1} \eta_t$$

$$= \frac{H_\tau^2}{2} \sum_{t=0}^{K-1} \frac{1}{c + \tau\,(t+1)} + (c + \tau) \mathrm{KL}(\pi_\tau^*(\cdot|s)||\pi_0(\cdot|s)) \qquad \text{(Setting } \eta_t = \frac{1}{c + \tau\,(t+1)} \text{)}$$

$$\le \frac{H_\tau^2}{2} \sum_{t=0}^{K-1} \frac{1}{\tau\,(t+1)} + (c + \tau) \ln(A) \quad \text{(Since } \pi_0(\cdot|s) \text{ is a uniform distribution for all } s\text{)}$$

$$\le \frac{H_\tau^2}{2\,\tau} \left[ 1 + \ln(K) \right] + (c + \tau) \ln(A) \qquad \text{(Since } \sum_{t=1}^{K} 1/t \le 1 + \ln(K)\text{)}$$

Since the above bound holds for all $s$,

$$\max_s \left| \sum_{t=0}^{K-1} \left[ \langle \pi^*_\tau(\cdot|s) - \pi_t(\cdot|s), q^t_\zeta(s,\cdot) \rangle + \tau \left[ \mathcal{H}(\pi^*_\tau(\cdot|s)) - \mathcal{H}(\pi_t(\cdot|s)) \right] \right] \right|$$

$$\leq \frac{H_\tau^2}{2\,\tau} \left[ 1 + \ln(K) \right] + (c + \tau)\,\ln(A)$$

**NPG**: Using $u = \pi^*$ and a constant step-size i.e. $\eta_t = \eta$ for all $t$, in which case the regret bound can be simplified as:

$$\sum_{t=0}^{K-1} \left[ \langle \pi^*(\cdot|s) - \pi_t(\cdot|s), q^t_\zeta(s,\cdot) \rangle \right] \leq \frac{1}{\eta} \mathrm{KL}(\pi^*(\cdot|s)||\pi_0(\cdot|s) + \frac{\eta K}{2\,(1-\gamma)^2}$$

$$\leq \frac{\ln(A)}{\eta} + \frac{\eta K}{2\,(1-\gamma)^2}$$

$$\text{(Since } \pi_0 \text{ is the uniform distribution)}$$

$$\leq \frac{\sqrt{2\ln(A)}\,\sqrt{K}}{1-\gamma} \qquad \text{(Setting } \eta = \frac{\sqrt{2}\,(1-\gamma)\,\sqrt{\ln(A)}}{\sqrt{K}} \text{)}$$

Since the above bound holds for all $s$,

$$\max_s \left| \sum_{t=0}^{K-1} \left[ \langle \pi^*(\cdot|s) - \pi_t(\cdot|s), q^t_\zeta(s,\cdot) \rangle \right] \right| \leq \frac{\sqrt{2\ln(A)}\,\sqrt{K}}{1-\gamma}.$$

$$\square$$

### C.3.3 Regret Bound for (soft) SPMA

In this section, we instantiate the regret and policy evaluation bounds for (soft) SPMA.

**Corollary 11** (Regret Bounds). *Suppose $\pi_0(\cdot|s)$ is the uniform distribution over actions for each state $s$, and let $\eta_t = \frac{1}{c+\tau\,(t+1)}$ for some constant $c \geq 0$ to be determined later. For any sequence $\{q^t_\zeta\}_{t=0}^{K-1}$ satisfying $\left\| q^t_\zeta \right\|_\infty \leq H_\tau$, the regret for (soft) SPMA can be bounded as:*

- ***Soft SPMA**: Setting $\eta_t = \frac{1}{c+\tau\,(t+1)}$ for a constant $c \geq 2\max\{H_\tau, \zeta\ln(A)\}$, guarantees that,*

$$\max_s \left| \sum_{t=0}^{K-1} \left[ \langle \pi^*_\tau(\cdot|s) - \pi_t(\cdot|s), q^t_\zeta(s,\cdot) \rangle + \tau \left[ \mathcal{H}(\pi^*_\tau(\cdot|s)) - \mathcal{H}(\pi_t(\cdot|s)) \right] \right] \right|$$

$$\leq \frac{3\,H_\tau^2}{\tau} \left[ 1 + \ln(K) \right] + (c+\tau)\,\ln(A)$$

- ***SPMA**: Setting $\eta_t = \eta = \min \left\{ \frac{1-\gamma}{2}, \frac{\sqrt{2}\,(1-\gamma)\,\sqrt{\ln(A)}}{\sqrt{K}} \right\}$ guarantees that,*

$$\max_s \left| \sum_{t=0}^{K-1} \left[ \langle \pi^*(\cdot|s) - \pi_t(\cdot|s), q^t_\zeta(s,\cdot) \rangle \right] \right| \leq \frac{7\,\sqrt{\ln(A)}\,\sqrt{K}}{\sqrt{2}\,(1-\gamma)} + \frac{2}{1-\gamma}\,\ln(A).$$

*Proof.* For a fixed state $s \in \mathcal{S}$, first note that the (soft) SPMA update in Eq. (5b) can be equivalently be written as follows: if $\Delta^t(s,a) := q^t_\zeta(s,a) - v^t_\zeta(s)$ for $d_t := -\ln(1 + \eta_t\,\Delta^t(s,\cdot))$,

$$\pi_{t+1}(\cdot|s) = \underset{\pi \in \Delta}{\arg\min} \left[ \mathrm{KL}(\pi(\cdot|s)||\pi_t(\cdot|s) \left[ 1 + \eta_t\,\left( \Delta^t(s,\cdot) \right) \right]) - \tau_t \mathcal{H}(\pi(\cdot|s)) \right]$$

$$= \underset{\pi \in \Delta}{\arg\min} \left[ \langle d_t, \pi(\cdot|s) \rangle + \mathrm{KL}(\pi(\cdot|s)||\pi_t(\cdot|s)) - \tau_t \mathcal{H}(\pi(\cdot|s)) \right],$$

where we require that $1 + \eta_t \Delta^t(s,\cdot) \geq 0$. Note that since $\left\| q^t_\zeta(s,\cdot) \right\|_\infty \leq H_\tau$, $\|\Delta^t(s,\cdot))\|_\infty \leq H_\tau$ and we require that $\eta_t \leq \frac{1}{2H_\tau}$. With this choice, $|\eta_t\,\Delta^t(s,a)| \leq \frac{1}{2}$. By comparing to the

update in Theorem 4, we note that $d_t = -\ln(1 + \eta_t \Delta^t(s, \cdot))$. Since $|\ln(1+x)| \leq 2|x|$ for all $x \geq -\frac{1}{2}$,

$$|-\ln(1 + \eta_t \Delta^t(s, a))| \leq 2\,\eta_t\,|\Delta^t(s,a)| \leq 2\,\eta_t\,H_\tau \implies \|d_t\|_\infty \leq 2\,\eta_t\,H_\tau$$

Hence, $D_t = 2\,\eta_t\,H_\tau$. If $\tau_t = \eta_t \tau$ and $\pi_0(\cdot|s)$ is a uniform distribution for each state $s$, we can instantiate Theorem 4 for each state $s$, and obtain the following regret bound for the comparator $u$,

$$\sum_{t=0}^{K-1} \left[ \frac{\langle u(\cdot|s) - \pi_t(\cdot|s), \ln(1 + \eta_t \Delta^t(s, \cdot))\rangle}{\eta_t} + \tau\left[\mathcal{H}(u(\cdot|s)) - \mathcal{H}(\pi_t(\cdot|s))\right] \right]$$

$$\leq \sum_{t=0}^{K-1} \left[ \frac{KL(u(\cdot|s)||\pi_t(\cdot|s))}{\eta_t} - \frac{KL(u(\cdot|s)||\pi_{t+1}(\cdot|s))}{\eta_t} - \tau KL(u(\cdot|s)||\pi_{t+1}(\cdot|s)) \right]$$

$$+ 2H_\tau^2 \sum_{t=0}^{K-1} \eta_t$$

In order to simplify the above expression, first note that,

$$\langle \pi_t(\cdot|s),\, \ln(1 + \eta_t \Delta^t(s, \cdot))\rangle \leq \ln(1 - \eta_t \zeta\,\mathcal{H}(\pi_t(\cdot|s)))$$
$$\text{(Using Jensen's inequality and the fact } \sum_a \pi_t(a|s)\Delta^t(s,a) = -\zeta\,\mathcal{H}(\pi_t(\cdot|s)))$$

If $\eta_t$ is chosen such that $\eta_t\,\zeta\,\mathcal{H}(\pi_t(\cdot|s)) \leq \frac{1}{2}$, and since $\mathcal{H}(\pi) \in [0, \ln(A)]$, it suffices to ensure $\eta_t \leq \frac{1}{2\zeta\,\ln(A)}$, and use the fact $\ln(1+x) \leq x$ for $x > -1$ to guarantee:

$$\langle \pi_t(\cdot|s),\, \ln(1 + \eta_t \Delta^t(s, \cdot))\rangle \leq -\eta_t\,\zeta\,\mathcal{H}(\pi_t(\cdot|s)) \tag{11}$$

On the other hand, since choosing $\eta_t \leq \frac{1}{2H_\tau}$ ensures $\ln(1 + \eta_t \Delta^t(s, \cdot))$ is well-defined,

$$\langle u(\cdot|s),\, \ln(1 + \eta_t \Delta^t(s, \cdot))\rangle \geq [\langle u(\cdot|s),\, \eta_t \Delta^t(s, \cdot)\rangle - \langle u(\cdot|s),\, \eta_t^2 [\Delta^t(s, \cdot)]^2\rangle]$$
$$\text{(since } \ln(1+x) \geq x - x^2 \text{ for } x > -1/2)$$
$$= \eta_t \left(\langle u(\cdot|s),\, q_\zeta^t(s, \cdot)\rangle - v_\zeta^t(s)\right) - \eta_t^2 \langle u(\cdot|s),\, [\Delta^t(s, \cdot)]^2\rangle$$
$$= \eta_t \left(\langle u(\cdot|s),\, q_\zeta^t(s, \cdot)\rangle - \langle \pi_t,\, q_\zeta^t(s, \cdot)\rangle - \zeta\mathcal{H}(\pi_t(\cdot|s))\right)$$
$$- \eta_t^2 \langle u(\cdot|s),\, [\Delta^t(s, \cdot)]^2\rangle$$
$$\text{(since } v_\zeta^t(s) = \langle \pi_t(\cdot|s),\, q_\zeta^t\rangle + \zeta\mathcal{H}(\pi_t(\cdot|s)))$$
$$\geq \eta_t \left(\langle u(\cdot|s),\, q_\zeta^t(s, \cdot)\rangle - \langle \pi_t(\cdot|s),\, q_\zeta^t(s, \cdot)\rangle - \zeta\mathcal{H}(\pi_t(\cdot|s))\right) - \eta_t^2 H_\tau^2$$
$$\text{(since } \|\Delta^t(s, \cdot)\|_\infty \leq H_\tau)$$

Combining the above inequalities with Eq. (11),

$$\frac{\langle u(\cdot|s) - \pi_t(\cdot|s),\, \ln(1 + \eta_t \Delta^t(s, \cdot))\rangle}{\eta_t} \geq \langle u(\cdot|s),\, q_\zeta^t(s, \cdot)\rangle - \langle \pi_t,\, q_\zeta^t(s, \cdot)\rangle - \eta_t\,H_\tau^2$$
$$= \langle u(\cdot|s) - \pi_t(\cdot|s),\, q_\zeta^t(s, \cdot)\rangle - \eta_t H_\tau^2$$

Using the above relation with the regret expression,

$$\sum_{t=0}^{K-1} \left[ \langle u(\cdot|s) - \pi_t(\cdot|s),\, q_\zeta^t(s, \cdot)\rangle + \tau\left[\mathcal{H}(u(\cdot|s)) - \mathcal{H}(\pi_t(\cdot|s))\right] \right]$$

$$\leq \sum_{t=0}^{K-1} \left[ \frac{KL(u(\cdot|s)||\pi_t(\cdot|s))}{\eta_t} - \frac{KL(u(\cdot|s)||\pi_{t+1}(\cdot|s))}{\eta_t} - \tau KL(u(\cdot|s)||\pi_{t+1}(\cdot|s)) \right]$$

$$+ 3\,H_\tau^2 \sum_{t=0}^{K-1} \eta_t$$

Note that we require $\eta_t \leq \frac{1}{2H_\tau}$ and $\eta_t \leq \frac{1}{2\zeta \ln(A)}$ simultaneously for all $t$. Hence, it is sufficient to ensure that $\eta_t \leq \frac{1}{2\max\{H_\tau, \zeta \ln(A)\}}$ for all $t$, and hence require that $c \geq 2\max\{H_\tau, \zeta \ln(A)\}$. Now we consider two cases corresponding to SPMA and its soft variant.

**Soft SPMA**: Using that $\tau \neq 0$, setting $u = \pi_\tau^*$ and bounding the RHS in the above inequality,

$$\sum_{t=0}^{K-1} \left[ \langle \pi_\tau^*(\cdot|s) - \pi_t(\cdot|s), q_\zeta^t(s,\cdot) \rangle + \tau \left[ \mathcal{H}(\pi_\tau^*(\cdot|s)) - \mathcal{H}(\pi_t(\cdot|s)) \right] \right]$$

$$\leq \sum_{t=1}^{K-1} \mathrm{KL}(\pi_\tau^*(\cdot|s) \| \pi_t(\cdot|s)) \left[ \frac{1}{\eta_t} - \frac{1}{\eta_{t-1}} - \tau \right] + \frac{1}{\eta_0} \mathrm{KL}(\pi_\tau^*(\cdot|s) \| \pi_0(\cdot|s))$$

$$+ 3 H_\tau^2 \sum_{t=0}^{K-1} \eta_t$$

$$= 3 H_\tau^2 \sum_{t=0}^{K-1} \frac{1}{c + \tau(t+1)} + (c + \tau) \mathrm{KL}(\pi_\tau^*(\cdot|s) \| \pi_0(\cdot|s)) \qquad \text{(Since } \eta_t = \frac{1}{c+\tau(t+1)}\text{)}$$

$$\leq 3 H_\tau^2 \sum_{t=0}^{K-1} \frac{1}{c + \tau(t+1)} + (c + \tau) \ln(A)$$

$$\text{(Since } \pi_0(\cdot|s) \text{ is a uniform distribution for all } s\text{)}$$

$$\leq \frac{3 H_\tau^2}{\tau} \sum_{t=0}^{K-1} \frac{1}{t+1} + (c + \tau) \ln(A)$$

$$\leq \frac{3 H_\tau^2}{\tau} \left[ 1 + \ln(K) \right] + (c + \tau) \ln(A) \qquad \text{(Since } \sum_{t=1}^{K} 1/t \leq 1 + \ln(K)\text{)}$$

Since the above bound holds for all $s$,

$$\max_s \left| \sum_{t=0}^{K-1} \left[ \langle \pi_\tau^*(\cdot|s) - \pi_t(\cdot|s), q_\zeta^t(s,\cdot) \rangle + \tau \left[ \mathcal{H}(\pi_\tau^*(\cdot|s)) - \mathcal{H}(\pi_t(\cdot|s)) \right] \right] \right|$$

$$\leq \frac{3 H_\tau^2}{\tau} \left[ 1 + \ln(K) \right] + (c + \tau) \ln(A)$$

**SPMA**: Using $u = \pi^*$, $\tau = \zeta = 0$, and a constant step-size i.e. $\eta_t = \eta$ for all $t$, in which case the regret bound can be simplified as:

$$\sum_{t=0}^{K-1} \left[ \langle \pi^*(\cdot|s) - \pi_t(\cdot|s), q_\zeta^t(s,\cdot) \rangle \right] \leq \frac{1}{\eta} \mathrm{KL}(\pi^*(\cdot|s) \| \pi_0(\cdot|s)) + \frac{3 \eta K}{(1-\gamma)^2}$$

$$\leq \frac{\ln(A)}{\eta} + \frac{3 \eta K}{(1-\gamma)^2}$$

$$\text{(Since } \pi_0(\cdot|s) \text{ is a uniform distribution for all } s\text{)}$$

Recall that in the presence of entropy ensuring $\ln(1 + \eta_t \Delta^t(s,\cdot))$ is well-defined required us to choose $\eta_t \leq \frac{1}{2H_\tau}$. When $\tau = 0$ and $\eta_t = \eta$, the condition simplifies to $\eta \leq \frac{1-\gamma}{2}$. Setting $\eta = \min\left\{ \frac{\sqrt{2}\,(1-\gamma)\,\sqrt{\ln(A)}}{\sqrt{K}}, \frac{1-\gamma}{2} \right\}$ and using the fact that $\frac{1}{\min\{a,b\}} = \max\{1/a, 1/b\}$

$$\leq \ln(A) \max\left\{ \frac{\sqrt{K}}{\sqrt{2}\,(1-\gamma)\,\sqrt{\ln(A)}}, \frac{2}{1-\gamma} \right\}$$

$$+ \frac{3 K}{(1-\gamma)^2} \min\left\{ \frac{\sqrt{2}\,(1-\gamma)\,\sqrt{\ln(A)}}{\sqrt{K}}, \frac{1-\gamma}{2} \right\}$$

$$\leq \frac{7\sqrt{\ln(A)}\sqrt{K}}{\sqrt{2}\,(1-\gamma)} + \frac{2}{1-\gamma}\,\ln(A)$$

$$(\text{Since } \max\{a,b\} \leq a+b, \ \min\{a,b\} \leq a)$$

Since the above bound holds for all $s$,

$$\max_s \left| \sum_{t=0}^{K-1} \left[ \langle \pi^*(\cdot|s) - \pi_t(\cdot|s),\, q_\zeta^t(s,\cdot) \rangle \right] \right| \leq \frac{7\sqrt{\ln(A)}\sqrt{K}}{\sqrt{2}\,(1-\gamma)} + \frac{2}{1-\gamma}\,\ln(A).$$

$\square$

### C.4  Proof of Theorem 2

Finally, we put everything together, and in the following two subsections, we prove theorems that quantify the performance of (soft) NPG and (soft) SPMA when using the hard or soft Bellman operator (i.e., $\zeta = \tau$ or $\zeta = 0$) as well as the case of no entropy regularization (i.e, $\tau = 0$ and $\zeta = 0$).

#### C.4.1  Putting everything together for soft NPG

**Theorem 5** (Sub-optimality of soft NPG). *Let $\pi_\tau^*$ denote the optimal entropy-regularized policy with value function $v_\tau^*$. Consider the soft NPG update with step size $\eta_t = \frac{1}{c+\tau\,(t+1)}$, $c \geq \max\left\{ \frac{8\,(1+\tau\,\ln(A))}{(1-\gamma)}, 32\,\tau\,\ln(A) \right\}$ and, $\delta(\tau,\zeta) := \frac{|\tau-\zeta|\,\ln(A)}{1-\gamma}$. Let $\pi_0(\cdot|s)$ be the uniform policy over actions for all $s \in \mathcal{S}$ and assume the policy evaluation step in Eq. (1). Then the resulting mixture policy $\bar\pi_K$ satisfies the following sub-optimality bound,*

$$\left\| v_\tau^{\bar\pi_K} - v_\tau^* \right\|_\infty \leq \frac{1}{K\,(1-\gamma)} \left[ \frac{(1+\tau\,\ln(A))^2}{2\,\tau\,(1-\gamma)^2} \left[1 + \ln(K)\right] + (c+\tau)\,\ln(A) \right]$$

$$+ \frac{16(1+\tau\ln(A))\gamma^m}{(1-\gamma)^4\,K} \left[ \left(1 + \tau\ln\left(A\,K^4\right)\right) \right.$$

$$\left( \left(\ln(A\,K^4) + \frac{1+\tau\ln(A)}{\tau(1-\gamma)}\right) \left(1 + \ln(K) + \frac{1}{1-\sqrt{\gamma}}\right) \right.$$

$$\left. + \sqrt{\tau} \right)$$

$$\left. + \tau\,(\ln(A)+1)\,\frac{1}{K} \right] + \frac{2\,\delta(\tau,\zeta)}{(1-\gamma)^2}$$

*Proof.* Plugging the regret bound in Corollary 10 for soft NPG into the regret part of Theorem 1 immediately gives the first part of the upper-bound:

$$\frac{\|\text{Regret}(K)\|_\infty}{K\,(1-\gamma)} \leq \frac{1}{K\,(1-\gamma)} \left[ \frac{(1+\tau\,\ln(A))^2}{2\,\tau\,(1-\gamma)^2} \left[1 + \ln(K)\right] + (c+\tau)\,\ln(A) \right]$$

Using the result from Theorem 3 and Corollary 6 to upper-bound the error part $E_K := \frac{2\sum_{k\in[K]}\|\epsilon_k\|_\infty}{K\,(1-\gamma)}$ in Theorem 1 we obtain:

$$E_K \leq \frac{16(1+\tau\ln(A))\gamma^m}{(1-\gamma)^4\,K} \left[ \left(1 + \tau\ln\left(A\,K^4\right)\right) \right.$$

$$\left( \left(\ln(A\,K^4) + \frac{1+\tau\ln(A)}{\tau(1-\gamma)}\right) \left(1 + \ln(K) + \frac{1}{1-\sqrt{\gamma}}\right) + \sqrt{\tau} \right)$$

$$\left. + \tau\,(\ln(A)+1)\,\frac{1}{K} \right] + \frac{2\,\delta(\tau,\zeta)}{(1-\gamma)^2}$$

Where the above inequality can be obtained using the fact $\sum_{t=1}^{K} \frac{1}{t} \leq 1 + \ln(K)$ (using integration) and $\sum_{t=1}^{K}(\gamma^m)^{t/2} \leq \frac{1}{1-\gamma^{m/2}} \leq \frac{1}{1-\sqrt{\gamma}}$

$\square$

**Corollary 12** (Sub-optimality of soft `NPG` with critic entropy)**.** *Let $\pi_\tau^*$ denote the optimal entropy-regularized policy with value function $v_\tau^*$. Consider the soft `NPG` update with step size $\eta_t = \frac{1}{c+\tau\,(t+1)}$ and $c \geq \max\left\{\frac{8\,(1+\tau\,\ln(A))}{(1-\gamma)}, 32\,\tau\,\ln(A)\right\}$. Let $\pi_0(\cdot|s)$ be the uniform policy over actions for all $s \in \mathcal{S}$ and assume the policy evaluation step in Eq. (1) with $\zeta = \tau$. Then the resulting mixture policy $\bar{\pi}_K$ satisfies the following sub-optimality bound,*

$$
\left\|v_\tau^{\bar{\pi}_K} - v_\tau^*\right\|_\infty \leq \frac{1}{K\,(1-\gamma)}\left[\frac{(1+\tau\,\ln(A))^2}{2\,\tau\,(1-\gamma)^2}\left[1+\ln(K)\right] + (c+\tau)\,\ln(A)\right]
$$
$$
+ \frac{16(1+\tau\ln(A))\gamma^m}{(1-\gamma)^4\,K}\left[\left(1+\tau\ln\left(A\,K^4\right)\right)\right.
$$
$$
\left(\left(\ln(A\,K^4)+\frac{1+\tau\ln(A)}{\tau(1-\gamma)}\right)\left(1+\ln(K)+\frac{1}{1-\sqrt{\gamma}}\right)\right.
$$
$$
\left.+\sqrt{\tau}\right)
$$
$$
\left.+\tau\,(\ln(A)+1)\,\frac{1}{K}\right]
$$

*Proof.* Using Theorem 1 with Corollary 10 for soft `NPG`, Theorem 3, Corollary 6 and the result from Theorem 5. $\qquad\square$

**Corollary 13** (Sub-optimality of soft `NPG` without critic entropy)**.** *Let $\pi_\tau^*$ denote the optimal entropy-regularized policy with value function $v_\tau^*$. Consider the soft `NPG` update with step size $\eta_t = \frac{1}{c+\tau\,(t+1)}$ and $c \geq \max\left\{\frac{8\,(1+\tau\,\ln(A))}{(1-\gamma)}, 32\,\tau\,\ln(A)\right\}$. Let $\pi_0(\cdot|s)$ be the uniform policy over actions for all $s \in \mathcal{S}$ and assume the policy evaluation step in Eq. (1) with $\zeta = 0$. Then the resulting mixture policy $\bar{\pi}_K$ satisfies the following sub-optimality bound,*

$$
\left\|v_\tau^{\bar{\pi}_K} - v_\tau^*\right\|_\infty \leq \frac{1}{K\,(1-\gamma)}\left[\frac{(1+\tau\,\ln(A))^2}{2\,\tau\,(1-\gamma)^2}\left[1+\ln(K)\right] + (c+\tau)\,\ln(A)\right]
$$
$$
+ \frac{16(1+\tau\ln(A))\gamma^m}{(1-\gamma)^4\,K}\left[\left(1+\tau\ln\left(A\,K^4\right)\right)\right.
$$
$$
\left(\left(\ln(A\,K^4)+\frac{1+\tau\ln(A)}{\tau(1-\gamma)}\right)\left(1+\ln(K)+\frac{1}{1-\sqrt{\gamma}}\right)\right.
$$
$$
\left.+\sqrt{\tau}\right)
$$
$$
\left.+\tau\,(\ln(A)+1)\,\frac{1}{K}\right] + \frac{2\,\tau\,\ln(A)}{(1-\gamma)^3}
$$

*Proof.* Using Theorem 1 with Corollary 10 for soft `NPG`, Theorem 3, Corollary 7 and the result from Theorem 5. $\qquad\square$

**Theorem 6** (`NPG` + policy evaluation without entropy regularization)**.** *If $\pi^*$ is the optimal policy whose value function is equal to $v^*$, the `NPG` update with $\eta_t = \eta = \frac{\sqrt{2}\,(1-\gamma)\,\sqrt{\ln(A)}}{\sqrt{K}}$, $\pi_0(\cdot|s)$ as the uniform initial policy for each $s \in \mathcal{S}$ with the policy evaluation procedure in Eq. (1) with $\zeta = 0$ satisfies the following sub-optimality bound for the mixture policy $\bar{\pi}_K$,*

$$
\left\|v^{\bar{\pi}_K} - v^*\right\|_\infty \leq \frac{\sqrt{2\ln(A)}}{\sqrt{K}\,(1-\gamma)^2} + \frac{4\,\sqrt{\ln(A)}\,\gamma^m}{\sqrt{K}\,(1-\gamma)^4}
$$

*Proof.* Using Corollary 3 with Corollary 10 for `NPG` and Corollary 7. $\qquad\square$

### C.4.2 Putting everything together for soft SPMA

**Theorem 7** (Sub-optimality of soft SPMA). *Let $\pi_\tau^*$ denote the optimal entropy-regularized policy with value function $v_\tau^*$. Consider the soft SPMA update with step size $\eta_t = \frac{1}{c + \tau(t+1)}$, $c \geq \max\left\{\frac{4(1+\tau \ln(A))}{(1-\gamma)}, 32\,\tau \ln(A)\right\}$, and, $\delta(\tau, \zeta) := \frac{|\tau - \zeta| \ln(A)}{1 - \gamma}$. Let $\pi_0(\cdot|s)$ be the uniform policy over actions for all $s \in \mathcal{S}$ and assume the policy evaluation step in Eq. (1). Then the resulting mixture policy $\bar{\pi}_K$ satisfies the following sub-optimality bound,*

$$
\left\| v_\tau^{\bar{\pi}_K} - v_\tau^* \right\|_\infty \leq \frac{1}{K(1-\gamma)} \left[ \frac{3(1+\tau \ln(A))^2}{2\,\tau(1-\gamma)^2}\left[1 + \ln(K)\right] + (c+\tau)\ln(A) \right]
$$
$$
+ \frac{16(1+\tau \ln(A))\gamma^m}{(1-\gamma)^4 K}\left[ \left(1 + \tau \ln\left(A\,K^4\right)\right) \right.
$$
$$
\left( \left( \ln(A\,K^4) + \frac{1 + \tau \ln(A)}{\tau(1-\gamma)} \right) \left(1 + \ln(K) + \frac{1}{1 - \sqrt{\gamma}}\right) \right.
$$
$$
\left. + \sqrt{\tau} \right)
$$
$$
\left. + \tau(\ln(A)+1)\,\frac{1}{K} \right] + \frac{2\,\delta(\tau,\zeta)}{(1-\gamma)^2}
$$

*Proof.* Using Theorem 1 with Corollary 11 for soft SPMA, Theorem 3, Corollary 8 and the facts from the proof of Theorem 5. $\qquad \square$

**Corollary 14** (Sub-optimality of soft SPMA with critic entropy). *Let $\pi_\tau^*$ denote the optimal entropy-regularized policy with value function $v_\tau^*$. Consider the soft SPMA update with step size $\eta_t = \frac{1}{c + \tau(t+1)}$, $c \geq \max\left\{\frac{4(1+\tau \ln(A))}{(1-\gamma)}, 32\,\tau \ln(A)\right\}$. Let $\pi_0(\cdot|s)$ be the uniform policy over actions for all $s \in \mathcal{S}$ and assume the policy evaluation step in Eq. (1) with $\zeta = \tau$. Then the resulting mixture policy $\bar{\pi}_K$ satisfies the following sub-optimality bound,*

$$
\left\| v_\tau^{\bar{\pi}_K} - v_\tau^* \right\|_\infty \leq \frac{1}{K(1-\gamma)} \left[ \frac{3(1+\tau \ln(A))^2}{2\,\tau(1-\gamma)^2}\left[1 + \ln(K)\right] + (c+\tau)\ln(A) \right]
$$
$$
+ \frac{16(1+\tau \ln(A))\gamma^m}{(1-\gamma)^4 K}\left[ \left(1 + \tau \ln\left(A\,K^4\right)\right) \right.
$$
$$
\left( \left( \ln(A\,K^4) + \frac{1 + \tau \ln(A)}{\tau(1-\gamma)} \right) \left(1 + \ln(K) + \frac{1}{1 - \sqrt{\gamma}}\right) \right.
$$
$$
\left. + \sqrt{\tau} \right)
$$
$$
\left. + \tau(\ln(A)+1)\,\frac{1}{K} \right]
$$

*Proof.* Using Theorem 1 with Corollary 11 for soft SPMA, Theorem 3, Corollary 8, Theorem 7 and the facts from the proof of Theorem 5. $\qquad \square$

**Corollary 15** (Sub-optimality of soft SPMA without critic entropy). *Let $\pi_\tau^*$ denote the optimal entropy-regularized policy with value function $v_\tau^*$. Consider the soft SPMA update with step size $\eta_t = \frac{1}{c + \tau(t+1)}$, $c \geq \max\left\{\frac{4(1+\tau \ln(A))}{(1-\gamma)}, 32\,\tau \ln(A)\right\}$. Let $\pi_0(\cdot|s)$ be the uniform policy over actions for all $s \in \mathcal{S}$ and assume the policy evaluation step in Eq. (1) with $\zeta = 0$. Then the resulting mixture policy $\bar{\pi}_K$ satisfies the following sub-optimality bound,*

$$
\left\| v_\tau^{\bar{\pi}_K} - v_\tau^* \right\|_\infty \leq \frac{1}{K(1-\gamma)} \left[ \frac{3(1+\tau \ln(A))^2}{2\,\tau(1-\gamma)^2}\left[1 + \ln(K)\right] + (c+\tau)\ln(A) \right]
$$

$$+ \frac{16(1 + \tau \ln(A))\gamma^m}{(1 - \gamma)^4 \, K} \left[ \left(1 + \tau \ln\left(A \, K^4\right)\right) \right.$$

$$\left( \left(\ln(A \, K^4) + \frac{1 + \tau \ln(A)}{\tau(1 - \gamma)}\right) \left(1 + \ln(K) + \frac{1}{1 - \sqrt{\gamma}}\right) \right.$$

$$\left. + \sqrt{\tau} \right)$$

$$\left. + \tau \left(\ln(A) + 1\right) \frac{1}{K} \right] + \frac{2 \, \tau \, \ln(A)}{(1 - \gamma)^3}$$

*Proof.* Using Theorem 1 with Corollary 11 for soft SPMA, Theorem 3, Corollary 9, Theorem 7 and the facts from the proof of Theorem 5. □

**Theorem 8** (SPMA + policy evaluation without entropy regularization)**.** *If $\pi^*$ is the optimal policy whose value function is equal to $v^*$, the SPMA update in Eq. (4) with $\eta_t = \eta = \min\left\{ \frac{1 - \gamma}{2}, \frac{\sqrt{2} \, (1 - \gamma) \, \sqrt{\ln(A)}}{\sqrt{K}} \right\}$, $\pi_0(\cdot|s)$ as the uniform initial policy for each $s \in \mathcal{S}$ with the policy evaluation procedure in Eq. (1) with $\zeta = 0$ satisfies the following sub-optimality bound for the mixture policy $\bar{\pi}_K$,*

$$\left\| v^{\bar{\pi}_K} - v^* \right\|_\infty \le \frac{1}{K \, (1 - \gamma)} \left[ \frac{7 \, \sqrt{\ln(A)} \, \sqrt{K}}{\sqrt{2} \, (1 - \gamma)} + \frac{2}{1 - \gamma} \, \ln(A) \right] + \frac{2 \, \sqrt{\ln(A)} \, \gamma^m}{\sqrt{K} \, (1 - \gamma)^4}$$

*Proof.* Using Corollary 3 with Corollary 11 for SPMA and Corollary 9. □

## D  HELPER LEMMAS

**Lemma 1.** *For any constant $C \in (0, 1/2)$, if $P$ and $Q$ are discrete distributions with support $A$, if $\|P - Q\|_1 \le \frac{1}{2}$, then,*

$$|\mathcal{H}(Q) - \mathcal{H}(P)| \le \|Q - P\|_1 \, \ln\left(\frac{A}{C}\right) + \left(\frac{\ln(A - 1)}{2} + \sqrt{2}\right) \sqrt{C}$$

*Proof.* By Cover (1999, Theorem 17.3.3), if $\|Q - P\|_1 \le \frac{1}{2}$, then,

$$|\mathcal{H}(Q) - \mathcal{H}(P)| \le \|Q - P\|_1 \, \ln\left(\frac{A}{\|Q - P\|_1}\right) \tag{12}$$

Furthermore, using Sason (2013, Theorem 3), we also know that,

$$|\mathcal{H}(Q) - \mathcal{H}(P)| \le \frac{\ln(A - 1)}{2} \, \|Q - P\|_1 + h\left(\frac{1}{2} \, \|Q - P\|_1\right),$$

where $h$ is a binary entropy, i.e. for $0 < x < 1$, $h(x) = -x \ln(x) - (1 - x) \ln(1 - x)$. Using the fact that $h(x) \le 2\sqrt{x(1 - x)}$,

$$\le \frac{\ln(A - 1)}{2} \, \|Q - P\|_1 + \sqrt{2} \sqrt{\|Q - P\|_1 \left(1 - \frac{\|Q - P\|_1}{2}\right)}$$

$$\le \frac{\ln(A - 1)}{2} \, \|Q - P\|_1 + \sqrt{2} \sqrt{\|Q - P\|_1}$$

Assuming $\|Q - P\|_1 \le 2$

$$|\mathcal{H}(Q) - \mathcal{H}(P)| \le \left(\frac{\ln(A - 1)}{2} + \sqrt{2}\right) \sqrt{\|Q - P\|_1} \tag{13}$$

We will use each of these inequalities for different cases of $\|Q - P\|_1$, where $\|Q - P\|_1 \leq \frac{1}{2}$.

**Case (1)**: For any constant $C \in (0, 1/2)$, if $\|Q - P\|_1 \geq C$, then, using Eq. (12),

$$|\mathcal{H}(Q) - \mathcal{H}(P)| \leq \|Q - P\|_1 \ln\left(\frac{A}{C}\right)$$

**Case (2)**: On the other hand, if $\|Q - P\|_1 \leq C$, then we use Eq. (13) to get that,

$$|\mathcal{H}(Q) - \mathcal{H}(P)| \leq \left(\frac{\ln(A-1)}{2} + \sqrt{2}\right) \sqrt{C}$$

Combining both cases, if $\|Q - P\|_1 \leq \frac{1}{2}$, then, for a constant $C \in (0, 1/2)$,

$$|\mathcal{H}(Q) - \mathcal{H}(P)| \leq \|Q - P\|_1 \ln\left(\frac{A}{C}\right) + \left(\frac{\ln(A-1)}{2} + \sqrt{2}\right) \sqrt{C}$$

$\square$

**Lemma 2.** *For any constant $C \in (0, 1/2)$, and $\alpha = \eta\tau$ for $0 < \eta \leq 1$ to be determined, if $P$ and $Q$ are discrete distribution with support $A$ such that $\|Q - P\|_1 \leq \frac{1}{2}$, then,*

$$\frac{1}{2}\|Q - P\|_1^2 \leq \alpha(\mathcal{H}(Q) - \mathcal{H}(P)) \implies \|Q - P\|_1 \leq 4\,\eta\,\tau \ln(\frac{A}{C}) + 2\,\sqrt{\tau}C^{\frac{1}{4}}$$

*Proof.* If the condition on the left-hand side of the lemma is true we have:

$$\|Q - P\|_1^2 \leq 2\,\eta\,\tau(\mathcal{H}(Q) - \mathcal{H}(P))$$

$$\leq 2\,\eta\,\tau\left(2\,\|Q - P\|_1 \ln(\frac{A}{C}) + 2\sqrt{C}\right) \qquad \text{(Using the result from Lemma 1)}$$

$$\leq 4\,\eta\,\tau\,\|Q - P\|_1 \ln(\frac{A}{C}) + 4\,\tau\sqrt{C} \qquad \text{(since } \eta \leq 1\text{)}$$

After completing the square w.r.t $\|Q - P\|_1$ we obtain:

$$\|Q - P\|_1 \leq 4\,\eta\,\tau \ln(\frac{A}{C}) + 2\,\sqrt{\tau}\,C^{\frac{1}{4}}$$

$\square$

**Lemma 3.** *For all $k \in [K]$,*

$$\sum_{i=1}^{k} \frac{\gamma^{k-i}}{(i+1)} \leq \frac{2}{k\,(1-\gamma)} + \frac{\gamma^{k/2}}{1-\gamma}$$

*Proof.* Define $j = k - i$, in which case, we need to bound,

$$\sum_{j=0}^{k-1} \frac{\gamma^j}{k-j+1} = \underbrace{\sum_{j=0}^{\lfloor k/2 \rfloor} \frac{\gamma^j}{k-j+1}}_{\text{Term (i)}} + \underbrace{\sum_{j=\lfloor k/2 \rfloor+1}^{k-1} \frac{\gamma^j}{k-j+1}}_{\text{Term (ii)}}$$

For bounding Term (i), note that $j \leq \frac{k}{2} + 1$, meaning that $k - j + 1 \geq \frac{k}{2}$. Hence,

$$\text{Term (i)} = \sum_{j=0}^{\lfloor k/2 \rfloor} \frac{\gamma^j}{k-j+1} \leq \frac{2}{k} \sum_{j=0}^{\lfloor k/2 \rfloor} \gamma^j \leq \frac{2}{k} \sum_{j=0}^{\infty} \gamma^j \leq \frac{2}{k} \frac{1}{1-\gamma}$$

For bounding Term (ii), note that since $j \leq k$, $k - j + 1 \geq 1$. Hence,

$$\text{Term (ii)} = \sum_{j=\lfloor k/2 \rfloor+1}^{k-1} \frac{\gamma^j}{k-j+1} \leq \sum_{j=\lfloor k/2 \rfloor+1}^{k-1} \gamma^j \leq \sum_{j=\lfloor k/2 \rfloor+1}^{\infty} \gamma^j \leq \frac{\gamma^{k/2}}{1-\gamma}$$

Combining both terms,

$$\sum_{j=0}^{k-1} \frac{\gamma^j}{k-j+1} \leq \frac{2}{k} \frac{1}{1-\gamma} + \frac{\gamma^{k/2}}{1-\gamma}$$

$\square$

**Lemma 4.** *For any state-action value function q, for any $m \geq 1$,*

$$|(T_\tau^{\pi_1}q)^m(s,a) - (T_\zeta^{\pi_1}q)^m(s,a)| \leq \frac{|\tau - \zeta|}{1 - \gamma} \ln(A)$$

*Proof.*

$$(T_\tau^{\pi_1}q)^m(s,a) = \mathbb{E}_{\substack{s_{t+1} \sim \mathcal{P}(\cdot|s_t,a_t) \\ a_t \sim \pi_1(\cdot|s_t)}} \left[ \sum_{t=0}^{m-1} \gamma^t \big(r(s_t,a_t) - \tau \ln \pi_1(a_t|s_t)\big) \right. \tag{14}$$

$$\left. + \gamma^m q(s_m, a_m) \,\Big|\, s_0 = s, \, a_0 = a \right] \qquad \text{(By definition of } T_\tau^{\pi_1})$$

$$= \mathbb{E}\left[ \sum_{t=0}^{m-1} \gamma^t (r(s_t,a_t)) + \gamma^m q(s_m,a_m) \right] - \tau \, \mathbb{E}\left[ \sum_{t=0}^{m-1} \gamma^t \ln(\pi_1(a_t|s_t)) \right] \tag{15}$$

$$+ \zeta \, \mathbb{E}\left[ \sum_{t=0}^{m-1} \gamma^t \ln(\pi_1(a_t|s_t)) \right] - \zeta \, \mathbb{E}\left[ \sum_{t=0}^{m-1} \gamma^t \ln(\pi_1(a_t|s_t)) \right]$$

$$\text{(Add/Subtract } \zeta \, \mathbb{E}\left[ \sum_{t=0}^{m-1} \gamma^t \ln(\pi_1(a_t|s_t)) \right])$$

$$= (T_\zeta^{\pi_1}q)^m(s,a) - (\tau - \zeta) \, \mathbb{E}\left[ \sum_{t=0}^{m-1} \gamma^t \ln(\pi_1(a_t|s_t)) \right] \qquad \text{(By definition of } T_\zeta^{\pi_1})$$

$$= (T_\zeta^{\pi_1}q)^m(s,a) + (\tau - \zeta) \, \mathbb{E}\left[ \sum_{t=0}^{m-1} \gamma^t \, \mathcal{H}(\pi_1(\cdot|s_t)) \right]$$

$$\text{(By definition of } \mathcal{H}(\pi_1(\cdot|s_t)))$$

$$\leq (T_\zeta^{\pi_1}q)^m(s,a) + \frac{|\tau - \zeta|}{1 - \gamma} \ln(A) \qquad \text{(Since } \mathbb{E}[\mathcal{H}(\pi_1(\cdot|s_t))] \leq \ln(A))$$

$$\square$$

# E    Experimental Details and Additional Results

## E.1    Details for Stable Baselines Experiments

Following Tomar et al. (2020); Asad et al. (2024) we use the default hyperparameters from `stable-baselines3` (Raffin et al., 2021) for each method. This choice is motivated by prior work, which focuses on evaluating the effectiveness of different surrogate losses rather than performing exhaustive hyperparameter searches. Such searches are particularly impractical for CNN-based actor and critic networks, where tuning multiple hyperparameters (e.g., framestack, buffer size, minibatch size, discount factor) is computationally expensive. The full list of hyperparameters for the Atari experiments is provided in Table 1.

For our forward and reverse KL objectives, we set $\eta_t$ to a constant chosen via a small grid search over $[0.01, 0.1, 1.0]$. Following prior work (Haarnoja et al., 2018; Christodoulou, 2019; Mnih et al., 2013) for policy evaluation, $q_\zeta^{t-1}$ in Eq. (2) is parameterized using a separate target network, whose parameters are updated via an exponential moving average of those in the critic model. For on-policy `PPO`, we adopt the optimal hyperparameters reported in RL Baselines3 Zoo (Raffin, 2020).

| Hyperparameter | FKL Objectives | RKL Objectives | DSAC | DQN |
|---|---|---|---|---|
| Reward normalization | ✗ | ✗ | ✗ | ✗ |
| Observation normalization | ✗ | ✗ | ✗ | ✗ |
| Orthogonal weight initialization | ✗ | ✗ | ✗ | ✗ |
| Value function clipping | ✗ | ✗ | ✗ | ✗ |
| Gradient clipping | ✗ | ✗ | ✗ | ✗ |
| Probability ratio clipping | ✗ | ✗ | ✗ | ✗ |
| Entropy coefficient | auto | auto | auto | $\epsilon$-greedy |
| Adam step-size | $3 \times 10^{-4}$ | | | |
| Buffer size | $10^6$ | | | |
| Minibatch size | 256 | | | |
| Framestack | 4 | | | |
| Number of environment copies | 8 | | | |
| Discount factor | 0.99 | | | |
| Total number of timesteps | $10^7$ | | | |
| Number of runs for plot averages | 5 | | | |
| Confidence interval for plot runs | $\sim 95\%$ | | | |

**Table 1:** Hyper-parameters for Atari experiments.

## E.2 Does Double $Q$ Learning Hurt or Improve Performance?

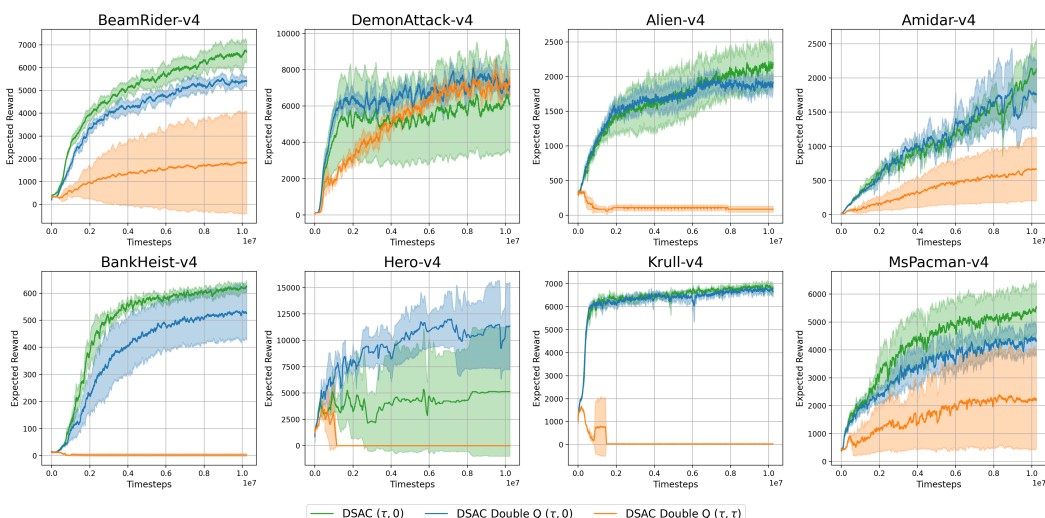

**Figure 6:** When critic entropy is disabled, Double-$Q$ learning performs comparably to using a single $Q$, with no conclusive evidence that one approach consistently outperforms the other.

In Fig. 6, we evaluate single vs. double $Q$-learning on 8 Atari games under our setup, which uses the hard Bellman operator, adaptive entropy via entropy coefficient loss, and a fixed target entropy. Double $Q$-learning performs comparably to single $Q$-learning. Notably, for the bottom four games, our DSAC configuration substantially outperforms the results reported in Xu et al. (2021), despite their use of an adaptive target entropy (see Figure2 in Xu et al. (2021))

### E.3  Our Objectives Benefit Similarly from the Hard Bellman Operator

Using our proposed objectives, we consistently observe the same trend as with `DSAC`: employing the soft Bellman operator ($\zeta = \tau$) with an adaptive entropy coefficient leads to poor performance compared to `DQN`. In contrast, switching to the hard Bellman operator yields substantial improvements (see left vs. right columns of Fig. 7).

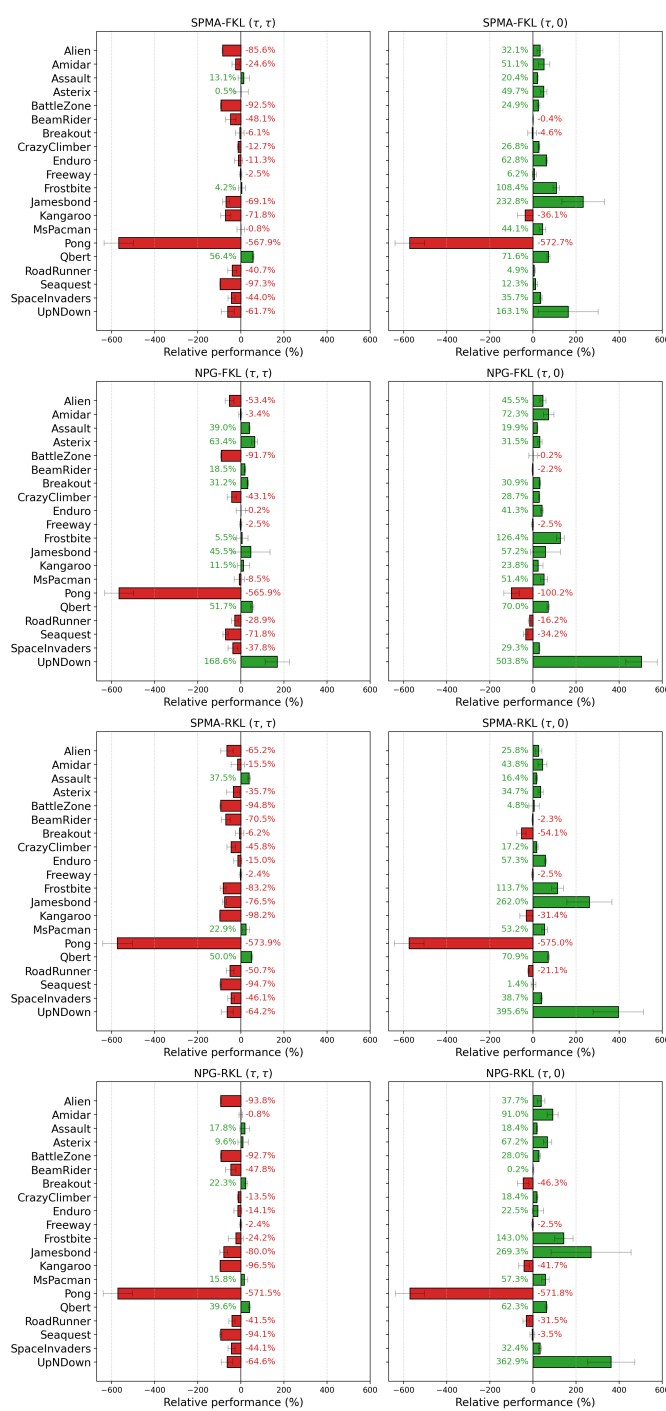

**Figure 7:** Disabling critic entropy while retaining the adaptive entropy coefficient loss, as in `DSAC`, yields substantial performance gains for both our forward and reverse KL actors (left vs. right columns).

### E.4 ADDITIONAL RESULTS: IS ENTROPY REGULARIZATION NECESSARY?

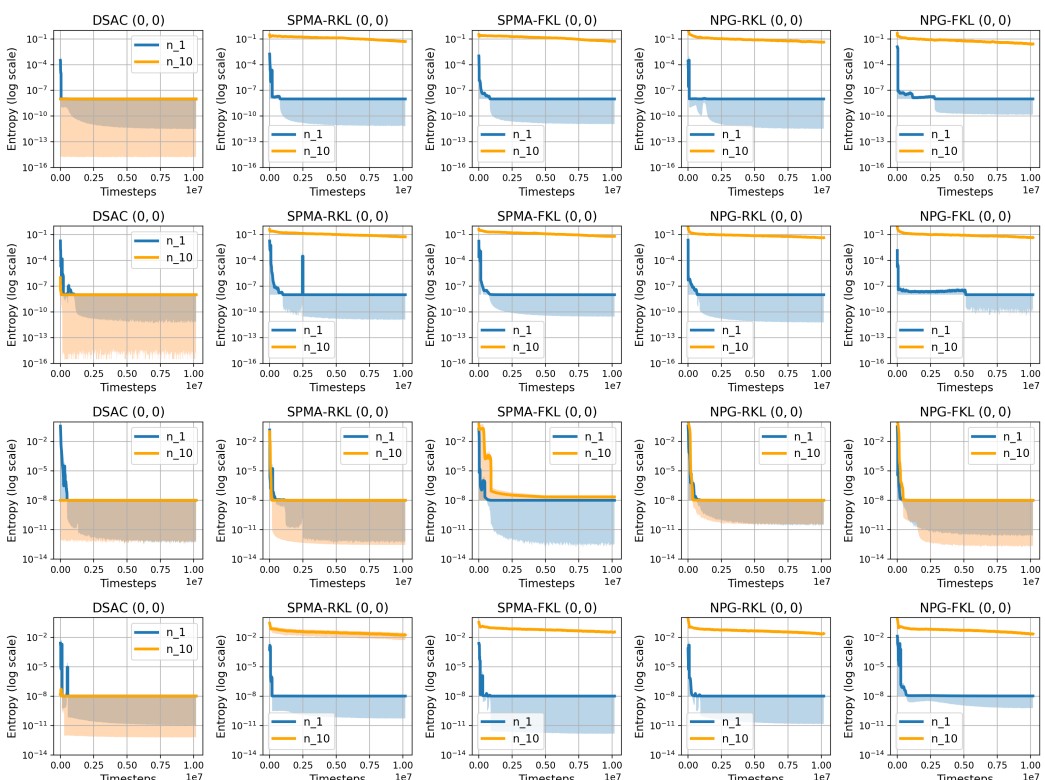

**Figure 8:** Policy Shannon entropy during training. Without entropy regularization, increasing $n$ from 1 to 10 substantially increases and stabilizes the Shannon entropy for our forward and reverse KL-based methods, compared to `DSAC`, across four Atari games. The rows correspond to MsPacman-v4, Seaquest-v4, Freeway-v4, and Breakout-v4.

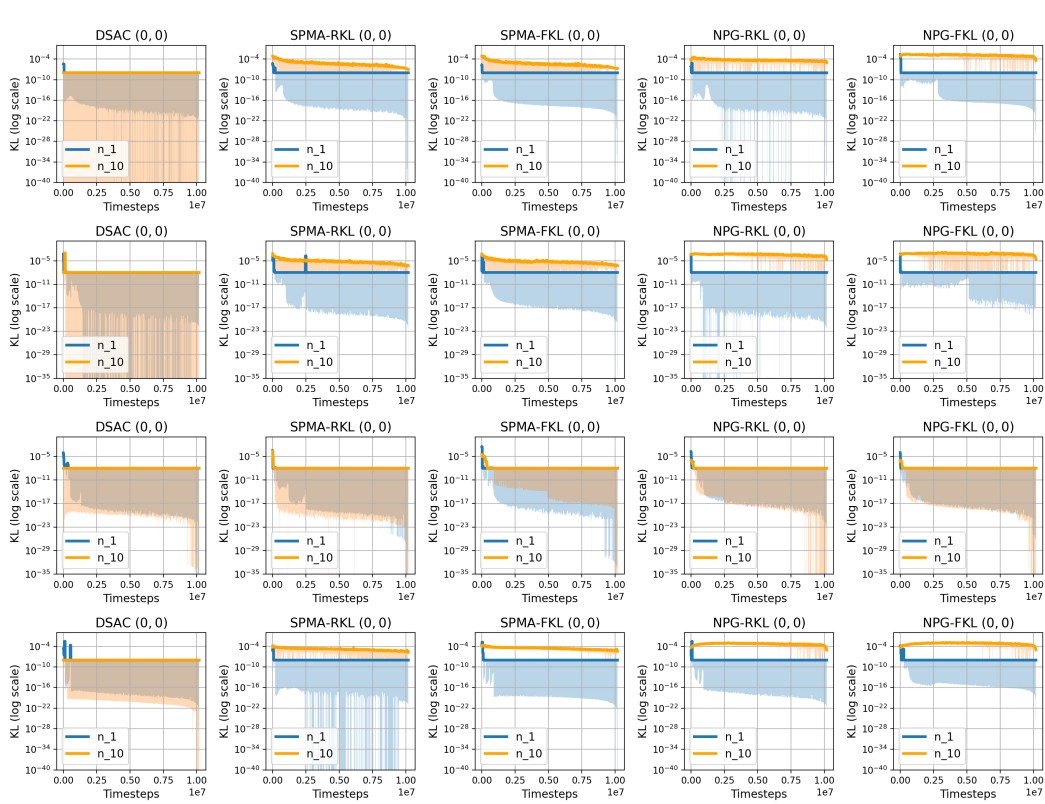

**Figure 9:** KL divergence during training. Consistent with the Shannon entropy results, in the absence of entropy regularization, increasing $n$ to 10 allows our forward- and reverse-KL-based actors to achieve higher KL divergence than `DSAC`, which also stabilizes over the course of training. The rows correspond to MsPacman-v4, Seaquest-v4, Freeway-v4, and Breakout-v4.

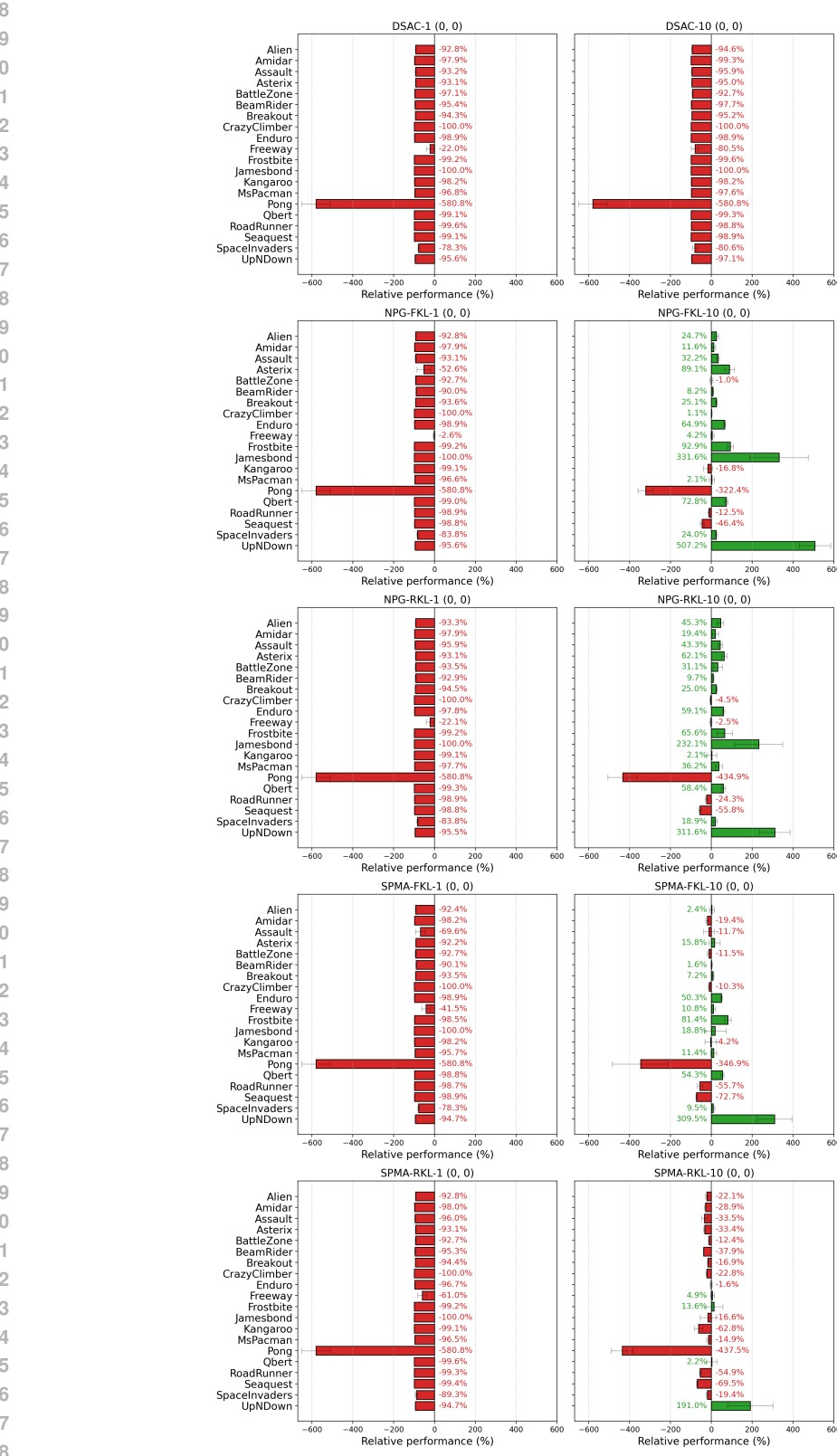

**Figure 10:** Effect of the number of actor optimization steps $n$ without entropy regularization. Unlike DSAC, all our objectives (except SPMA-RKL$(0,0)$) achieve performance comparable to DQN when $n$ is increased from 1 to 10 (left vs. right columns).

## E.5 ADDITIONAL RESULTS: DIRECTION OF KL MOSTLY DOES NOT MATTER

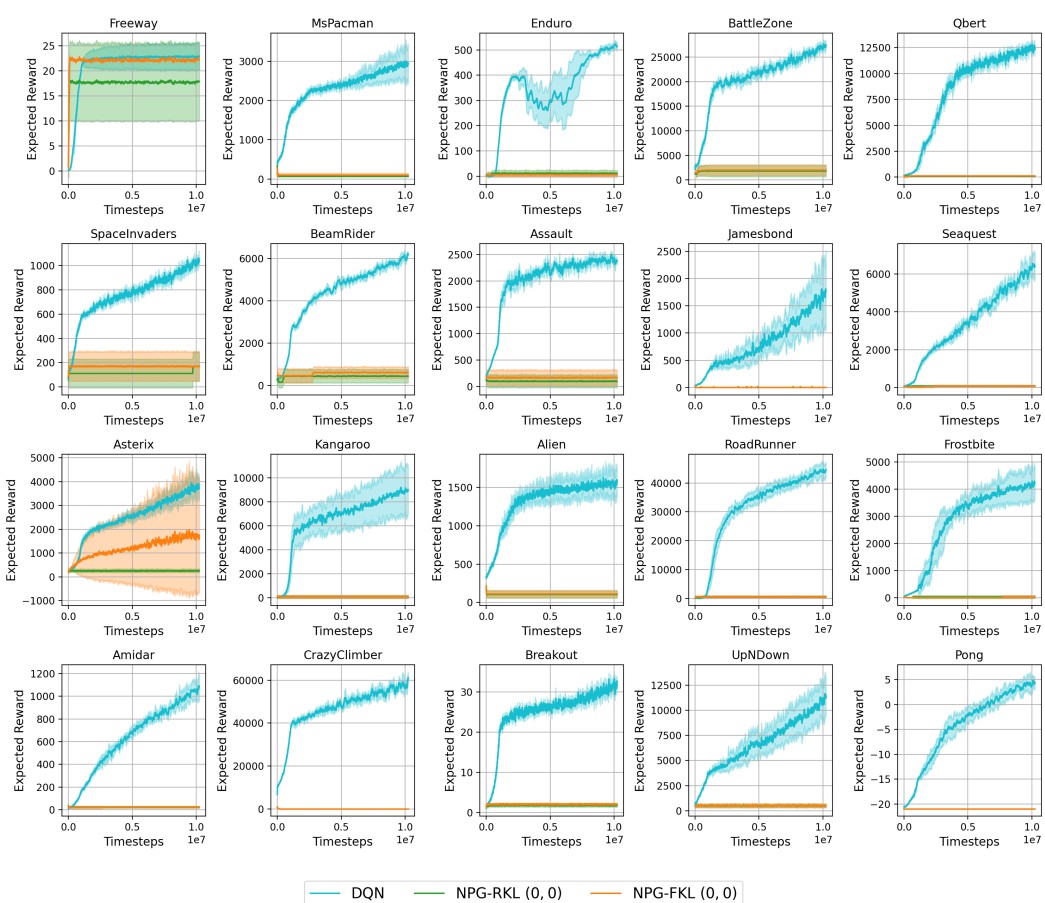

**Figure 11:** Direction of KL divergence without entropy regularization, using $n = 1$ and `NPG` as the intermediate policy. Across nearly all games, performance remains poor relative to `DQN`, and the KL direction has no conclusive effect.

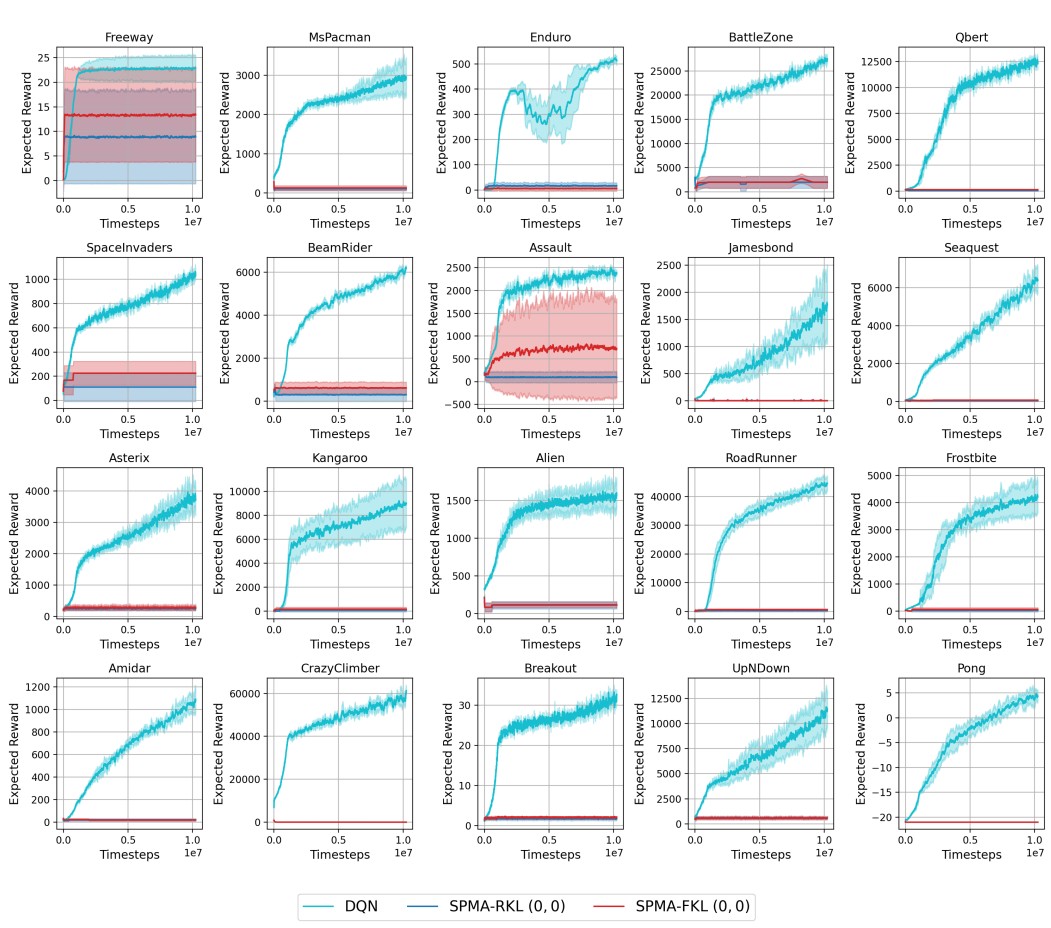

**Figure 12:** Direction of KL divergence without entropy regularization, using $n = 1$ and `SPMA` as the intermediate policy. Across nearly all games, performance remains poor relative to `DQN`, and the KL direction has no conclusive effect.

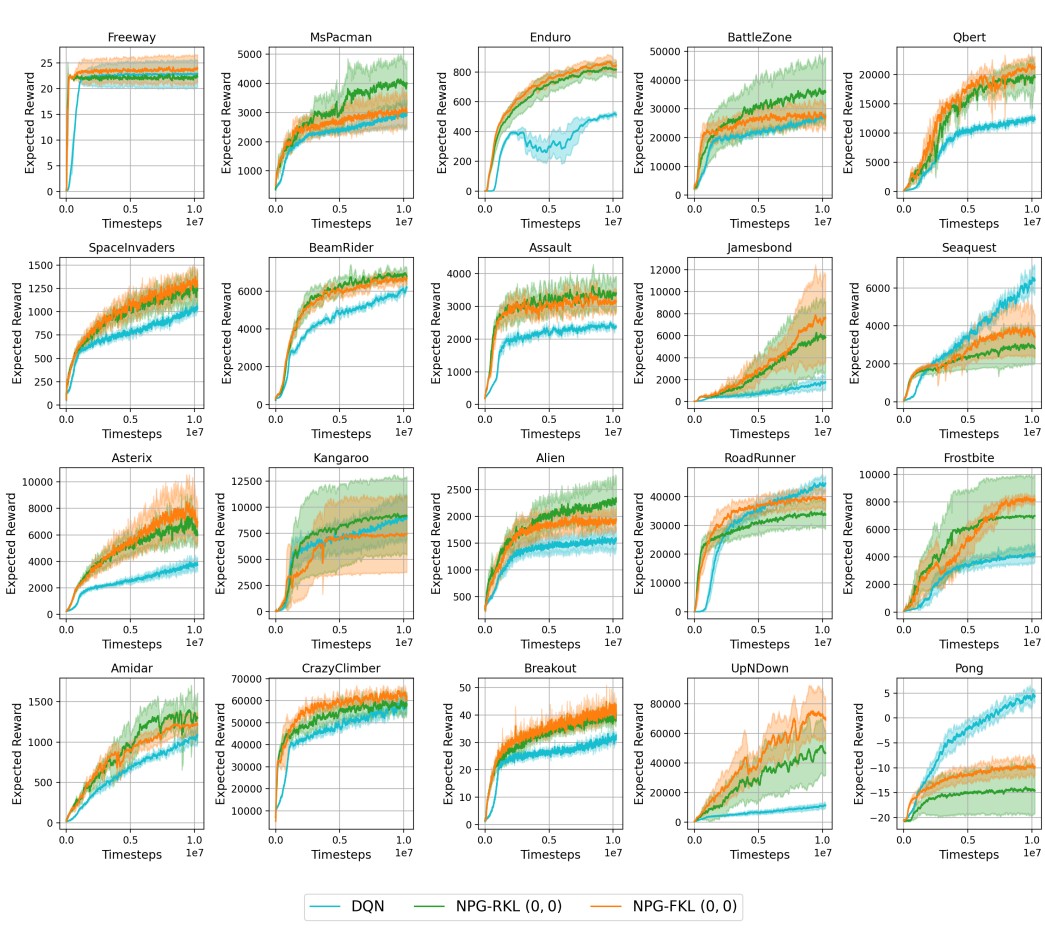

**Figure 13:** Direction of KL divergence without entropy regularization, using $n = 10$ and NPG as the intermediate policy. Both NPG-RKL$(0, 0)$ and NPG-FKL$(0, 0)$ achieve performance comparable to DQN, but no KL direction consistently outperforms the other.

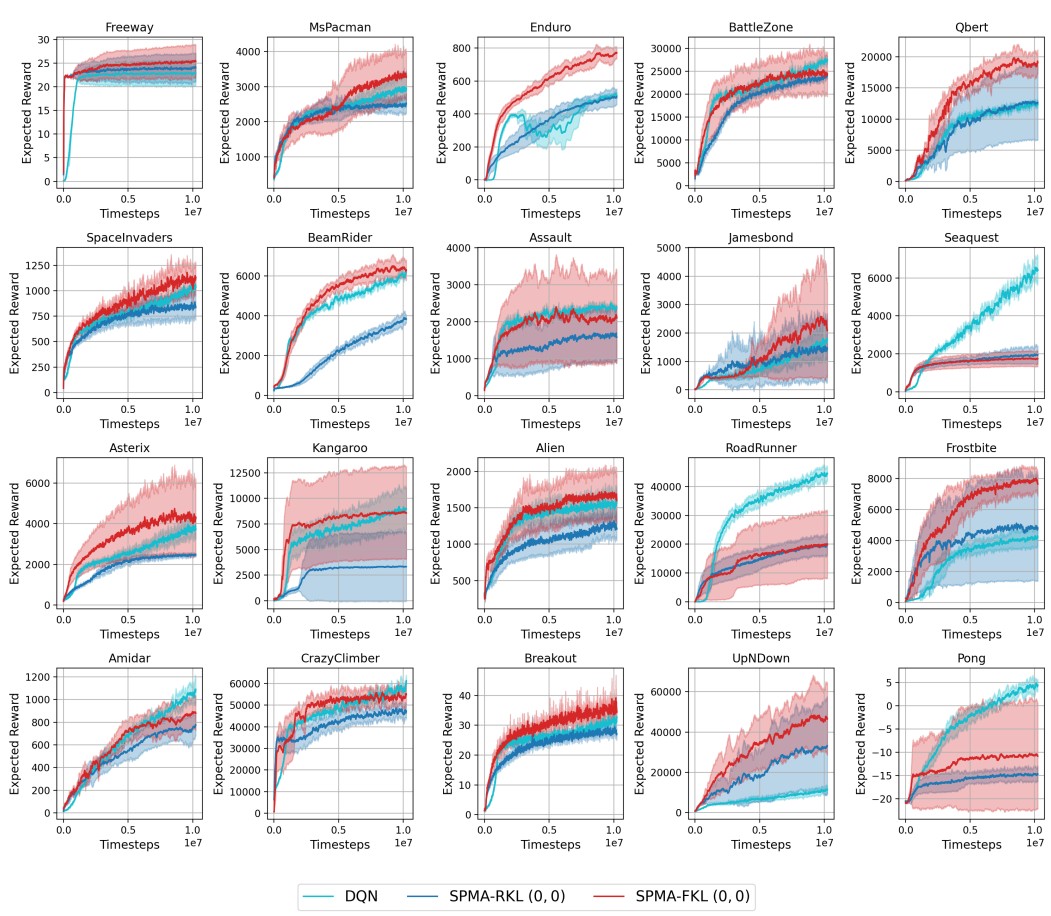

**Figure 14:** Direction of KL divergence without entropy regularization, using $n = 10$ and SPMA as the intermediate policy. SPMA-FKL$(0, 0)$ achieves performance comparable to DQN, and overall, the forward KL direction appears to outperform SPMA-RKL$(0, 0)$.

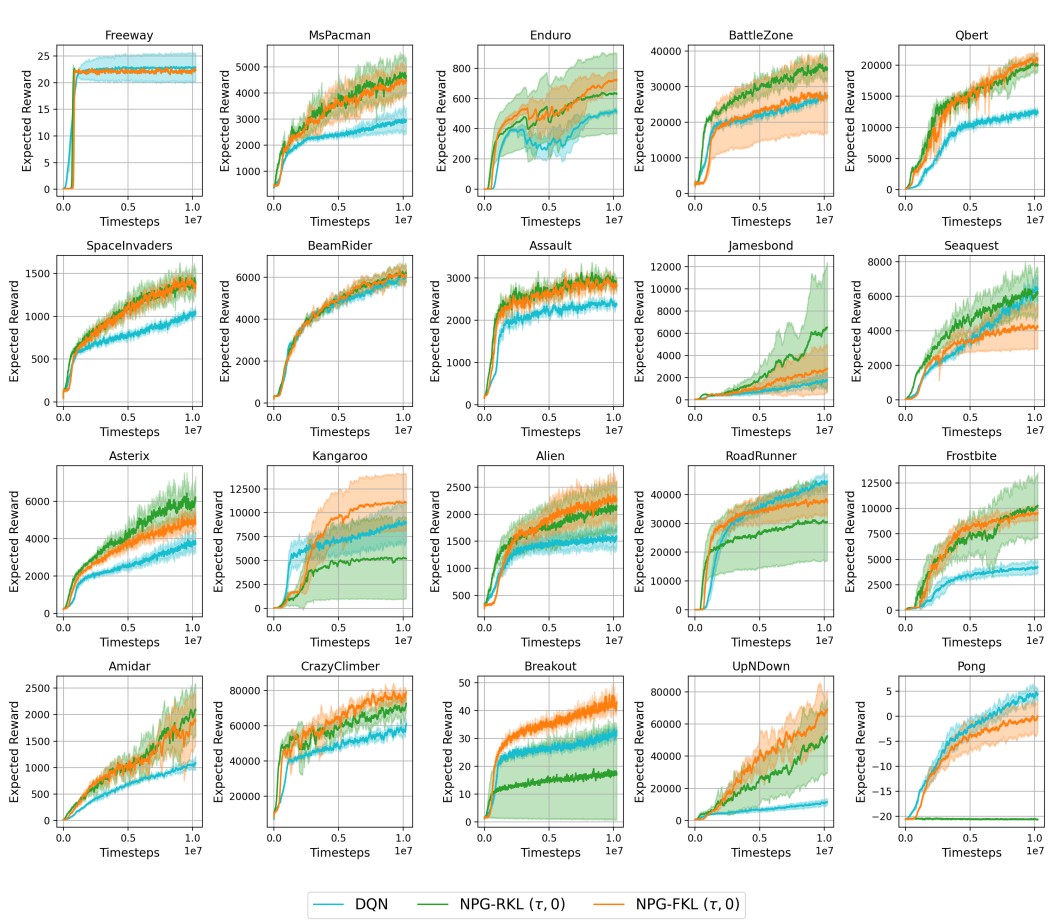

**Figure 15:** Direction of KL divergence with actor entropy regularization, using $n = 1$ and NPG as the intermediate policy. Both NPG-FKL$(\tau, 0)$ and NPG-RKL$(\tau, 0)$ achieve performance comparable to DQN, even with $n = 1$ but no KL direction consistently outperforms the other.

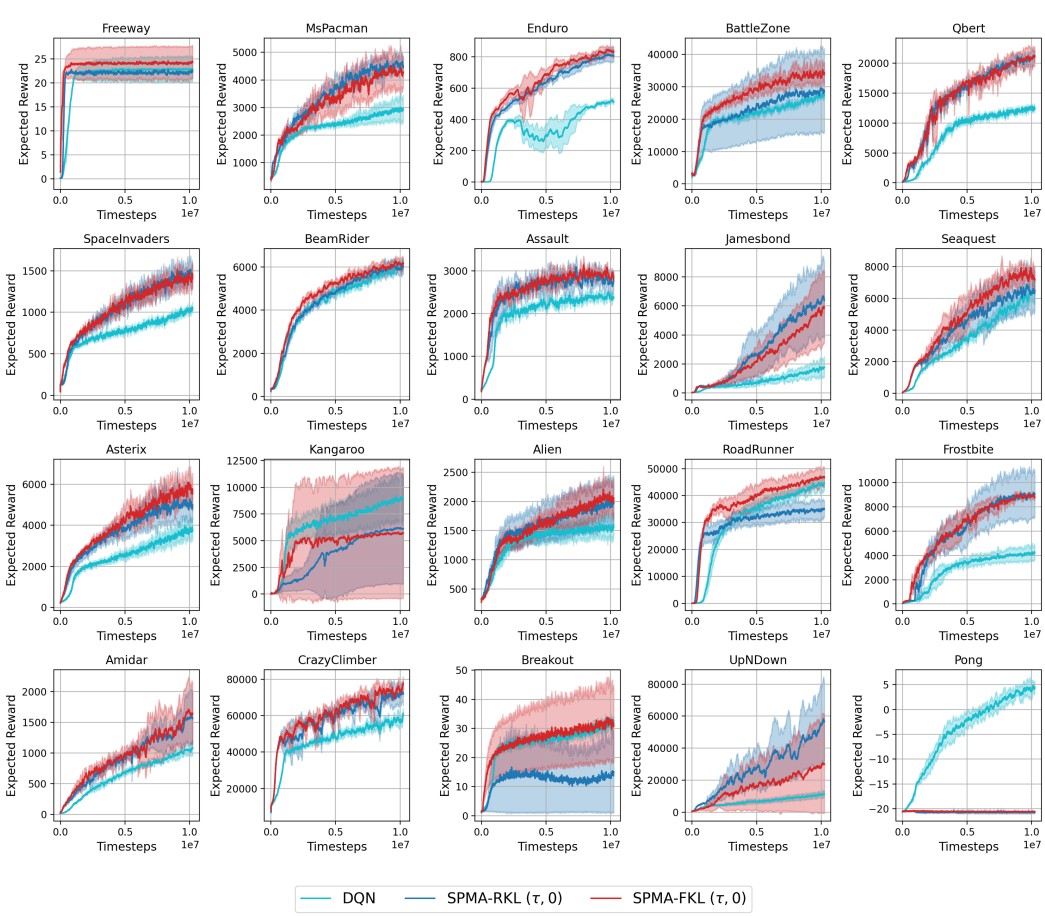

**Figure 16:** Direction of KL divergence with actor entropy regularization, using $n = 1$ and SPMA as the intermediate policy. In the presence of actor entropy, both SPMA-FKL$(\tau, 0)$ and SPMA-RKL$(\tau, 0)$ achieve performance comparable to DQN, even with $n = 1$, but no KL direction consistently outperforms the other.

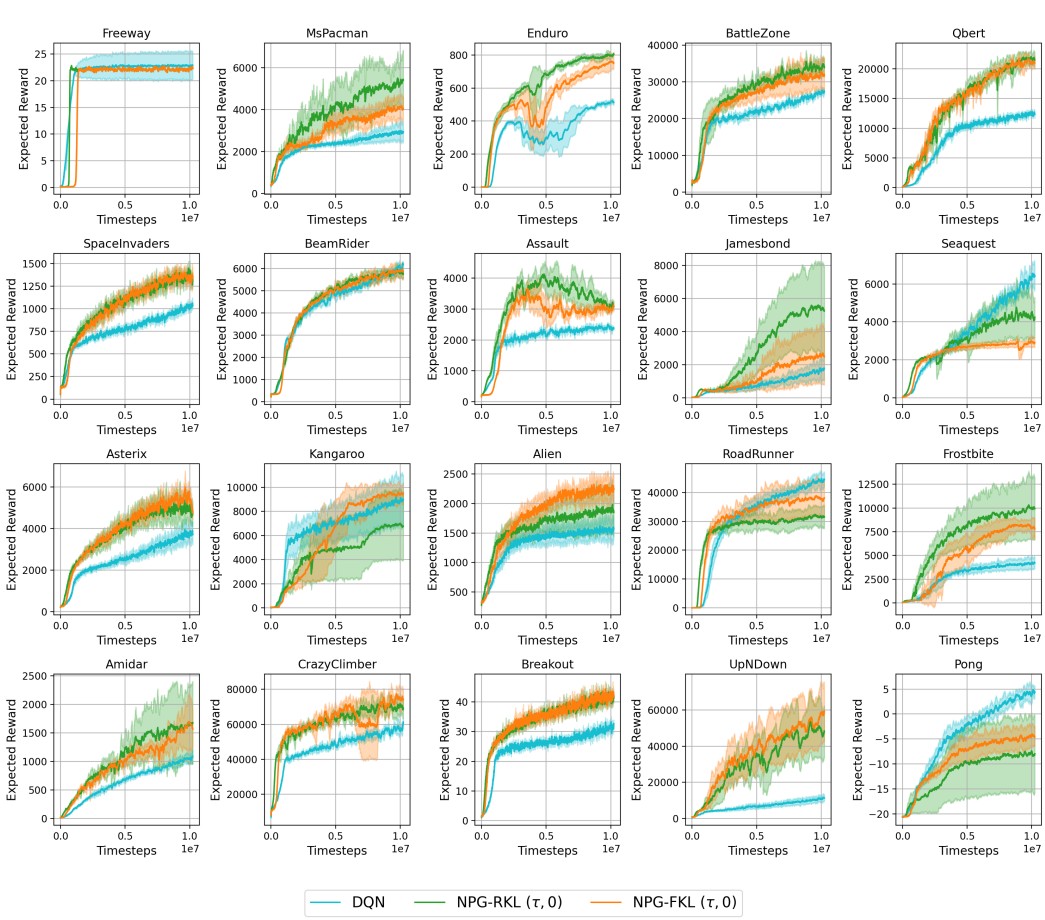

**Figure 17:** Direction of KL divergence with actor entropy regularization, using $n = 10$ and NPG as the intermediate policy. In the presence of actor entropy, both NPG-FKL$(\tau, 0)$ and NPG-RKL$(\tau, 0)$ achieve performance comparable to DQN, but no KL direction consistently outperforms the other.

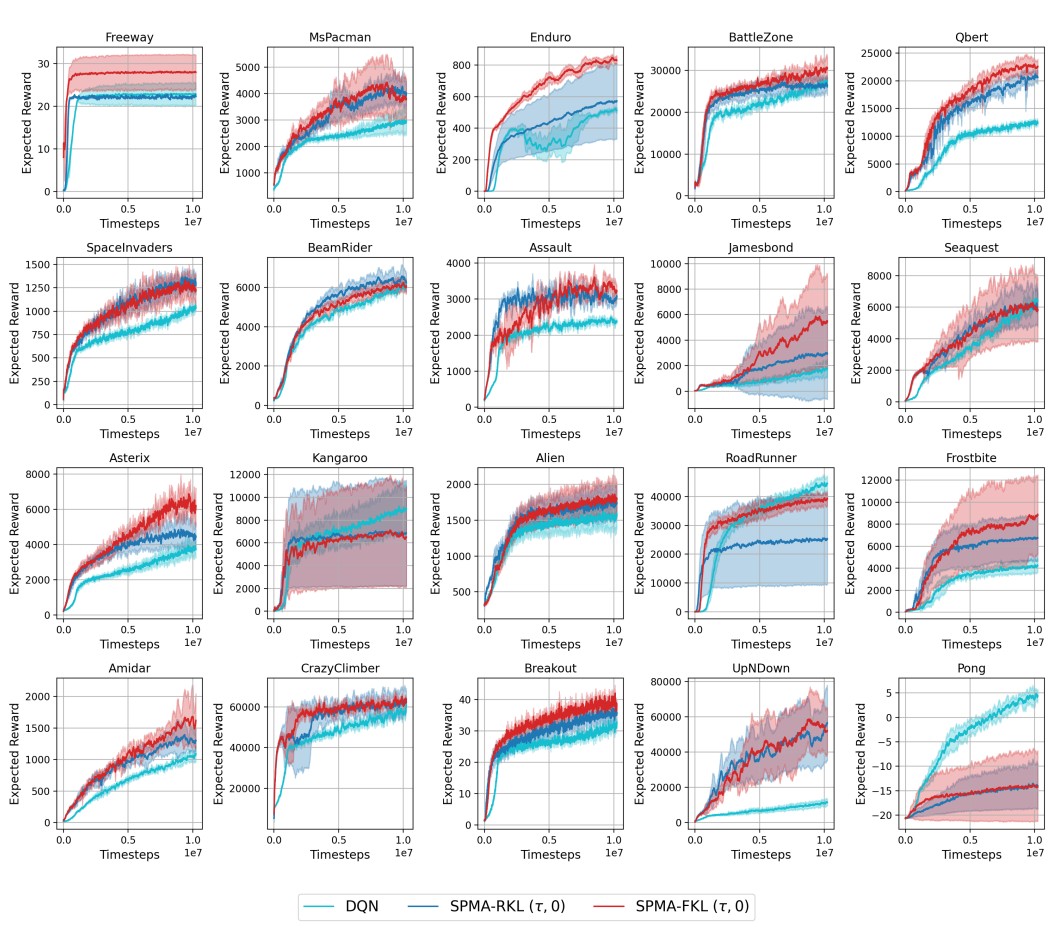

**Figure 18:** Direction of KL divergence with actor entropy regularization, using $n = 10$ and SPMA as the intermediate policy. In the presence of actor entropy, both SPMA-FKL$(\tau, 0)$ and SPMA-RKL$(\tau, 0)$ achieve performance comparable to DQN, but no KL direction consistently outperforms the other.

### E.6 COMPARISON TO ON-POLICY PPO

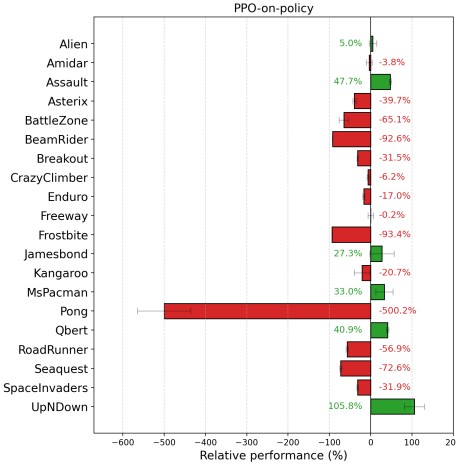

**Figure 19:** Performance of on-policy PPO relative to DQN across 20 Atari games. Overall, PPO underperforms DQN on most games.

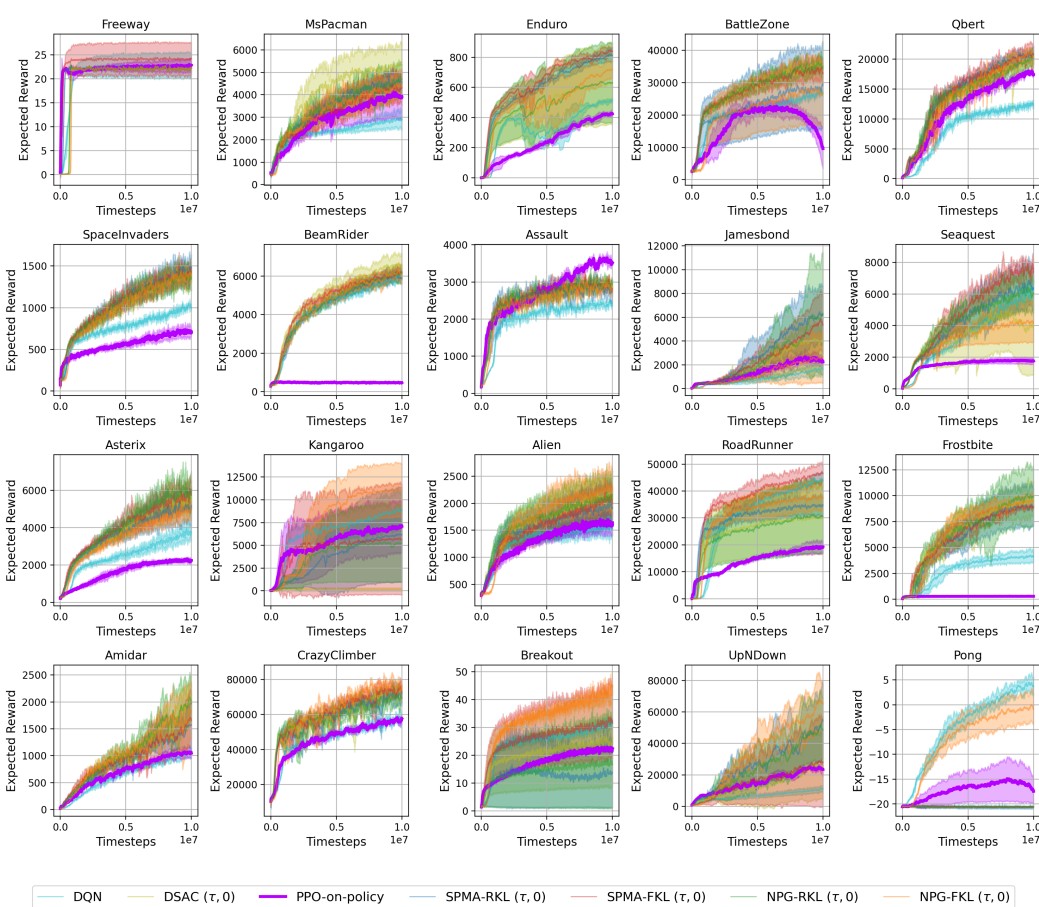

**Figure 20:** Comparison of on-policy PPO with all off-policy objectives presented in this paper over the full training horizon. Overall, PPO underperforms on most games.

### E.7 Comparison to Adam LMCDQN

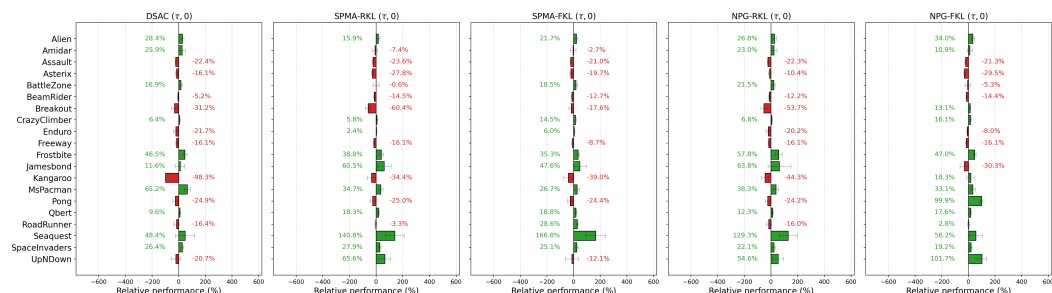

**Figure 21:** Performance of the objectives from our proposed framework relative to `Adam LMCDQN` across 20 Atari games. Overall, we observe performance comparable to `Adam LMCDQN`.

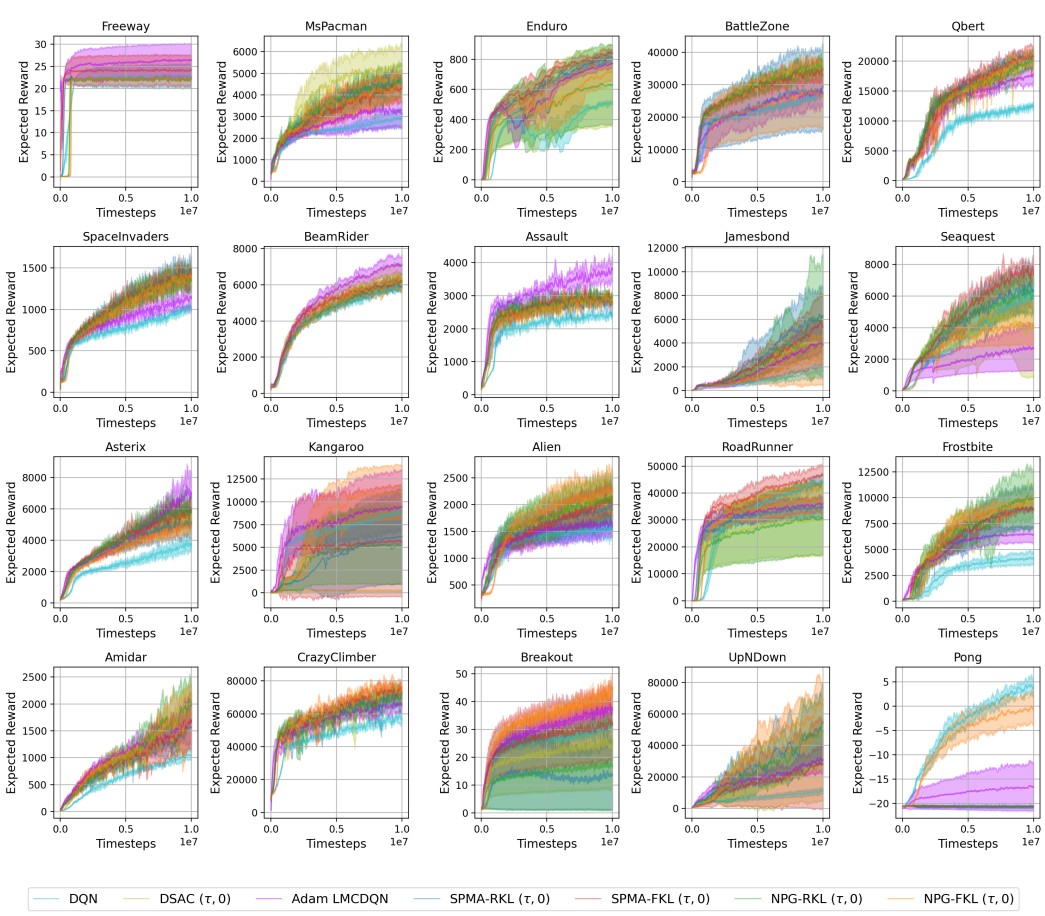

**Figure 22:** Comparing `Adam LMCDQN` with all off-policy objectives presented in this paper over the full training horizon. Overall, the objectives from our proposed framework exhibit performance comparable to `Adam LMCDQN`.

## E.8 DSAC Ablation on Pong

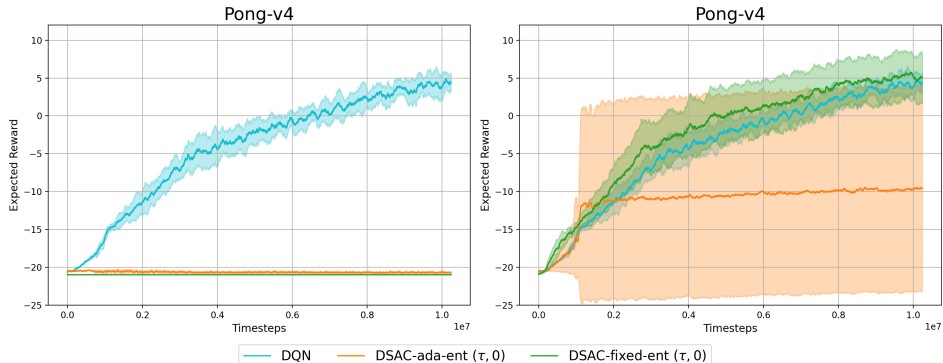

**Figure 23:** Left: With a single actor update, `DSAC` underperforms `DQN` under both adaptive and fixed entropy coefficients, even in the absence of critic entropy. Right: Increasing the number of actor updates substantially improves performance, and using a small fixed entropy coefficient (i.e., `DSAC`-fixed-ent$(\tau, 0)$) yields the strongest results.

### E.9 COMPARISON TO GreedyAC

We examine the empirical finding in our DSAC ablation in Section 5 (see Fig. 2) on MuJoCo's discrete control tasks. The results in Figures 24 and 25 support our observation: DSAC with a fixed, per-environment tuned entropy coefficient, with or without critic entropy, attains strong performance on par with the prior discrete actor-critic framework GreedyAC (Neumann et al., 2018).

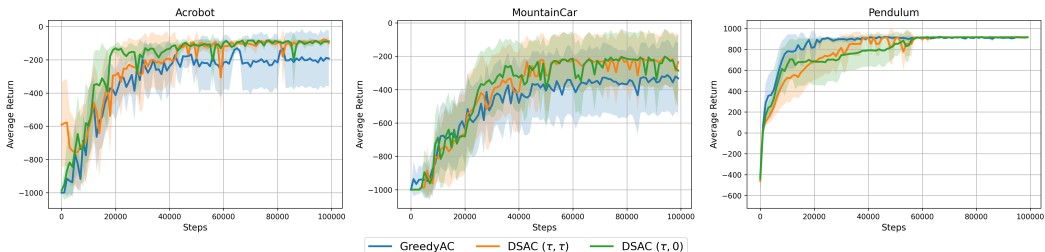

**Figure 24:** Comparing GreedyAC with the default DSAC (with and without critic entropy and a fixed entropy coefficient) using batch size 32.

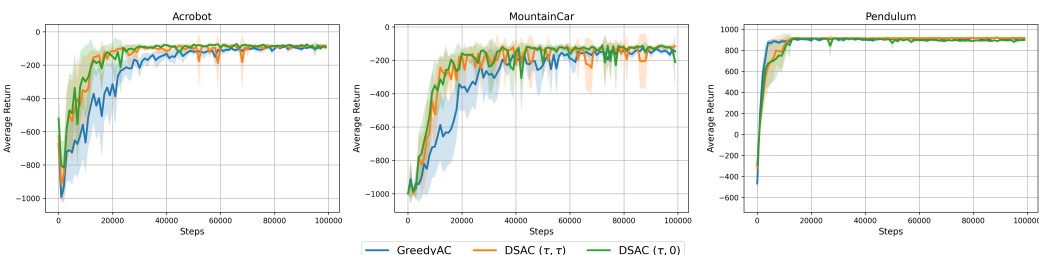

**Figure 25:** Comparing GreedyAC with the default DSAC (with and without critic entropy and a fixed entropy coefficient) using batch size 256.

When ablating the default `DSAC` on four Atari games in Section 5 (Fig. 2), we find that it performs well on some games but fails on others (Alien and BeamRider). The results in Fig. 26 show that poor performance correlates with a collapse in policy entropy (columns 1 and 3, orange curve). Prior work (Xu et al., 2021) attributes this collapse to the fixed target entropy in `DSAC`, and our observations support this: reducing the learning rate for the entropy coefficient loss improves performance (blue curve). Nevertheless, removing critic entropy eliminates this issue, without requiring additional tuning of the entropy coefficient loss and results in strong performance.

### E.10    Entropy Collapse of Default `DSAC`

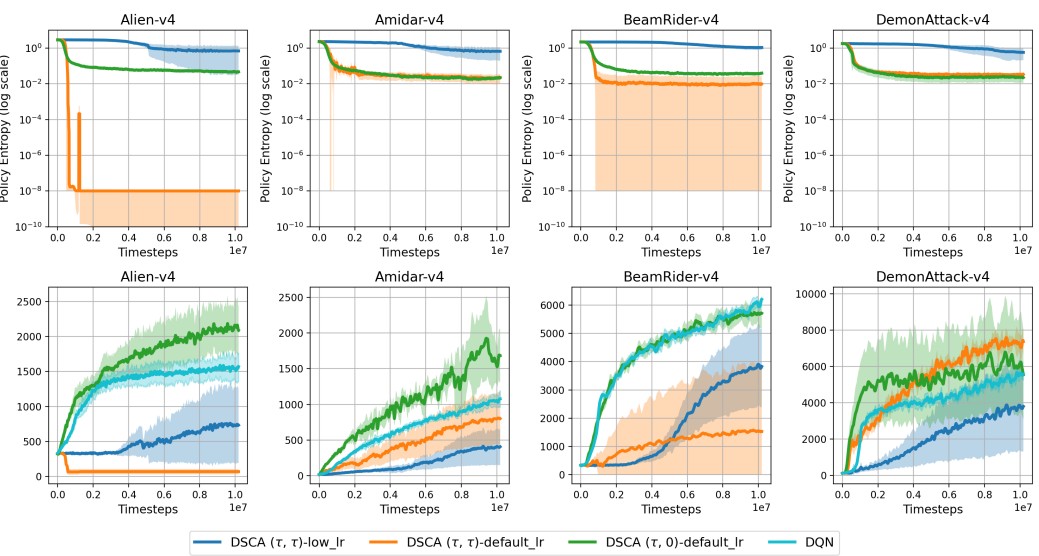

**Figure 26:** The poor performance of default `DSAC` (orange curve) correlates with a collapse in policy entropy (columns 1 and 3). Although lowering the learning rate for the entropy coefficient loss improves performance, removing the critic entropy term resolves the issue without additional hyperparameter tuning.

## F  LLM Usage

We acknowledge the use of OpenAI's ChatGPT to improve the grammar and compactness of sentences originally written by the authors, helping them fit within the page limit. All ideas, results, and contributions in this paper are solely those of the authors.

