# OpenReview forum: "Revisiting Actor-Critic Methods in Discrete Action Off-Policy Reinforcement Learning"
_ICLR.cc/2026/Conference — Submitted to ICLR 2026_

### Official Review · Reviewer_7fk8 · 2025-10-20

**Soundness:** 2
**Presentation:** 1
**Contribution:** 1
**Rating:** 2
**Confidence:** 3

**Summary:**

This paper investigates off-policy actor-critic methods for discrete action tasks. Specifically, it empirically examines the effect of entropy regularization in both the critic and actor training processes, as well as the impact of variations in the objective function. The empirical results demonstrate how the inclusion of the entropy term influences performance in discrete action settings.

**Strengths:**

- Off-policy actor-critic methods for discrete action tasks remain relatively under-explored, making this study a timely and interesting contribution.

- The finding that entropy regularization in the critic is not necessary for discrete action tasks is particularly noteworthy and potentially useful for practitioners.

I agree that DQN variants currently dominate discrete action tasks, and thus exploring off-policy actor-critic methods in this domain is a meaningful research direction. The insight regarding the limited necessity of entropy regularization in the critic could offer practical implications for future algorithm design.

**Weaknesses:**

- The presentation lacks clarity in several sections, making the paper somewhat difficult to follow.

- The reported performance remains weak and still far from the current state of the art.

Regarding Equation (3), while it is referred to as the “natural policy gradient (NPG),” I do not believe this terminology is appropriate. Although both the solution in Equation (3) and NPG can be derived from reward maximization under a KL-divergence constraint, the solution represents a non-parametric solution to the reinforcement learning problem (Peters et al., 2010; Abdolmaleki et al., 2018). In contrast, NPG refers to the gradient direction of policy updates defined via the Fisher information matrix. While I acknowledge the conceptual connections, I am not convinced that Equation (3) should be described as NPG.

Additionally, the term “proposed methods” appears to refer to algorithms based on the objective functions shown in lines 251–262. However, these algorithms have already been introduced in prior studies, and it may not be appropriate to refer to them as “proposed.”

The experimental results are reported based on the parameters $\zeta$ and $\tau$. However, this notation is not intuitive, and I found it necessary to reread multiple sections to fully understand the meaning of the results.

The authors conclude that the investigated objective function achieved performance comparable to that of DQN. However, since DQN itself is far from state-of-the-art methods such as BTR [R1], the performance of the investigated off-policy algorithms also appears to be limited. Therefore, the reported results are not particularly promising.

Reference:
[R1] Beyond The Rainbow: High Performance Deep Reinforcement Learning on a Desktop PC. Tyler Clark, Mark Towers, Christine Evers, Jonathon Hare. ICML 2025.

**Questions:**

- Although the authors investigated the objective function based on SPMA and NPG, I am not sure why SPMA is employed as the baseline. NPG is much more well-known than SPMA, and the performance of NPG-based methods appear to be better than those based on SPMA. Please explain why the analysis based on SPMA was necessary.

- I believe there are additional reasons why DQN variants dominate in the discrete action domain, beyond the effects of entropy regularization and the KL-divergence constraint. One important factor is that, in the discrete action setting, the Bellman optimality operator can be directly applied because the maximum over actions can be explicitly computed. This is not feasible in the continuous action domain.
This fundamental difference may explain why DQN variants are typically preferred for discrete action tasks, whereas off-policy actor-critic methods are more suitable for continuous action domains. Actor-critic methods are inherently indirect, as they rely on policy iteration, in contrast to value iteration methods such as DQN, which can directly approximate the optimal value function.
However, this important discussion appears to be missing from the paper. I would like to ask the authors for their thoughts on this point.

---

> ### Author Response · Authors · 2025-11-22
>
> We thank the reviewer for their feedback and address their main questions and comments below.
> > "The reported performance remains weak and still far from the current state of the art".
>
> > "The authors conclude that the investigated objective function achieved performance comparable to that of DQN. However, since DQN itself is far from state-of-the-art methods such as BTR [R1], the performance of the investigated
>     off-policy algorithms also appears to be limited. Therefore, the reported results are not particularly promising."
>
> - We emphasize that our goal is not to compete with the full suite of highly engineered $\texttt{DQN}$ variants, but to understand why $\texttt{DSAC}$ [1] underperforms even vanilla $\texttt{DQN}$ [2, 3]. We use this insight to build a principled actor-critic framework for the off-policy discrete-action setting. Given the importance of policy optimization methods to applications including aligning LLMs, we believe that this is an important research direction. We agree that incorporating heuristics (similar to recent variants of $\texttt{DQN}$) and improving the performance of $\texttt{DSAC}$ is an interesting direction for future work. In order to do so, we believe that our paper takes an important first step.
> - To further address the reviewer's concern, we also compared our objectives to a recent strong baseline, $\texttt{Adam LMCDQN}$ [4]. We chose this baseline for two reasons:
>     1. $\texttt{Adam LMCDQN}$ outperforms or is on par with strong methods such as $\texttt{NoisyNetDQN}$ [5], $\texttt{QRDQN}$ [6], $\texttt{IQN}$ [7], and $\texttt{C51}$ [8]
>     2. $\texttt{Adam LMCDQN}$ does not use $\epsilon$-greedy and is implemented as a modified Adam optimizer that can be easily integrated into frameworks such as stable-baseline that we use.
>  - Overall across 20 Atari games, the objectives in our proposed framework achieve performance comparable to $\texttt{Adam LMCDQN}$ (see Figures 19 and 20 in Appendix E.7).
>
> > "Regarding Equation (3), while it is referred to as the “natural policy gradient (NPG),” I do not believe this terminology is appropriate. Although both the solution in Equation (3) and NPG can be derived from reward maximization under a KL-divergence constraint, the solution represents a non-parametric solution to the reinforcement learning problem (Peters
>     et al., 2010; Abdolmaleki et al., 2018). In contrast, NPG refers to the gradient direction of policy updates defined via the Fisher information matrix. While I acknowledge the conceptual connections, I am not convinced that Equation (3)
>     should be described as NPG."
> - We respectfully disagree. Several high-impact, peer-reviewed works in theoretical RL refer to the update in Eq. 3 as the natural policy gradient. For example, [9] consistently describe this KL-regularized update as $\texttt{NPG}$ (see Algorithm 1 and Lemma 10). Likewise, Lemma 5.1 in [10] and Algorithm 1 in [11] explicitly use the same terminology for the same update. These works adopt the term $\texttt{NPG}$ in the non-parametric sense, referring to the KL-constrained mirror-ascent update in policy space. Our terminology follows this established convention.
>
> >"Additionally, the term “proposed methods” appears to refer to algorithms based on the objective functions shown in
>     lines 251–262. However, these algorithms have already been introduced in prior studies, and it may not be appropriate
>     to refer to them as “proposed.”"
> - We acknowledge the feedback and will adjust the phrasing: we intended to refer to the proposed **framework**, not to claim that all resulting objectives are entirely new. In the revised version, we will use the term "proposed framework" to avoid any ambiguity.
> - In particular, we propose a unifying framework, which systematically recovers existing objectives **while designing new and principled ones**. We briefly summarize lines 279-293 from the paper here. First, the decoupling of actor and critic entropy is novel even for objectives recovered through our framework, such as $\texttt{DSAC}$ and off-policy $\texttt{MDPO}$ [12], and we observe the empirical benefit of this decoupling (e.g., Fig. 1). Second, $\texttt{SPMARKL}$ (with or without entropy) and $\texttt{SPMAFKL}(\tau, \zeta)$ are novel algorithms. Finally, under function approximation, the update $\texttt{NPGFKL}(\tau, \zeta)$ obtained from our framework differs from [13] even when $\zeta = \tau$.

---

> ### Author Response · Authors · 2025-11-22
>
> > "The experimental results are reported based on the parameters $\zeta$ and $\tau$ . However, this notation is not intuitive, and I found it necessary to reread multiple sections to fully understand the meaning of the results."
> - Our work decouples the entropy coefficient for the actor (denoted by $\tau$) and the one for the critic (denoted by $\zeta$). As described on lines 248-249, each method is labelled with a specific choice of $(\tau, \zeta)$ to make the exact configuration explicit; for example, $\texttt{DSAC}(\tau, 0)$ denotes the use of actor entropy with coefficient $\tau$ and no critic entropy. In order to help the reader, we will add a brief reminder of this notation in the caption of each figure in the camera-ready version. If the reviewer has suggestions for making the notation more clear, we would be happy to incorporate their suggestion.
>
> > "I am not sure why SPMA is employed as the baseline. NPG is much more well-known than SPMA, and the performance of NPG-based methods appear to be better than those based on SPMA. Please explain why the analysis based on SPMA was necessary."
> - In the on-policy setting $\texttt{SPMAFKL}(0, 0)$ achieves better or comparable performance to $\texttt{NPGRKL}(0, 0)$ across various benchmarks [14]. This motivated us to study its behaviour in the off-policy setting as well. Moreover, $\texttt{SPMA}$ offers properties that make it a useful intermediate policy candidate (lines 477-481).
>     - Unlike $\texttt{NPG}$, when $\zeta=0$ and $\eta$ is sufficiently small, $\texttt{SPMAFKL}(\tau, 0)$ does not require a normalization across the action space, making it more computationally efficient in large discrete action spaces.
>     - For log-linear polices unlike the $\texttt{NPG}$-based objectives, $\texttt{SPMAFKL}(0, 0)$ results in logistic regression and is therefore  convex w.r.t $\theta$. Using this insight, we can provide convergence guarantees for $\texttt{SPMAFKL}(0, 0)$ in the linear function approximation setting (see our response to Reviewer 1).
> - These properties make $\texttt{SPMA}$ an interesting candidate to study in depth.
>
> [1] Christodoulou, P., 2019. Soft actor-critic for discrete action settings. arXiv preprint arXiv:1910.07207.
>
> [2] Zhou, H., Lin, Z., Li, J., Fu, Q., Yang, W. and Ye, D., 2022. Revisiting discrete soft actor-critic, TMLR 2024.
>
> [3] Xu, Y., Hu, D., Liang, L., McAleer, S., Abbeel, P. and Fox, R., 2021. Target entropy annealing for discrete soft actor-critic. arXiv preprint arXiv:2112.02852.
>
> [4] Ishfaq, H., Lan, Q., Xu, P., Mahmood, A.R., Precup, D., Anandkumar, A. and Azizzadenesheli, K., 2023. Provable and practical: Efficient exploration in reinforcement learning via langevin monte carlo, ICLR 2024.
>
> [5] Fortunato, M., Azar, M.G., Piot, B., Menick, J., Osband, I., Graves, A., Mnih, V., Munos, R., Hassabis, D., Pietquin, O. and Blundell, C., 2017. Noisy networks for exploration. arXiv 2017. arXiv preprint arXiv:1706.10295.
>
> [6] Dabney, W., Rowland, M., Bellemare, M. and Munos, R., 2018, April. Distributional reinforcement learning with quantile regression. In Proceedings of the AAAI conference on artificial intelligence (Vol. 32, No. 1).
>
> [7] Dabney, W., Ostrovski, G., Silver, D. and Munos, R., 2018, July. Implicit quantile networks for distributional reinforcement learning. In International conference on machine learning (pp. 1096-1105). PMLR.
>
> [8] Bellemare, M.G., Dabney, W. and Munos, R., 2017, July. A distributional perspective on reinforcement learning. In International conference on machine learning (pp. 449-458). PMLR.
>
> [9] Liu, Q., Weisz, G., György, A., Jin, C. and Szepesvári, C., 2023. Optimistic natural policy gradient: a simple efficient policy optimization framework for online rl. Advances in Neural Information Processing Systems, 36, pp.3560-3577.
>
> [10] Agarwal, A., Kakade, S.M., Lee, J.D. and Mahajan, G., 2021. On the theory of policy gradient methods: Optimality, approximation, and distribution shift. Journal of Machine Learning Research, 22(98), pp.1-76.
>
> [11] Cen, S., Cheng, C., Chen, Y., Wei, Y. and Chi, Y., 2022. Fast global convergence of natural policy gradient methods with entropy regularization. Operations Research, 70(4), pp.2563-2578.
>
> [12] Tomar, M., Shani, L., Efroni, Y. and Ghavamzadeh, M., 2020. Mirror descent policy optimization, ICLR 2022.
>
> [13] Mei, J., Xiao, C., Huang, R., Schuurmans, D. and Müller, M., 2019, August. On principled entropy exploration in policy optimization. In Proceedings of the 28th International Joint Conference on Artificial Intelligence (pp. 3130-3136).
>
> [14] Asad, R., Babanezhad, R., Laradji, I., Roux, N.L. and Vaswani, S., 2024. Fast convergence of softmax policy mirror ascent, AISTATS 2025.

---

> > ### Author Response · Authors · 2025-11-22
> >
> > > "One important factor is that, in the discrete action setting, the Bellman optimality operator can be directly applied because the maximum over actions can be explicitly computed. This is not feasible in the continuous action domain.This fundamental difference may explain why DQN variants are typically preferred for discrete action tasks, whereas off-policy actor-critic methods are more suitable for continuous action domains."
> >
> > > "However, this important discussion appears to be missing from the paper. I would like to ask the authors for their thoughts on this point."
> >
> > - We agree with the reviewer that, unlike the continuous action space, in the discrete-action setting, one can directly apply max-backup targets for learning the $Q$ function. This is well-known and is why we did not explicitly mention it in the paper. However, there exist important applications where policy optimization methods are used in discrete action settings. For example,
> >     1. For text generation and Reinforcement Learning with Human Feedback (RLHF), the action is a token from a very large vocabulary and the state-of-the-art RLHF pipelines [15, 16, 17] use policy optimization methods such as $\texttt{PPO}$, $\texttt{GRPO}$, or REINFORCE rather than a $Q$-based argmax over the entire action space.
> >     2. In recommender systems, the action is a slate (an ordered list of items) presented to a user. For practical applications, policy optimization methods such as REINFORCE [18] or actor-critic methods [19, 20] are commonly employed.
> > - These examples underscore the importance of policy optimization in discrete-action environments. To the best of our knowledge, no **off-policy discrete-action actor-critic** achieves competitive performance even with vanilla $\texttt{DQN}$. Consequently, we believe this is an important problem and our work is an initial step towards resolving it.
> >
> > [15] Ouyang, L., Wu, J., Jiang, X., Almeida, D., Wainwright, C., Mishkin, P., Zhang, C., Agarwal, S., Slama, K., Ray, A. and Schulman, J., 2022. Training language models to follow instructions with human feedback. Advances in neural information processing systems, 35, pp.27730-27744.
> >
> > [16] Shao, Z., Wang, P., Zhu, Q., Xu, R., Song, J., Bi, X., Zhang, H., Zhang, M., Li, Y.K., Wu, Y. and Guo, D., 2024. Deepseekmath: Pushing the limits of mathematical reasoning in open language models, 2024. URL https://arxiv. org/abs/2402.03300, 2(3), p.5.
> >
> > [17] Ahmadian, A., Cremer, C., Gallé, M., Fadaee, M., Kreutzer, J., Pietquin, O., Üstün, A. and Hooker, S., 2024. Back to basics: Revisiting reinforce style optimization for learning from human feedback in llms. arXiv preprint arXiv:2402.14740.
> >
> > [18] Chen, M., Beutel, A., Covington, P., Jain, S., Belletti, F. and Chi, E.H., 2019, January. Top-k off-policy correction for a REINFORCE recommender system. In Proceedings of the twelfth ACM international conference on web search and data mining (pp. 456-464).
> >
> > [19] Zhao, X., Li, J., Fang, Z., Guo, Y., Zhao, J., He, J., Chen, W., Peng, C. and Ding, G., 2024, May. JDRec: Practical Actor-Critic Framework for Online Combinatorial Recommender System. In Proceedings of the 23rd International Conference on Autonomous Agents and Multiagent Systems (pp. 2612-2614).
> >
> > [20] Xin, X., Karatzoglou, A., Arapakis, I. and Jose, J.M., 2022, February. Supervised advantage actor-critic for recommender systems. In Proceedings of the fifteenth ACM international conference on web search and data mining (pp. 1186-1196).

---

> > > ### Comment · Reviewer_7fk8 · 2025-11-27
> > >
> > > Thank you for your detailed response.
> > >
> > > While the authors refer to RLHF for LLMs as a motivation for their work, I find the explanation unconvincing. The learning dynamics of large-scale models differ significantly from those of smaller models. Recent studies, such as [1], highlight the challenges of applying off-policy RL to LLMs. It is unclear how the findings in this paper contribute to that domain.
> > > If the authors intend to motivate their work based on LLMs, they should evaluate their framework using LLMs.
> > >
> > > [1] Noukhovitch et al., Asynchronous RLHF: Faster and More Efficient Off-Policy RL for Language Models. ICLR 2025.
> > >
> > > In this regard, I share a similar perspective with Reviewer Ifti. Overall, the authors seem to address a rather narrow question, and the contribution of this paper appears fairly limited.

---

> > > > ### Author Response · Authors · 2025-11-28
> > > >
> > > > Dear reviewer,
> > > >
> > > > Thank you for the response. We believe there is a misunderstanding. The paper does not claim that RLHF for LLMs is the motivation of our work. Our comment in the discussion was specifically addressing your earlier statement that “DQN variants are typically preferred for discrete action tasks, whereas actor-critic methods are more suitable for continuous action domains.”
> > > >
> > > > Our aim was to highlight that actor-critic methods are also widely used in important discrete-action applications. RLHF for LLMs and recommender systems were mentioned purely as examples of such settings. There are several other domains, such as constrained MDPs, in which policy optimization is the standard approach. These examples were intended to emphasize that discrete-action actor-critic methods are relevant and practically important.
> > > >
> > > > We hope this clarification resolves the misunderstanding, and we apologize for any confusion.

---

### Official Review · Reviewer_HpKy · 2025-10-27

**Soundness:** 3
**Presentation:** 2
**Contribution:** 2
**Rating:** 4
**Confidence:** 4

**Summary:**

This paper proposed a generalized actor critic framework extending discrete SAC with a focus on discrete actions. The authors considered various possibilities such as different entropy coefficients for critic and actor, NPG/SPMA + forward/reverse KL divergence in the greedy step. The authors empirically surveyed these choices and the number of actor updates as well as auto coefficient tuning to give concrete conclusions on what combination might be more preferrable. Theoretical justifications were also provided with the same rate as prior work.

**Strengths:**

This paper is a nice summary of entropy regularized actor critic in discrete actions. The proposed general off-policy actor-critic framework is sufficiently large as a result of carefully distilling prior work. I work in entropy regularized RL and I appreciate that the authors comprehensively and accurately listed related papers along the way of developing their method. Decoupling of actor/critic entropy coefficient, NPG/SPMA for greedy step, the number of actor updates have all been discussed and experimented. The conclusions are to the point and of interest to the community.

**Weaknesses:**

Compared to the technical sections (though techniques from prior works are heavily used), the first section fails to deliver a convincing motivation. For example, in the abstract:
```
Common policy-based methods are either on-policy and do not effectively learn from off-policy data or have poor empirical performance in the discrete-action setting.
```
claims like this seem to be overreaching and are not well supported by the paper. Did the authors take methods like Greedy-AC/InAC/TAWAC [1-3] into account? These methods have shown good performance in the discrete case as well and do not necessarily introduce entropy. Motivating entropy and therefore DSAC as a starting point for the proposed general family is reasonable, but it is not well connected to the claim that existing methods are flawed in discrete actions. I do think discrete-action AC is important, but I believe the current story needs to be modified to avoid criticizing all existing ACs. The motivation can come from motivating e.g. RL for healthcare that necessitates combinations of treatment.

**References:**\
[1] The In-Sample Softmax for Offline Reinforcement Learning, ICLR 2023\
[2] Greedy Actor-Critic: A New Conditional Cross-Entropy Method for Policy Improvement, ICLR 2023\
[3] q Exponential Family Policy Optimization, ICLR 2025

**Questions:**

1. M-DQN has their actor-critic version [1], have you compared to it?
2. have the authors tried other actor critic methods like GreedyAC/InAC/TAWAC to see how they perform against DQN?
3. the authors claimed SPMA-RKL$(\zeta, \tau)$ is a novel method, but I can't find it in the experiments
4. in Figure 2 why DSAC-auto-ent$(\tau, \tau)$ is bad on the left column but good on the right?


**References:** \
[1] Implicitly Regularized RL with Implicit Q-values

---

> ### Author Response · Authors · 2025-11-22
>
> We thank the reviewer for their feedback and address their main questions and comments below.
> > "have the authors tried other actor critic methods like GreedyAC/InAC/TAWAC to see how they perform against DQN?"
> - Thank you for the suggestion. InAC [1] is designed for the offline RL setting, whereas our work focuses on online RL with discrete-actions and off-policy learning. Extending our framework to the offline regime would be a valuable future direction, especially since InAC highlights a complementary issue arising from max-backup targets in offline datasets (i.e., sampling actions that have not been seen in the dataset).
>
> - Regarding TAWAC [2], its analysis and experiments focus on continuous action spaces with **Gaussian policy** parameterizations and **$q$-exponential** policies. Our contribution is centered on the **discrete** action setting, motivated by understanding the failure modes of $\texttt{DSAC}$ and developing a general discrete-action actor-critic framework. Because both InAC and TAWAC operate in settings orthogonal to our focus, direct comparisons would not be meaningful.
>
> - On the other hand, $\texttt{GreedyAC}$ [3] can indeed be applied to discrete-action settings, and the original paper includes experiments on such tasks. However, these experiments are limited to MuJoCo's toy discrete environments (Acrobot, MountainCar, Pendulum, each with three actions) and two MinAtar games, which provide a substantially easier setting than the original Atari framework [4] (e.g., the observation space is reduced from a $210\times160\times3$ image to a $10\times10$ grid with binary object indicators). Moreover, these experiments do not include comparisons against $\texttt{DQN}$. To avoid potential discrepancies arising from reimplementation, we evaluated our empirical observation on $\texttt{DSAC}$ directly within $\texttt{GreedyAC}$'s public codebase. The codebase, however, only provides the MLP policy used for MuJoCo’s environments. Because the MinAtar policy is not included, and to avoid modifying the authors’ implementation, we restricted our evaluation to MuJoCo’s discrete tasks. Our findings are summarized below.
>     - The $\texttt{DSAC}$ implementation in $\texttt{GreedyAC}$ does not model the $Q$ function in a way that matches the formulation in the original $\texttt{DSAC}$ paper [5]. Whereas $\texttt{SAC}$'s $Q$ function takes sampled state-action pairs, **the discrete extension should take only sampled states and output a value for each action** (see Eq. 8 versus Eq. 12 in $\texttt{DSAC}$ [5]). We modified the $Q$ function to match the original $\texttt{DSAC}$.
>     - $\texttt{GreedyAC}$'s codebase does not include the entropy coefficient loss for $\texttt{DSAC}$, and instead the coefficient is tuned as a fixed constant. We follow this convention and leave the code unchanged.
>     - Our results (Figures 22 and 23 in Appendix E.9) show that by simply correcting the $\texttt{DSAC}$ implementation while using a fixed well-tuned entropy coefficient (with or without critic entropy, over the same grid search reported in the paper), yields performance comparable to $\texttt{GreedyAC}$. This observation aligns with our empirical finding on lines 380-385. For reference, Fig. 22 in our revised paper can be directly compared to Fig. 2 in $\texttt{GreedyAC}$.
> - Finally, we reiterate that the key observation from the $\texttt{DSAC}$ ablation in our paper is that when using $\texttt{DSAC}$'s adaptive entropy scheme (via an entropy coefficient loss), simply removing the critic entropy substantially improves performance *without requiring any per-environment tuning of the coefficient*.
>
> > "M-DQN has their actor-critic version, have you compared to it?"
> - The paper referenced by the reviewer extends $\texttt{MDQN}$ [9] to **continuous** action spaces by defining an implicit $Q$ function of the form $Q_{\theta,\phi}(s,a) = \tau \ln \pi_\theta(a|s) + V_\phi(s)$, and using it to define a joint loss over the actor and critic parameters. All experiments in the cited paper are conducted in continuous control and the method is positioned as an alternative to $\texttt{SAC}$. In contrast, our focus is entirely on the **discrete** action setting and specifically addressing the poor performance of $\texttt{DSAC}$. Because the objectives and experimental domains do not overlap, a direct empirical comparison would not be meaningful.

---

> > ### Author Response · Authors · 2025-11-22
> >
> > > "Motivating entropy and therefore DSAC as a starting point for the proposed
> >     general family is reasonable, but it is not well connected to the claim that existing methods are flawed in discrete
> >     actions. I do think discrete-action AC is important, but I believe the current story needs to be modified to avoid
> >     criticizing all existing ACs."
> >
> > - We clarify that our critique is directed specifically at $\texttt{DSAC}$ [5] and its follow-up work [6, 7], rather than at all discrete-action actor-critic methods (see lines 30-53). Our motivation is to start from the well-established method $\texttt{SAC}$ [8] that performs strongly in continuous action space, and to understand why its discrete-action counterpart underperforms even vanilla $\texttt{DQN}$.
> >
> > - To the reviewer’s point, we will adjust the abstract, clarify this further in the revised version, and avoid generalizing our critique to *all actor-critic* approaches. Having said that, we are not aware of an off-policy actor-critic method that matches the performance of $\texttt{DQN}$ on the Atari benchmark. The papers pointed out by the reviewer either focus on the continuous action domain or do not provide evaluation on a large and diverse set of Atari games, instead reporting results on toy discrete action environments (e.g., the discrete version of MountainCar, Pendulum, and Acrobot in MuJoCo with only three actions)
> >
> > > "The authors claimed $\texttt{SPMARKL}(\zeta, \tau)$ is a novel method, but I can't find it in the experiments"
> > - The $\texttt{SPMARKL}$ objective is novel both with and without entropy. Consistent with our empirical observations on $\texttt{DSAC}$, we found that choosing $\zeta = 0$ compared to $\zeta = \tau$ significantly boosts the performance of $\texttt{SPMARKL}$. Please see the third row of Fig. 5 in Appendix E.3 comparing $\texttt{SPMARKL}(\tau, \tau)$ and $\texttt{SPMARKL}(\tau, 0)$.
> >
> > > in Figure 2 why $\texttt{DSAC}$-auto-ent $(\tau , \tau)$ is bad on the left column but good on the right?
> > - When ablating $\texttt{DSAC}$ on four Atari games in Fig. 2 we observed:
> >     1. Default $\texttt{DSAC}$ (the orange curve, $\texttt{DSAC}$-auto-ent $(\tau,\tau)$) that uses an adaptive entropy coefficient performs well on some games but poorly on others.
> >     2. A fixed, well-tuned entropy coefficient per game yields consistent good performance (red curve).
> >     3. Setting the critic entropy in default $\texttt{DSAC}$ to zero also results in consistent good performance without the need for a per-game tuned coefficient (green curve).
> > - From observations 1 and 2, we concluded that the adaptive entropy scheme is the source of instability. Further analysis shows that for the two games with poor performance (left column of Fig. 2), the policy entropy collapses; we have added this plot in Appendix E.10 (see Fig. 24). Prior work [7] identifies a similar issue and proposes an adaptive target entropy to counter entropy collapse or excessive entropy, but this introduces additional hyper-parameters. In contrast, from observation 3, we find that simply removing the critic entropy resolves this failure mode without requiring extra hyper-parameters to tune or the need to modify $\texttt{DSAC}$'s objective. This motivates Fig. 1, where observation 3 on four Atari games generalizes across 20 games, significantly improving $\texttt{DSAC}$'s performance.
> >
> > [1] Xiao, C., Wang, H., Pan, Y., White, A. and White, M., 2023. The in-sample softmax for offline reinforcement learning, ICLR 2023.
> >
> > [2] Zhu, L., Shah, H., Wang, H., Nagai, Y. and White, M., 2024. q-exponential family for policy optimization, ICLR 2025.
> >
> > [3] Neumann, S., Lim, S., Joseph, A., Pan, Y., White, A. and White, M., 2018. Greedy actor-critic: A new conditional cross-entropy method for policy improvement, ICLR 2023.
> >
> > [4] Young, K. and Tian, T., 2019. Minatar: An atari-inspired testbed for thorough and reproducible reinforcement learning experiments. arXiv preprint arXiv:1903.03176.
> >
> > [5] Christodoulou, P., 2019. Soft actor-critic for discrete action settings. arXiv preprint arXiv:1910.07207.
> >
> > [6] Zhou, H., Lin, Z., Li, J., Fu, Q., Yang, W. and Ye, D., 2022. Revisiting discrete soft actor-critic, TMLR 2024.
> >
> > [7] Xu, Y., Hu, D., Liang, L., McAleer, S., Abbeel, P. and Fox, R., 2021. Target entropy annealing for discrete soft actor-critic. arXiv preprint arXiv:2112.02852.
> >
> > [8] Haarnoja, T., Zhou, A., Abbeel, P. and Levine, S., 2018, July. Soft actor-critic: Off-policy maximum entropy deep reinforcement learning with a stochastic actor. In International conference on machine learning (pp. 1861-1870). Pmlr.
> >
> > [9] Vieillard, N., Pietquin, O. and Geist, M., 2020. Munchausen reinforcement learning. Advances in Neural Information Processing Systems, 33, pp.4235-4246.

---

> > > ### Comment · Reviewer_HpKy · 2025-11-26
> > >
> > > Thank you for the detailed response.
> > >
> > > I believe during rebuttal it is allowed to have 10 pages in total. This room can be used to accommodate the new results and discussion we had above. Specifically, I would suggest include the following:
> > >  - different setting of the existing ACs like InAC, TAWAC, GreedyAC etc. It is good that the authors have compared against it.
> > > -  clarification why the extension of M-DQN is not considered
> > > -  new results of SPMARKL
> > >
> > > I will consider raising my score if the authors could explicitly reflect these changes in the updated draft

---

> > > > ### Author Response · Authors · 2025-11-27
> > > >
> > > > Dear reviewer,
> > > >
> > > > Thank you for your thoughtful feedback. We have revised our submission to address your concerns, highlighted in red. In particular:
> > > > - We modified the abstract and introduction to incorporate your suggested references, including $\texttt{InAC}$, $\texttt{TAWAC}$, and $\texttt{GreedyAC}$, clarifying how they differ from our method. We included the plots described in our rebuttal to the main paper comparing $\texttt{DSAC}$ and $\texttt{GreedyAC}$ (section 5, conclusion 1, and Fig. 3).
> > > > - We added a clarification in the introduction on why the extension of $\texttt{M-DQN}$ is not considered in our work and how it differs from our framework.
> > > > - We included plots highlighting the novel objectives we introduce, including $\texttt{SPMARKL}$ and $\texttt{SPMAFKL}$ (section 5, conclusion 2, and Fig. 4).
> > > >
> > > > We hope these revisions address your concerns. Please let us know if there are any additional questions or clarifications that could further assist in your evaluation.

---

### Official Review · Reviewer_ifti · 2025-11-02

**Soundness:** 3
**Presentation:** 3
**Contribution:** 3
**Rating:** 6
**Confidence:** 3

**Summary:**

This paper proposes a well-organized framework for discrete action off-policy actor–critic that decouples the actor and critic entropies, spans choices like KL direction (FKL/RKL), update style (NPG/SPMA), and m-step evaluation, and recovers several prior families as special cases. The empirical story leans on using this framework to diagnose DSAC and related design knobs—showing, for instance, that turning off critic entropy while keeping actor entropy helps a lot—and offering practical guidance on KL direction and entropy strength for discrete control. The theory is modular but for the tabular setting. It gives bounds that help reason about the design space but don’t address the non-linear function-approximation regime. The empirical results are competitive with DQN in many games, though notable gaps remain (e.g., Pong).

**Strengths:**

- This is an interesting paper that proposes a nice framework that can recover some prior families of algorithms. I am always happy to see well-organized papers that unify certain methods in a good framework.

- The core empirical results and discussions focus on using the proposed framework—which decouples the critic and actor’s entropy regularization—to examine DSAC and choices like KL direction, strength of entropy, etc. These provide nice insights with practical implications for using off-policy actor-critic methods in discrete settings. In that sense, I see the value of the paper for applying these methods to problems. It improves our understanding in this setting. I am personally in favor of papers that improve our understanding of existing methods in practice.

- The theoretical analysis in tabular settings seem to imply that the proposed framework and its bounds can match existing bounds as special instances. This has the added benefit of unifying different theoretical results / bounds under a single result, which I again appreciate.

**Weaknesses:**

- Theoretical results are restricted to the tabular setting, which is unrealistic. This kind of matches the existing theory, but still the setting of the paper is "discrete action" but not "tabular", so it is a bit disappointing.

- The paper does not necessarily position itself as a novel algorithm/approach compared to baselines; the contribution is more diagnostic/unifying than algorithmically new. This does not personally bother me, since I appreciate papers that improve our understanding. However it must be said.

- Vanilla DQN is perhaps okay for making the paper's statement. However, when it comes to Atari, I would have liked seeing a stronger DQN-variant. Looking at Rainbow, we can see how much better Rainbow or even Distributional DQN, Dueling DDQN etc. are compared to vanilla DQN. This makes me wonder if the results we see here would still hold against a much stronger DQN-variant than vanilla DQN.

- There is no explicit Related Works section, connecting the paper into existing research.

**Questions:**

1. You state that Vieillard et al. (2020a) gives an $O(\frac{1}{K})$ for the $\zeta = \tau$ and $m=1$ case, but it is not immediately clear to me what bound we would get from your result for this setting. Can you confirm that you match this? If not, which term is the culprit here?

2. It seems that no matter what you have tried, DSAC or any other actor-critic variant cannot match DQN in Pong, and the gap is substantial. In your opinion, what could be the reason?

3. What was the reason for using vanilla DQN in your comparisons? Please correct me if you have not used vanilla DQN. There are much stronger DQN variants for Atari, so this makes the relative performance comparisons suspect.

---

> ### Author Response · Authors · 2025-11-22
>
> We thank the reviewer for their positive review and address their main questions and comments below.
> > "Theoretical results are restricted to the tabular setting, which is unrealistic. This kind of matches the existing theory, but still the setting of the paper is discrete action but not tabular, so it is a bit disappointing."
>
> - First, we emphasize that even in the tabular setting, our theoretical results are novel. In particular, as explained in lines 296-297, our theoretical results capture decoupled actor and critic entropy (a setting that leads to strong empirical results), $\\texttt{SPMA}$ (an alternative algorithm to $\texttt{NPG}$) and can recover existing results in special cases.
>
> - Having said that, we agree that extending our theory beyond the tabular setting is indeed an important future direction. In order to theoretically handle function approximation in the future, we designed our framework to be modular -- combining different choices of the intermediate tabular policy and decoupled it from the projection step (which depends on the function approximation scheme). As an example, consider using $\texttt{SPMAFKL}(0,0)$ (i.e., without entropy regularization, corresponding to results in the third column of Fig. 3) *with log-linear policies*. Since we have already done the proof for the tabular variant (Theorem 8 in Appendix C.4.2), in order to theoretically analyze this variant, we need to analyze the effect of the projection step. Since the $\texttt{SPMAFKL}(0,0)$ results in a convex objective in the log-linear case, we can quantify the projection error similar to [1] with  modifications to handle the off-policy setting. However, handling the entropy regularization and the other variants is non-trivial. Since doing a comprehensive theoretical analysis is tangential to the main contribution of this paper, we leave this interesting direction to future work.
>
> > "It seems that no matter what you have tried, DSAC or any other actor-critic variant cannot match DQN in Pong, and the gap is substantial. In your opinion, what could be the reason?"
>
> - Thank you for the great observation. We investigated Pong for $\texttt{DSAC}$ to diagnose why (even without critic entropy), its performance remains worse than $\texttt{DQN}$. Fig. 21 in Appendix E.8 shows that using a small *fixed* entropy coefficient and increasing the number of actor gradient updates (to reduce the function approximation error) substantially improves performance. Finally, we reiterate that the goal of our paper was not to tune performance on individual games. Instead, our objective was to resolve the underlying issue behind $\texttt{DSAC}$'s poor performance relative to even vanilla $\texttt{DQN}$ across a large and diverse range of Atari games, and to use this insight to develop a principled actor-critic framework for discrete action off-policy RL.
>
> > "What was the reason for using vanilla DQN in your comparisons? Please correct me if you have not used vanilla DQN. There are much stronger DQN variants for Atari, so this makes the relative performance comparisons suspect."
>
> - We emphasize that our goal is not to compete with the full suite of highly engineered $\texttt{DQN}$ variants, but to understand why $\texttt{DSAC}$ underperforms even vanilla $\texttt{DQN}$ [2, 3]. We use this insight to build a principled actor-critic framework for the off-policy discrete-action setting. Given the importance of policy optimization methods to applications including aligning LLMs, we believe that this is an important research direction. We agree that incorporating heuristics (similar to recent variants of $\texttt{DQN}$) and improving the performance of $\texttt{DSAC}$ is an interesting direction for future work. In order to do so, we believe that our paper takes an important first step.
> - To further address the reviewer's concern, we also compared our objectives to a recent strong baseline, $\texttt{Adam LMCDQN}$ [4]. We chose this baseline for two reasons:
>     1. $\texttt{Adam LMCDQN}$ outperforms or is on par with strong methods such as $\texttt{NoisyNetDQN}$ [5], $\texttt{QRDQN}$ [6], $\texttt{IQN}$ [7], and $\texttt{C51}$ [8]
>     2. $\texttt{Adam LMCDQN}$ does not use $\epsilon$-greedy and is implemented as a modified Adam optimizer that can be easily integrated into frameworks such as stable-baseline that we use.
>  - Overall across 20 Atari games, the objectives in our proposed framework achieve performance comparable to $\texttt{Adam LMCDQN}$ (see Figures 19 and 20 in Appendix E.7).

---

> ### Author Response · Authors · 2025-11-22
>
> > "The paper does not necessarily position itself as a novel algorithm/approach compared to baselines"
> - Our framework unifies several existing objectives while also introducing novel ones. For clarity, we briefly summarize lines 279-293 from the paper here. In particular, we recover prior objectives such as $\texttt{DSAC}$ [9] and off-policy $\texttt{MDPO}$ [10]. However, the decoupling of actor and critic entropy regularization is a novel contribution, and we demonstrate its empirical benefit (e.g., Fig. 1). In addition, both $\\texttt{SPMARKL}$ (with or without entropy) and $\\texttt{SPMAFKL}(\tau,\zeta)$ are novel algorithms introduced by our framework. Finally, in the function approximation setting, the proposed $\\texttt{NPGFKL}(\tau,\zeta)$ update is different from the formulation of [11], even in the case $\zeta=\tau$.
>
> > "There is no explicit Related Works section, connecting the paper into existing research."
> - Since the related work for our paper is a combination of empirical and theoretical works, we decided to integrate it into the introduction and algorithm sections to provide a more natural flow. For example, we motivate the paper from an empirical perspective (by introducing $\texttt{DSAC}$) and compare it to the relevant empirical work such as [2, 3, 12]. When abstracting things out and designing our tabular framework, we compare it to the more theoretical papers, including [13, 14]. Finally, when incorporating function approximation, we again compare to the more empirical policy optimization papers such as [1, 10, 11]. We believe that this approach leads to a more coherent narrative, while citing all the relevant work.
>
> > "You state that Vieillard et al. (2020a) gives an $O(1/K)$ for the $\zeta=\tau$ and $m = 1$ case, but it is not immediately clear to me what bound we would get from your result for this setting. Can you confirm that you match this? If not, which term is the culprit here?"
>
> - Yes, we match the result in Vieillard et al. (2020a). As summarized on line 352, our sub-optimality in Theorem 2 can be bounded as $O(\frac{1}{K} + \frac{\gamma^m}{K} + \delta(\tau, \zeta))$ and for $\zeta = \tau$ (which yields $\delta(\tau, \zeta) = 0$) and $m=1$, we recover the $O(1/K)$ rate of Vieillard et al. (2020a).
>
>
> [1] Asad, R., Babanezhad, R., Laradji, I., Roux, N.L. and Vaswani, S., 2024. Fast convergence of softmax policy mirror ascent, AISTATS 2025.
>
> [2] Zhou, H., Lin, Z., Li, J., Fu, Q., Yang, W. and Ye, D., 2022. Revisiting discrete soft actor-critic, TMLR 2024.
>
> [3] Xu, Y., Hu, D., Liang, L., McAleer, S., Abbeel, P. and Fox, R., 2021. Target entropy annealing for discrete soft actor-critic. arXiv preprint arXiv:2112.02852.
>
> [4] Ishfaq, H., Lan, Q., Xu, P., Mahmood, A.R., Precup, D., Anandkumar, A. and Azizzadenesheli, K., 2023. Provable and practical: Efficient exploration in reinforcement learning via langevin monte carlo, ICLR 2024.
>
> [5] Fortunato, M., Azar, M.G., Piot, B., Menick, J., Osband, I., Graves, A., Mnih, V., Munos, R., Hassabis, D., Pietquin, O. and Blundell, C., 2017. Noisy networks for exploration. arXiv 2017. arXiv preprint arXiv:1706.10295.
>
> [6] Dabney, W., Rowland, M., Bellemare, M. and Munos, R., 2018, April. Distributional reinforcement learning with quantile regression. In Proceedings of the AAAI conference on artificial intelligence (Vol. 32, No. 1).
>
> [7] Dabney, W., Ostrovski, G., Silver, D. and Munos, R., 2018, July. Implicit quantile networks for distributional reinforcement learning. In International conference on machine learning (pp. 1096-1105). PMLR.
>
> [8] Bellemare, M.G., Dabney, W. and Munos, R., 2017, July. A distributional perspective on reinforcement learning. In International conference on machine learning (pp. 449-458). PMLR.
>
> [9] Christodoulou, P., 2019. Soft actor-critic for discrete action settings. arXiv preprint arXiv:1910.07207.
>
> [10] Tomar, M., Shani, L., Efroni, Y. and Ghavamzadeh, M., 2020. Mirror descent policy optimization, ICLR 2022.
>
> [11] Mei, J., Xiao, C., Huang, R., Schuurmans, D. and Müller, M., 2019, August. On principled entropy exploration in policy optimization. In Proceedings of the 28th International Joint Conference on Artificial Intelligence (pp. 3130-3136).
>
> [12] Wang, Z., Bapst, V., Heess, N., Mnih, V., Munos, R., Kavukcuoglu, K. and De Freitas, N., 2016. Sample efficient actor-critic with experience replay. arXiv preprint arXiv:1611.01224.
>
> [13] Vieillard, N., Kozuno, T., Scherrer, B., Pietquin, O., Munos, R. and Geist, M., 2020. Leverage the average: an analysis of kl regularization in reinforcement learning. Advances in Neural Information Processing Systems, 33, pp.12163-12174.
>
> [14] Xiao, L., 2022. On the convergence rates of policy gradient methods. Journal of Machine Learning Research, 23(282), pp.1-36.

---

> > ### Comment · Reviewer_ifti · 2025-11-25
> >
> > Dear authors,
> >
> > Thank you for your detailed response. I believe it is an interesting question to study why discrete SAC cannot match the performance of value-based approaches. However, this in itself is already a very narrow and specific question. From the paper and the rebuttal, you are narrowing this even more by scoping the paper's goal as "to understand why DSAC underperforms even vanilla DQN". What if your approach matches vanilla DQN's perspective, but gets defeated by more advanced DQN variants? What does this tell us? It seems the Adam LMCDQN results are quite different compared to the vanilla DQN results.
> >
> > In my opinion, studying why discrete AC methods under-perform against value-based methods is an interesting yet narrow question. A good and general answer to this would be valuable. However, narrowing it further to "under-perform against vanilla DQN", the question becomes too narrow.
> >
> > I will maintain my score for now, but I am open to discussions with fellow reviewers. In any case, I look forward to reading the improved version.

---

> ### Author Response · Authors · 2025-11-26
>
> Dear Reviewer,
>
> Thank you for your thoughtful engagement. Given the importance of policy optimization in discrete-action environments, including those in recommender systems and RL with LLMs (see our response to Reviewer 3 for references), we believe that it is essential to close the performance gap between actor-critic and value-based methods in large-scale off-policy settings. *To the best of our knowledge, no policy-based method can compete even with $\texttt{DQN}$ in this large-scale off-policy setup on Atari*. Consequently, the first step is to ensure that the most commonly used actor-critic method ($\texttt{DSAC}$) can perform competitively with the standard value-based baseline ($\texttt{DQN}$).
>
> Regarding your second question, we conducted a comparison against $\texttt{Adam LMCDQN}$ and found that our modified $\texttt{DSAC}$ and the objectives from our proposed framework perform competitively with it. As demonstrated in [4], $\texttt{Adam LMCDQN}$ is a strong baseline and is known to match or outperform several advanced $\texttt{DQN}$ variants.
>
> However, the goal of this paper is not to achieve state-of-the-art results across all $\texttt{DQN}$ variants, nor is it limited to making $\texttt{DSAC}$ competitive with $\texttt{DQN}$. Our work also introduces a general actor-critic framework, together with a novel theoretical foundation, that unifies existing actor objectives and yields new ones. We also present careful ablations examining the effect of forward versus reverse KL divergences in the policy projection step, the impact of multiple actor gradient updates, and the influence of entropy. Notably, our experiments show that entropy is not required for strong performance (see Fig. 3). We view these findings as important contributions that can guide future algorithm design.
>
> We do not believe that getting state-of-the-art results is the only way to contribute to the community. The ICLR reviewer guidelines (https://iclr.cc/Conferences/2026/ReviewerGuide) explicitly state that “What is the significance of the work? Does it contribute new knowledge and sufficient value to the community? Note, this does not necessarily require state-of-the-art results. Submissions bring value to the ICLR community when they convincingly demonstrate new, relevant, impactful knowledge (incl., empirical, theoretical, for practitioners, etc).”
>
> We welcome continued discussion of the scientific claims in our paper and hope that our work provides a useful perspective on the role of actor-critic methods in off-policy RL.

---

### Meta-Review · Area_Chair_tmgC · 2026-01-06

**Summary:**

Theoretical result was limited to the tabular setting, which is not practically useful.

The performance for Atari is far from SoTA.

**Reviewer Concerns:**

The missing references were added by authors, but the overall contributions were not recognised by multiple reviewers.

**Reviewer Scores:**

N/A.

---

### Decision · Program_Chairs · 2026-01-26

Reject